# LA-PROTEINA: ATOMISTIC PROTEIN GENERATION VIA PARTIALLY LATENT FLOW MATCHING

**Tomas Geffner**[1] * **Kieran Didi**[12*] **Zhonglin Cao**[1] **Danny Reidenbach**[1] **Zuobai Zhang**[134]
**Christian Dallago**[1] **Emine Kucukbenli**[1] **Karsten Kreis**[1†] **Arash Vahdat**[1†]

[1]NVIDIA  [2]University of Oxford  [3]Mila - Québec AI Institute  [4]Université de Montréal

*Project page:* https://research.nvidia.com/labs/genair/la-proteina/

## ABSTRACT

While many generative models for protein design have emerged, few tackle the difficult task of jointly generating full-atom structures and their corresponding sequences, a challenge compounded by sequence-dependent side-chain dimensionality. We introduce *La-Proteina* for atomistic protein design based on a novel partially latent representation: the protein backbone is modeled explicitly, while sequence and all other atomistic details are captured in per-residue latent variables of fixed size. Flow matching in this partially latent space then models the joint distribution over sequences and full-atom structures. *La-Proteina* achieves state-of-the-art performance on key generation benchmarks, including all-atom co-designability, diversity, and structural validity. Notably, it also surpasses previous models in atomistic motif scaffolding, unlocking critical structure-conditioned design tasks. Moreover, *La-Proteina* generates co-designable proteins of up to 800 residues, a regime where most baselines collapse or fail to produce valid samples, demonstrating its unique scalability and robustness.

## 1 INTRODUCTION

The design of novel proteins with specific structures and functions has immense potential in various fields (Richardson & Richardson, 1989; Huang et al., 2016; Kuhlman & Bradley, 2019). A challenge in de novo protein design is capturing the relationship between protein sequence and structure. Most existing methods decouple these aspects, generating sequences that are later folded (Wang et al., 2024b) or designing backbones that are subsequently sequenced (Watson et al., 2023). However, accurately modeling the *joint* distribution over sequences and fully atomistic structures could unlock fine-grained control over functional sites and enable key protein design tasks, such as atomistic motif scaffolding. This problem is made inherently difficult by the need to handle both discrete sequences and continuous coordinates, along with the sequence-dependent dimensionality of side chains. Recent methods tackling this problem learn generative models directly in data space (Qu et al., 2024; Chu et al., 2024), though these often struggle with modeling accuracy and scalability. Other approaches use latent representations (Lu et al., 2024; Fu et al., 2024; McPartlon et al., 2024; Yim et al., 2025) but often fail to deliver competitive performance despite their conceptual appeal (Sec. 4).

We introduce *La-Proteina* (Latent Proteina), a method for atomistic protein design based on *partially latent flow matching*, combining the strengths of explicit and latent modeling. *La-Proteina* models the $\alpha$-carbon coordinates explicitly, while capturing sequence and coordinates of all non-$\alpha$-carbon atoms within a continuous, fixed-size latent representation per residue. We first train a Variational Autoencoder (VAE) (Kingma et al., 2013; Rezende et al., 2014), encoding sequence and side chain details in latent space, followed by a flow matching model (Lipman et al., 2023) that jointly generates $\alpha$-carbon coordinates and latent variables. New proteins are generated by sampling the flow model and decoding the $\alpha$-carbons and latent variables into sequences and fully atomistic structures (Fig. 1).

*La-Proteina*'s partially latent approach shifts the core learning problem from a mixed discrete–continuous space with variable dimensionality to a per-residue, continuous space of fixed dimensionality, making it amenable to powerful used generative modeling techniques such as flow matching.

---

*Equal contribution †equal advising.

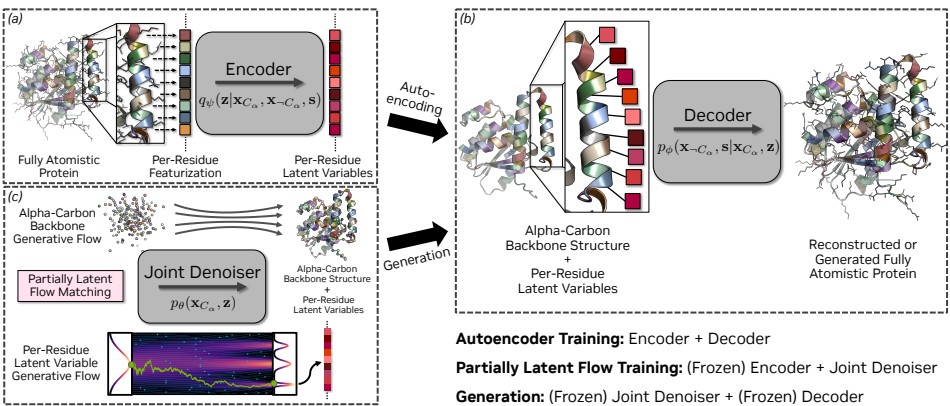

Figure 1: *La-Proteina* consists of encoder $q_\psi$ *(a)*, decoder $p_\phi$ *(b)*, and joint denoiser $p_\theta$ *(c)*. The encoder featurizes the input protein and predicts per-residue latent variables $\mathbf{z}$ of constant dimensionality. Together with the underlying $\alpha$-carbon backbone $\mathbf{x}_{C_\alpha}$, the decoder outputs sequence $\mathbf{s}$ and all other atoms $\mathbf{x}_{\neg C_\alpha}$ and reconstructs the atomistic protein. To facilitate generation of de novo proteins, a partially latent flow model jointly generates novel $\alpha$-carbon backbone structures $\mathbf{x}_{C_\alpha}$ and latents $\mathbf{z}$. The model is trained in two stages and all networks are implemented leveraging the same transformer architecture (Geffner et al., 2025); see details in Sec. 3.

Meanwhile, maintaining the explicit separation of the $\alpha$-carbon coordinates and the latent variables allows greater flexibility during generation. In particular, it enables the structural scaffold and the remaining atomic details to be generated using different generation schemes, i.e., different discretization schedules to simulate the underlying generative stochastic differential equation. *La-Proteina*'s neural networks are implemented using efficient transformer architectures (Vaswani et al., 2017; Geffner et al., 2025), guaranteeing the model's scalability to long proteins, many model parameters, and large training data—we train *La-Proteina* on up to 46 million protein structure-sequence pairs.

We empirically compare our model against leading publicly available methods for atomistic protein design and achieve state-of-the-art atomistic performance as measured by the all-atom co-designability and diversity metrics. *La-Proteina* can generate co-designable proteins of up to 800 residues, a regime where existing models collapse or run out of memory, demonstrating our method's strong scalability. We further assess the generated structures' geometric quality through analyses of side-chain conformations and validate overall structural integrity (Davis et al., 2007). *La-Proteina* significantly surpasses existing methods in these evaluations as well. Next, we apply *La-Proteina* to atomistic motif scaffolding, a critical task for protein engineering that most prior work has addressed only at the coarser backbone level (Watson et al., 2023; Yim et al., 2024; Lin et al., 2024; Geffner et al., 2025). We tackle both all-atom and tip-atom scaffolding, where in the latter case only functionally critical side chain tip atoms are given, rather than all atoms of the motif residues. Our model performs these tasks in two setups: the standard indexed task, where motif residue sequence indices are specified; and the more challenging unindexed task (Ahern et al., 2025), where these sequence indices are unknown. Our approach solves most benchmark tasks across all setups and outperforms baselines. We provide further insights through ablation studies and careful analysis of the model's latent space, which shows that *La-Proteina* encodes atomistic residue structure and amino acid type in a localized and consistent manner. In conclusion, *La-Proteina* represents a versatile, high-quality, fully atomistic protein structure generative model, with the potential to enable new, challenging protein design tasks.

**Main contributions.** (i) We propose *La-Proteina*, a partially latent flow matching framework designed for the joint generation of protein sequence and fully atomistic structure. (ii) *La-Proteina* achieves state-of-the-art performance in unconditional protein generation. (iii) We verify *La-Proteina*'s scalability, generating diverse, co-designable and structurally valid fully atomistic proteins of up to 800 residues. (iv) We successfully apply *La-Proteina* to indexed and unindexed atomistic motif scaffolding, two important conditional protein design tasks. (v) We provide further insights through ablations, latent space analyses, and biophysical assessments of *La-Proteina*'s generated atomistic protein structures, demonstrating our model's superiority over previous all-atom generators.

## 2 PRELIMINARIES

**VAEs** (Kingma et al., 2013; Rezende et al., 2014) learn a probabilistic representation of data $\mathbf{x}$ within a latent space employing two neural networks: an encoder mapping a sample $\mathbf{x}$ to a distribution

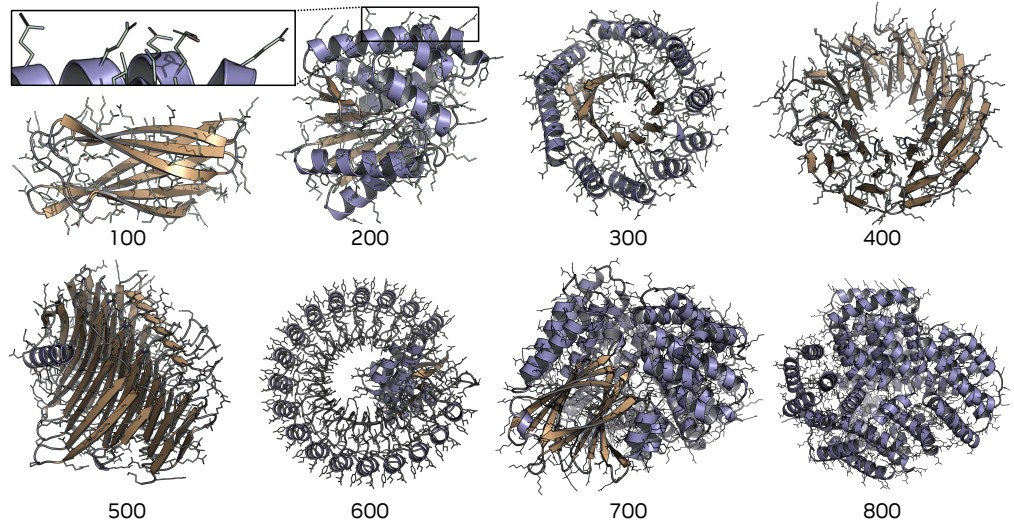

Figure 2: **Fully atomistic *La-Proteina* samples.** Numbers denote residue count. All samples co-designable.

$q_\psi(\mathbf{z} \,|\, \mathbf{x})$ over latent variables $\mathbf{z}$, and a decoder mapping $\mathbf{z}$ to a distribution in data space $p_\phi(\mathbf{x}|\mathbf{z})$. VAEs are trained by maximizing the Evidence Lower Bound, $\text{ELBO}(\phi, \psi) = \mathbb{E}_{\mathbf{x},\mathbf{z}}[\log p_\phi(\mathbf{x}|\mathbf{z})] - \text{KL}(q_\psi(\mathbf{z}|\mathbf{x}) \,\|\, p(\mathbf{z}))$. This objective balances reconstruction quality with a KL divergence-based regularization term that pushes the learned posterior $q_\phi(\mathbf{z}|\mathbf{x})$ towards an uninformative prior $p(\mathbf{z})$.

**Flow matching** (Lipman et al., 2023; Albergo & Vanden-Eijnden, 2023; Liu et al., 2023) trains a neural network $\mathbf{v}_\theta(\mathbf{x}^t, t)$ to model the velocity field $\mathbf{v}_t(\mathbf{x})$ that transports samples from a base distribution $p_0$ to the data distribution $p_1$ along a probability path $p_t$, for $t \in [0, 1]$. This path is often defined by linearly interpolating between samples $\mathbf{x}^0 \sim p_0$ and $\mathbf{x}^1 \sim p_1$ as $\mathbf{x}^t = (1-t)\mathbf{x}^0 + t\mathbf{x}^1$. The denoiser network $\mathbf{v}_\theta$ is trained by minimizing the conditional flow matching objective, $\mathbb{E}_{t, p_0, p_1}[\|\mathbf{v}_\theta(\mathbf{x}^t, t) - (\mathbf{x}^1 - \mathbf{x}^0)\|^2]$. Flow matching can be applied directly in data space or in latent spaces learned by models like VAEs (Rombach et al., 2022; Vahdat et al., 2021). Furthermore, when $p_0$ is Gaussian, flow matching is equivalent to diffusion models (Song et al., 2021; Gao et al., 2025), allowing us to compute the intermediate score functions $\nabla_{\mathbf{x}^t} \log p_t(\mathbf{x}^t)$ as a function of the trained network $\mathbf{v}_\theta(\mathbf{x}^t, t)$.

**Protein representation.** Protein data includes sequence (20 residue types) and 3D structure. Different residues share a common backbone, including the $\alpha$-carbon atom, but contain distinct atoms in their side chains. The Atom37 representation defines a standardized superset of 37 potential atoms per residue, which allows storing the structure of an $L$-residue protein as a tensor of shape $[L, 37, 3]$. The relevant subset of coordinates is selected based on each residue's type.

**Related work.** Early diffusion-based protein generators, such as RFDiffusion (Watson et al., 2023) and Chroma (Ingraham et al., 2023), focused on backbone generation. This area has since diversified, with some approaches leveraging diffusions on the SO(3) manifold (Yim et al., 2023b;a; Bose et al., 2024; Huguet et al., 2024), while others employ Euclidean Flow Matching (Lin & Alquraishi, 2023; Lin et al., 2024; Geffner et al., 2025). ProtComposer uses an auxiliary statistical model and 3D primitives (Stark et al., 2025). Several works (Lin et al., 2024; Qu et al., 2024; Geffner et al., 2025) obtained good performance training on synthetic structures from the AlphaFold database (AFDB) (Jumper et al., 2021; Varadi et al., 2021), which is significantly larger than the protein databank (PDB) (Berman et al., 2000). Recently, the task of sequence-structure co-design has gained prominence. Some methods address this by jointly modeling protein backbones and sequences (Campbell et al., 2024; Ren et al., 2024; Yim et al., 2025). Others tackle fully atomistic structures, including side chains, operating either in data space (Qu et al., 2024; Chu et al., 2024; Lisanza et al., 2023; Chen et al., 2025) or via latent variable models (McPartlon et al., 2024; Fu et al., 2024; Lu et al., 2024; Yim et al., 2025). Language models have also been used for protein design, with some methods focusing on protein sequences (Wang et al., 2024b); others tokenize structural information and model sequence and structure jointly (Hayes et al., 2024; Wang et al., 2024c). In this work we introduce *La-Proteina* for atomistic protein design and provide a thorough evaluation of its capabilities in unconditional monomer generation and motif scaffolding. Concurrent work by Anonymous (2025) extends *La-Proteina* to binder design. Their contributions are orthogonal to ours, focusing on modeling protein-protein interactions and inference-time search for that application.

# 3 LA-PROTEINA

## 3.1 MOTIVATION: PARTIALLY LATENT REPRESENTATION FOR ATOMISTIC PROTEIN DESIGN

While prior works have been able to successfully tackle high-quality protein backbone design, fully atomistic structure generation comes with additional challenges. The model needs to jointly reason over large-scale backbone structure, amino acid types, and side-chains, whose dimensionality depends on the amino acid—this represents a complex continuous-categorical generative modeling problem.

How can we best build on top of successful backbone generation frameworks (Watson et al., 2023; Geffner et al., 2025; Lin et al., 2024), while addressing the additional fully atomistic modeling challenges? We propose to encode per-residue atomistic detail and residue type in a fixed-length, continuous latent space, while maintaining explicit backbone modeling through the $\alpha$-carbon coordinates. This has several key advantages: (i) By encoding atomistic details, including *varying-length* side chains, together with their *categorical* residue type, into a *fixed-length*, *fully-continuous* latent space, we elegantly avoid mixed continuous-categorical modeling challenges in the model's main generative component. Together with the continuous backbone coordinates, the per-residue latent variables can be generated using efficient, fully-continuous flow matching methods, while mixed modality modeling complexities are handled by encoder and decoder. (ii) It is critical to maintain the explicit $\alpha$-carbon-based backbone representation in *La-Proteina*'s hybrid, *partially latent* framework. That way, we can build on top of advances in high-performance backbone modeling. Our ablations show that also encoding $\alpha$-carbons in latent space leads to significantly worse results (ablations in App. G.1.3).

(iii) Maintaining explicit backbone modeling capabilities also allows us to use different generation schedules for global $\alpha$-carbon backbone structure and per-residue atomistic (latent) details (see Sec. 3.4), a critical detail in our framework to achieve high performance (ablations in App. G.2). We argue that our hybrid approach is a key reason why *La-Proteina* significantly outperforms existing latent frameworks for protein structure generation, all of which opt for fully-latent modeling instead. (iv) Our partially latent framework increases scalability. Explicit modeling of all atoms in large proteins may require complex and memory-consuming neural networks—in fact, for that reason some approaches that treat all atoms explicitly can only be trained on small proteins (Qu et al., 2024). In contast, *La-Proteina*'s per-residue latent variables simply become additional channels on top of the $\alpha$-carbon coordinates, enabling the application of established, high-performance backbone-processing architectures (Geffner et al., 2025) without increasing

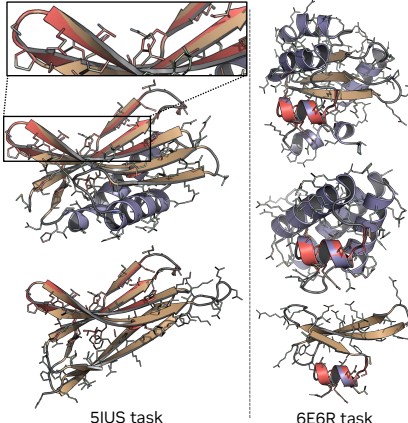

5IUS task          6E6R task

Figure 3: **Atomistic Motif Scaffolding.** *La-Proteina* accurately reconstructs the atomistic motif (red), while generating diverse scaffolds. Visualization overlays generated protein and motif.

the length of internal sequence representations. Hence, we can keep the model's memory footprint manageable, and scale the model to large protein generation tasks of up to 800 residues (see Sec. 4.1).

Next, we introduce *La-Proteina* (Fig. 1). First, we train a VAE, with its encoder mapping input proteins (sequence and structure) to latent variables, and its decoder reconstructing complete proteins from the latent variables and $\alpha$-carbon coordinates. Leveraging the VAE, we then train a flow matching model to learn the joint distribution over latent variables and coordinates of the $\alpha$-carbon atoms.

**Notation.** $L$ denotes protein length, $\mathbf{x}_{C_\alpha} \in \mathbb{R}^{L \times 3}$ the $\alpha$-carbon coordinates, $\mathbf{x}_{\neg C_\alpha} \in \mathbb{R}^{L \times 36 \times 3}$ the coordinates of other heavy atoms (Atom37 representation without $\alpha$-carbons, see Sec. 2), $\mathbf{s} \in \{0, ..., 19\}^L$ the protein sequence, and $\mathbf{z} \in \mathbb{R}^{L \times 8}$ the (8-dimensional) per-residue latent variables.

## 3.2 PROBABILISTIC FORMULATION

We learn a latent variable model $p(\mathbf{x}_{C_\alpha}, \mathbf{x}_{\neg C_\alpha}, \mathbf{s}, \mathbf{z})$, trained so that its marginal $\int p \, d\mathbf{z}$ approximates the target distribution over proteins $p_{\text{data}}(\mathbf{x}_{C_\alpha}, \mathbf{x}_{\neg C_\alpha}, \mathbf{s})$. Central to our approach is the factorization

$$p_{\theta,\phi}(\mathbf{x}_{C_\alpha}, \mathbf{x}_{\neg C_\alpha}, \mathbf{s}, \mathbf{z}) = p_\theta(\mathbf{x}_{C_\alpha}, \mathbf{z}) \, p_\phi(\mathbf{x}_{\neg C_\alpha}, \mathbf{s} \mid \mathbf{x}_{C_\alpha}, \mathbf{z}), \tag{1}$$

which enables the model to capture complex dependencies between backbone, sequence, and side chains through the latent variable $\mathbf{z}$. The first component of this factorization, $p_\theta(\mathbf{x}_{C_\alpha}, \mathbf{z})$, defined over a continuous, per-residue, fixed-dimensional space, is captured by our partially latent flow matching model (Sec. 3.2.2). The second component, $p_\phi(\mathbf{x}_{\neg C_\alpha}, \mathbf{s} \mid \mathbf{x}_{C_\alpha}, \mathbf{z})$, denotes the VAE's decoder, which, together the encoder, maps between latent variables $\mathbf{z}$ and proteins and handles complexities arising from mixed discrete/continuous data types (sequence and structure), and the variable dimensionality of side chains. Critically, by conditioning on both $\alpha$-carbon coordinates $\mathbf{x}_{C_\alpha}$ and expressive latent variables $\mathbf{z}$, this conditional distribution can be effectively represented by simple factorized distributions.

### 3.2.1 VARIATIONAL AUTOENCODER

The VAE's decoder models sequence and full-atom structure. Formally, it parameterizes the conditional likelihood term $p_\phi(\mathbf{x}_{\neg C_\alpha}, \mathbf{s} \mid \mathbf{x}_{C_\alpha}, \mathbf{z})$ from Eq. (1). We model this distribution assuming conditional independence between the sequence $\mathbf{s}$ and the coordinates of non-$\alpha$-carbon atoms $\mathbf{x}_{\neg C_\alpha}$

$$p_\phi(\mathbf{x}_{\neg C_\alpha}, \mathbf{s} \mid \mathbf{x}_{C_\alpha}, \mathbf{z}) = p_\phi(\mathbf{s} \mid \mathbf{x}_{C_\alpha}, \mathbf{z})\, p_\phi(\mathbf{x}_{\neg C_\alpha} \mid \mathbf{x}_{C_\alpha}, \mathbf{z}), \tag{2}$$

where we define $p_\phi(\mathbf{s} \mid \mathbf{x}_{C_\alpha}, \mathbf{z})$ as a factorized categorical distribution and $p_\phi(\mathbf{x}_{\neg C_\alpha} \mid \mathbf{x}_{C_\alpha}, \mathbf{z})$ as a factorized Gaussian with unit variance. These choices are standard in the VAE literature (Kingma et al., 2013), justified by expressive conditioning on the latent variables and $\alpha$-carbon coordinates, which capture underlying dependencies and enable accurate approximations using simple factorized forms.

The decoder network takes the latent variables $\mathbf{z}$ and $\alpha$-carbon coordinates $\mathbf{x}_{C_\alpha}$ as input, producing parameters for the distributions over sequence $\mathbf{s}$ and non-$\alpha$-carbon atom coordinates $\mathbf{x}_{\neg C_\alpha}$. To handle the varying non-$\alpha$-carbon atom count across residue types while maintaining a fixed output dimensionality, the decoder generates Atom37 coordinates for each residue structure, yielding a $[L, 37, 3]$ tensor. The appropriate subset of all Atom37 entries is selected on the basis of the sequence, using the ground truth sequence during training (supervising only the selected entries) and the decoded sequence during inference. Further, the coordinates of the $\alpha$-carbons are set to the ones passed as input.

The VAE encoder, on the other hand, is used to map proteins to their corresponding latent representation. Formally, the encoder parameterizes $q_\psi(\mathbf{z} \mid \mathbf{x}_{C_\alpha}, \mathbf{x}_{\neg C_\alpha}, \mathbf{s})$, a factorized Gaussian designed to approximate the posterior distribution $p_{\phi,\theta}(\mathbf{z} \mid \mathbf{x}_{C_\alpha}, \mathbf{x}_{\neg C_\alpha}, \mathbf{s})$. This network takes the complete protein structure $(\mathbf{x}_{C_\alpha}, \mathbf{x}_{\neg C_\alpha}, \mathbf{s})$ as input, and outputs the mean and log-scale parameters for $q_\psi(\mathbf{z} \mid \cdot)$.

The encoder and decoder are optimized maximizing the $\beta$-weighted ELBO (Higgins et al., 2017), a common objective for VAE training in the context of generative modeling in latent spaces, given by

$$\max_{\phi,\psi} \mathbb{E}_{p_{\mathrm{data}}(\mathbf{x}_{C_\alpha}, \mathbf{x}_{\neg C_\alpha}, \mathbf{s}), q_\psi(\mathbf{z} \mid \ldots)} \left[ \log p_\phi(\mathbf{x}_{\neg C_\alpha}, \mathbf{s} \mid \mathbf{x}_{C_\alpha}, \mathbf{z}) \right] - \beta\, \mathrm{KL}\left( q_\psi(\mathbf{z} \mid \mathbf{x}_{C_\alpha}, \mathbf{x}_{\neg C_\alpha}, \mathbf{s}) \| p(\mathbf{z}) \right). \tag{3}$$

For the modeling choices described above, the reconstruction term in Eq. (3) reduces to the cross entropy loss for the sequence and the squared $L_2$ loss for the structure. For training, we set $\beta = 10^{-4}$ and use a standard isotropic Gaussian prior over latent variables $p(\mathbf{z}) = \mathcal{N}(\mathbf{z} \mid 0, I)$.

### 3.2.2 PARTIALLY LATENT FLOW MATCHING

The second stage of training *La-Proteina* involves optimizing a flow matching model to approximate the target distribution $p_{\mathrm{data},\psi}(\mathbf{x}_{C_\alpha}, \mathbf{z})$.[1] This model trains a denoiser network $\mathbf{v}_\theta(\mathbf{x}_{C_\alpha}^{t_x}, \mathbf{z}^{t_z}, t_x, t_z)$ to predict the velocity field transporting samples from a standard Gaussian reference distribution, $p_0(\mathbf{x}_{C_\alpha}^0, \mathbf{z}^0)$, to the target data distribution, $p_1(\mathbf{x}_{C_\alpha}^1, \mathbf{z}^1)$, for $t_x, t_z \in [0, 1]$. These are defined as

$$p_0\left(\mathbf{x}_{C_\alpha}^0, \mathbf{z}^0\right) = \mathcal{N}\left(\mathbf{x}_{C_\alpha}^0 \mid \mathbf{0}, \boldsymbol{I}\right) \mathcal{N}\left(\mathbf{z}^0 \mid \mathbf{0}, \boldsymbol{I}\right) \quad \text{and} \quad p_1(\mathbf{x}_{C_\alpha}^1, \mathbf{z}^1) \approx p_{\mathrm{data}}(\mathbf{x}_{C_\alpha}, \mathbf{z}). \tag{4}$$

The denoiser network $\mathbf{v}_\theta$ is trained by minimizing the conditional flow matching (CFM) objective

$$\min_\theta \mathbb{E}\left[ \left\| \mathbf{v}_\theta^x\left(\mathbf{x}_{C_\alpha}^{t_x}, \mathbf{z}^{t_z}, t_x, t_z\right) - \left(\mathbf{x}_{C_\alpha} - \mathbf{x}_{C_\alpha}^0\right) \right\|^2 + \left\| \mathbf{v}_\theta^z\left(\mathbf{x}_{C_\alpha}^{t_x}, \mathbf{z}^{t_z}, t_x, t_z\right) - \left(\mathbf{z} - \mathbf{z}^0\right) \right\|^2 \right], \tag{5}$$

where the expectation is over $p_{\mathrm{data},\psi}(\mathbf{x}_{C_\alpha}, \mathbf{z})$ (i.e., $p_1$), noise distributions $\mathcal{N}(\mathbf{x}_{C_\alpha}^0 \mid \mathbf{0}, \boldsymbol{I})$ and $\mathcal{N}(\mathbf{z}^0 \mid \mathbf{0}, \boldsymbol{I})$ (i.e., $p_0$), and interpolation time distributions $p_{t_x}(t_x)$ and $p_{t_z}(t_z)$. The use of two

---

[1] For this stage the VAE parameters are frozen. The target distribution over $\alpha$-carbon coordinates and latent variables is defined by the data distribution $p_{\mathrm{data}}(\mathbf{x}_{C_\alpha}, \mathbf{x}_{\neg C_\alpha}, \mathbf{s})$ and the VAE encoder $q_\psi(\mathbf{z} \mid \mathbf{x}_{C_\alpha}, \mathbf{x}_{\neg C_\alpha}, \mathbf{s})$, and can be sampled by $(\mathbf{x}_{C_\alpha}, \mathbf{x}_{\neg C_\alpha}, \mathbf{s}) \sim p_{\mathrm{data}}(\mathbf{x}_{C_\alpha}, \mathbf{x}_{\neg C_\alpha}, \mathbf{s})$ and $\mathbf{z} \sim q_\psi(\mathbf{z} \mid \mathbf{x}_{C_\alpha}, \mathbf{x}_{\neg C_\alpha}, \mathbf{s})$.

separate interpolation times $t_x$ and $t_z$ is a critical design decision that enables the use of different integration schedules for the coordinates of the $\alpha$-carbons $\mathbf{x}_{C_\alpha}$ and latent variables $\mathbf{z}$ during inference. This flexibility is vital for achieving strong performance; employing a single, coupled time $t$ would enforce an identical schedule for both modalities, which leads to worse results (see App. G.2).

The specific form for the time sampling distributions represent an important design decision (Esser et al., 2024). Inspired by Geffner et al. (2025), who, in the context of backbone design, employ a mixture of Uniform and Beta distributions, we adopt independent sampling for $t_x$ and $t_z$ given by

$$p_{t_x} = 0.02\,\mathrm{Unif}(0,1) + 0.98\,\mathrm{Beta}(1.9,1) \quad \text{and} \quad p_{t_z} = 0.02\,\mathrm{Unif}(0,1) + 0.98\,\mathrm{Beta}(1,1.5), \quad (6)$$

visualized in Fig. 13. The distribution parameters were chosen based on our observation that generating backbones using a faster schedule than that used for the latent variables yields superior results during inference (see Sec. 3.4). Hence, the distributions from Eq. (6) were chosen so that time pairs that satisfy $t_x > t_z$, relevant for the used inference schedules, are sampled more frequently.

### 3.3 NEURAL NETWORK ARCHITECTURES

The neural networks used by *La-Proteina*, the encoder, decoder, and denoiser, rely on a shared core architecture based on transformers with pair-biased attention mechanisms (Jumper et al., 2021), with our implementation closely following Geffner et al. (2025). While built on this common foundation, the three networks are distinguished by their specific inputs, outputs, and conditioning mechanisms. For instance, the encoder takes as input full proteins and outputs per-residue latent variables, while the decoder produces full proteins given latent variables and $\alpha$-carbon coordinates. Additionally, the denoiser is conditioned on interpolation times $(t_x, t_z)$ using adaptive layer normalization and output scaling techniques (Peebles & Xie, 2023). The encoder and decoder each consist of approximately 130M parameters, while the denoiser totals 160M. A key architectural choice for scalability is that our main models omit computationally expensive triangular update layers (Jumper et al., 2021), which, while effective in structural biology tasks (Lin et al., 2024; Abramson et al., 2024), incur significant memory and compute costs. Following Geffner et al. (2025), *La-Proteina* achieves high performance using only efficient transformer networks, maintaining strong scalability. These triangular multiplicative layers can optionally be added to improve pair representations and enhance protein co-designability (see Sec. 4.1). We include architectural details in App. H.

### 3.4 MODEL SAMPLING

New proteins can be generated by *La-Proteina* by sampling latent variables and $\alpha$-carbon coordinates using the partially latent flow matching model, and then feeding these through the decoder (Fig. 1).

**Sampling the partially latent flow matching model.** As we use Gaussian flows (Sec. 2) we can estimate the score of intermediate densities $\zeta(\mathbf{x}_{C_\alpha}^{t_x}, \mathbf{z}^{t_z}, t_x, t_z) \approx \nabla \log p_\theta^{t_x, t_z}(\mathbf{x}_{C_\alpha}^{t_x}, \mathbf{z}^{t_z})$ directly from $\mathbf{v}_\theta$ (Albergo et al., 2023) (see App. E). Access to scores enables the use of stochastic samplers to generate pairs of $\alpha$-carbon coordinates and latent variables $(\mathbf{x}_{C_\alpha}, \mathbf{z})$. We generate such samples by simulating the following stochastic differential equations (SDEs) from $(t_x, t_z) = (0, 0)$ to $(t_x, t_z) = (1, 1)$:

$$\begin{aligned}
\mathrm{d}\mathbf{x}_{C_\alpha}^{t_x} &= \mathbf{v}_\theta^x(\mathbf{x}_{C_\alpha}^{t_x}, \mathbf{z}^{t_z}, t_x, t_z)\mathrm{d}t_x + \beta_x(t_x)\zeta^x(\mathbf{x}_{C_\alpha}^{t_x}, \mathbf{z}^{t_z}, t_x, t_z)\mathrm{d}t_x + \sqrt{2\beta_x(t_x)\eta_x}\,\mathrm{d}\mathcal{W}_{t_x} \\
\mathrm{d}\mathbf{z}^{t_z} &= \mathbf{v}_\theta^z(\mathbf{x}_{C_\alpha}^{t_x}, \mathbf{z}^{t_z}, t_x, t_z)\mathrm{d}t_z + \beta_z(t_z)\zeta^z(\mathbf{x}_{C_\alpha}^{t_x}, \mathbf{z}^{t_z}, t_x, t_z)\mathrm{d}t_z + \sqrt{2\beta_z(t_z)\eta_z}\,\mathrm{d}\mathcal{W}_{t_z}.
\end{aligned} \quad (7)$$

Here, $\beta_x$ and $\beta_z$ are scaling functions that modulate the contribution of the Langevin-like term in the SDEs (Karras et al., 2022) (details in App. E). We also use noise scaling parameters $\eta_x$ and $\eta_z$, set to values less than or equal to one, to control the magnitude of the injected noise. This follows common practices in protein design; virtually all successful flow matching and diffusion-based methods adopt some form of reduced noise or temperature sampling, as it has been consistently observed to improve (co-)designability, albeit at the cost of reduced diversity (Yim et al., 2023a; Watson et al., 2023; Lin et al., 2024; Bose et al., 2024; Wang et al., 2024a; Campbell et al., 2024; Geffner et al., 2025).

We use the Euler-Maruyama method (Higham, 2001) to simulate Eq. (7). As discussed, independently scheduling the generation of $\alpha$-carbon coordinates $\mathbf{x}_{C_\alpha}$ and latent variables $\mathbf{z}$ is critical for good performance. Our empirical findings indicate that discretization strategies that generate $\mathbf{x}_{C_\alpha}$ at a faster rate than $\mathbf{z}$ yield improved results over alternative choices. Full details of our sampling algorithms, including ablations for these discretization schemes, are provided in Apps. E and G.

Table 1: ***La-Proteina* achieves state-of-the-art results on unconditional all-atom design, for lengths between 100 and 500 residues.** Diversity, novelty, and secondary structure computed on all-atom co-designable samples. The $_\text{tri}$ suffix indicates *La-Proteina* with multiplicative triangular update layers to update the pair representation. $\eta_x$ and $\eta_z$ denote the noise scaling factors during generation (Eq. (7)). Best scores **bold**, second best underlined.

| Method | Co-designability (%) ↑ | | pLDDT ↑ | Diversity (# clusters) ↑ | | | Novelty ↓ | | Designability (%) ↑ | | Sec. Str. (%) | |
|---|---|---|---|---|---|---|---|---|---|---|---|---|
| | All-atom | $\alpha$-carbon | ($\% \geq 80$) | Str | Seq | Seq+Str | PDB | AFDB | MPNN-8 | MPNN-1 | $\alpha$ | $\beta$ |
| P(all-atom) | 36.7 | 37.9 | 92 | 134 | 148 | 165 | **0.72** | **0.81** | 57.9 | 44.4 | 56 | 17 |
| Protpardelle-1c | 35.8 | 44.8 | 62 | 41 | 138 | 61 | 0.78 | 0.83 | 62.0 | 52.6 | 63 | 14 |
| APM | 19.0 | 32.2 | 85 | 32 | 64 | 59 | 0.84 | 0.89 | 61.8 | 42.8 | 73 | 8 |
| PLAID | 11.0 | 19.2 | 44 | 25 | 38 | 27 | 0.89 | 0.92 | 37.6 | 23.8 | 44 | 14 |
| ProteinGenerator | 9.8 | 17.8 | 52 | 12 | 28 | 24 | 0.83 | 0.89 | 54.2 | 42.8 | 78 | 5 |
| Protpardelle | 8.8 | 35.2 | 37 | 10 | 37 | 21 | 0.79 | 0.82 | 56.2 | 43.8 | 65 | 14 |
| *La-Proteina* $(\eta_x, \eta_z) = (0.1, 0.1)$ | 68.4 | 72.2 | 97 | **206** | **216** | **301** | 0.75 | 0.82 | 93.8 | 82.6 | 72 | 5 |
| *La-Proteina* $(\eta_x, \eta_z) = (0.2, 0.1)$ | 60.6 | 64.2 | 97 | 198 | 197 | 261 | 0.76 | 0.83 | **95.4** | 80.2 | 66 | 8 |
| *La-Proteina* $(\eta_x, \eta_z) = (0.3, 0.1)$ | 53.8 | 59.6 | 95 | 180 | 189 | 249 | 0.77 | 0.86 | 94.6 | 76.0 | 63 | 10 |
| *La-Proteina* $_\text{tri}$ $(\eta_x, \eta_z) = (0.1, 0.1)$ | **75.0** | **78.2** | **100** | 129 | 199 | 247 | 0.82 | 0.86 | 94.6 | **84.6** | 73 | 6 |
| *La-Proteina* $_\text{tri}$ $(\eta_x, \eta_z) = (0.3, 0.1)$ | 71.6 | 75.8 | 99 | 166 | 211 | 294 | 0.79 | 0.85 | 95.2 | 83.4 | 66 | 9 |

**Sampling the VAE decoder.** The $\alpha$-carbon coordinates $\mathbf{x}_{C_\alpha}$ and latent variables $\mathbf{z}$ produced by the flow matching model are passed to the VAE decoder. The non-$\alpha$-carbon coordinates $\mathbf{x}_{\neg C_\alpha}$ are then obtained by taking the mean of the Gaussian distribution $p_\phi(\mathbf{x}_{\neg C_\alpha} \mid \mathbf{x}_{C_\alpha}, \mathbf{z})$, while the amino acid sequence $\mathbf{s}$ is determined by taking the $\arg\max$ of the logits of the categorical $p_\phi(\mathbf{s} \mid \mathbf{x}_{C_\alpha}, \mathbf{z})$.

# 4 EXPERIMENTS

We evaluate *La-Proteina* on unconditional atomistic protein generation up to 800 residues as well as on atomistic motif scaffolding, a critical protein design task. We train all models on a filtered version of the Foldseek cluster representatives of the AFDB (van Kempen et al., 2024), except for long protein generation where we train on a custom subset of the AFDB consisting of ~46M samples. Unless otherwise specified, our trained *La-Proteina* models omit triangular update layers; any use of such layers is explicitly noted (used for a single model in Sec. 4.1). Full experimental details, including datasets, metrics, and training procedures, as well as ablations, in Apps. C, D and G.

## 4.1 ALL-ATOMISTIC UNCONDITIONAL PROTEIN STRUCTURE GENERATION BENCHMARK

Tab. 1 compares two variants of *La-Proteina*, one with triangular multiplicative layers and one without, against publicly available all-atom generation baselines, including P(all-atom) (Qu et al., 2024), APM (Chen et al., 2025), PLAID (Lu et al., 2024), ProteinGenerator (Lisanza et al., 2023), Protpardelle (Chu et al., 2024) and Protpardelle-1c (Lu et al., 2025).[2] Each method was used to generate 100 proteins for each length in $\{100, 200, 300, 400, 500\}$. We assess performance using several metrics (described in App. D), including all-atom co-designability, pLDDT, diversity, novelty (against PDB and AFDB), and standard designability, the last being a metric typically used to evaluate backbone design methods. Co-designability evaluates how well co-generated sequences fold into generated structures, while designability uses ProteinMPNN (Dauparas et al., 2022) to produce sequences for generated structures. We note that our co-designability filter does not use a predicted local distance difference (pLDDT) cutoff; we instead report pLDDT values of successfully refolded samples separately.

Results in Tab. 1 show that both variants of *La-Proteina* outperform all baselines in all-atom co-designability, designability, and diversity, while remaining highly competitive in novelty. Additionally, we observe that *La-Proteina* with triangular layers tends to achieve higher co-designability values, albeit at the cost of diversity, and that all *La-Proteina* models yield higher pLDDT values for successfully refolded samples than baselines (a more detailed pLDDT analysis is provided in Fig. 34). Crucially, *La-Proteina* without triangular multiplicative layers establishes state-of-the-art performance while being highly scalable. This contrasts sharply with the second-best performing method, P(all-atom), which relies on computationally expensive triangular update layers (Jumper et al., 2021), thereby limiting it to short proteins. Due to its favorable scalability and performance, all remaining experiments in the upcoming sections rely on *La-Proteina* without triangular update layers.

**Generation of Large All-Atomistic Structures.** To demonstrate scalability, we trained another version of *La-Proteina* on an AFDB dataset with ~46M samples with length up to 896 residues

---

[2]Protpardelle-1c was trained conditionally for atomistic scaffolding. However, Lu et al. (2025) did not train an all-atom unconditional model. We sampled the conditional model unconditionally for this evaluation.

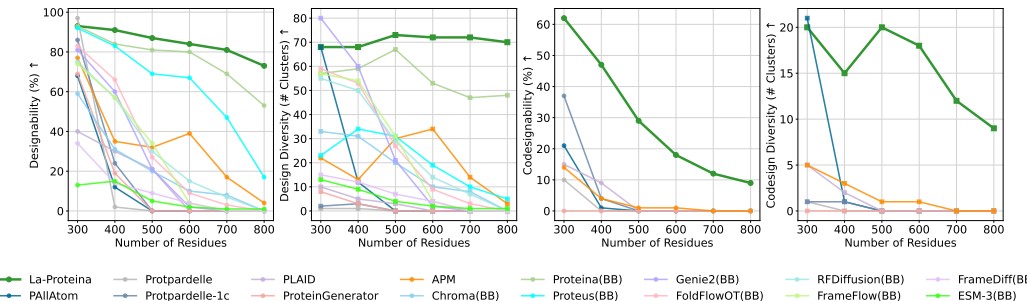

Figure 4: *La-Proteina*'s **strong performance for unconditional long length generation.** *La-Proteina* produces co-designable and diverse proteins of over 500 residues, where all all-atom baselines collapse, yielding no co-designable samples. Left plots show backbone metrics (designability, diversity) against backbone and all-atom baselines; right plots show all-atom metrics (all-atom codesignability, diversity). Metrics detailed in App. D.

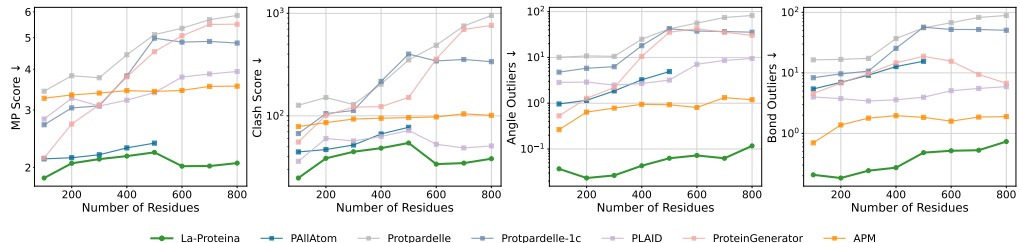

Figure 5: *La-Proteina* **produces structures with higher structural validity than existing all-atom generation baselines.** MolProbity metrics assessing structural quality: overall *MP score*, *clash score*, Ramachandran *angle outliers*, and covalent *bond outliers* (details in App. D). P(all-atom) limited to 500 residues; generating longer proteins is computationally prohibitive, requiring over 140GB of GPU memory to produce a single sample.

(details in App. C.1). We see in Fig. 4 that *La-Proteina* performs best in terms of (co-)designability and diversity for the task of backbone design (left two panels) as well as all-atom design (right two panels). Notably, *La-Proteina* outperforms the previous state-of-the-art Proteina method (Geffner et al., 2025) in backbone design tasks at all lengths, and is far ahead in co-designability compared to other all-atom generation methods, which fail to produce realistic samples of length 500 and above.

**Biophysical Analysis of All-Atom Structure Validity.**
To examine the biophysical quality of generated structures, we evaluate our model and all-atom baselines using two approaches (details in App. D): First, we use the MolProbity tool (Davis et al., 2007) to assess the structural validity in terms of bond angles, clashes and other physical quantities. Fig. 5 shows that *La-Proteina* produces more high quality structures, scoring significantly better than all baselines. The structures generated by *La-Proteina* are the most physically realistic ones, similar to real proteins.

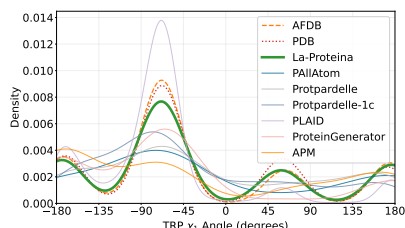

Figure 6: Distribution of TRP $\chi_1$ angle.

Most side chain torsion angles do not vary freely, but cluster due to steric repulsions into so-called rotamers (Haddad et al., 2019). As a second validation to judge the coverage of conformational space, we visualize side-chain dihedral angle distributions and compare their rotamer populations to PDB and AFDB references, similar to how rotamer libraries operate (Dunbrack Jr, 2002). *La-Proteina* models these distribution accurately, as shown in Fig. 6 for the tryptophan $\chi_1$ angle. *La-Proteina*'s samples accurately recover all major rotameric states as well as their respective frequencies with respect to the reference PDB/AFDB. In contrast, baselines often deviate from these references, missing modes or populating unrealistic angular regions. Plots for all residues and angles in App. D.3.2.

## 4.2 ATOMISTIC MOTIF SCAFFOLDING

Two advantages of all-atom generative models are their ability to incorporate atomistic conditioning information as well as designing new protein structures independent of backbone or rotamer constraints. To this end, we trained *La-Proteina* on the challenging task of atomistic motif scaffolding, where given the atomic structure of a predefined motif the model should generate a protein structure that scaf-

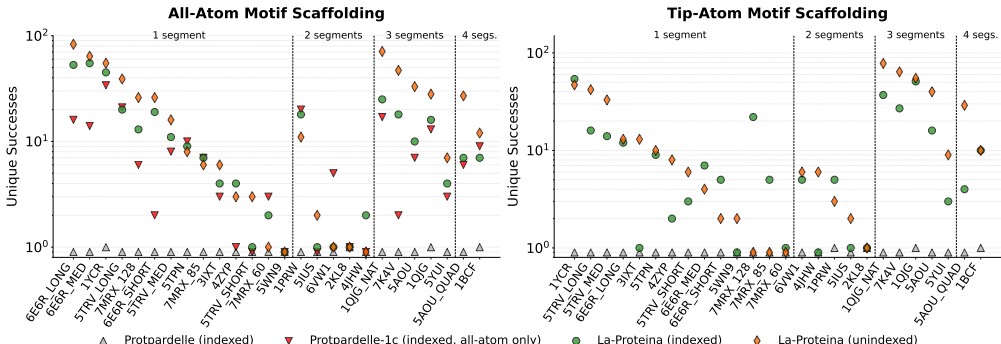

Figure 7: **Atomistic motif scaffolding.** 26 atomistic motif-scaffilding tasks (x-axis), comparing Protpardelle (limited to indexed), Protpardelle-1c (limited to all-atom indexed), *La-Proteina* (indexed) and *La-Proteina* (unindexed). *La-Proteina* solves between 21 and 25 of the 26 tasks, depending on the task type (all-atom or tip-atom, indexed or unindexed), vastly outperforms Protpardelle (which solves 4 out of 26 tasks), and also outperforms Protpardelle-1c, achieving a larger number of unique successes in 21 out of the 26 tasks. "# segments" refers to the number of residue segments in the motif. Detailed evaluation criteria in App. F.

folds this motif accurately. We assessed performance under two distinct levels of input motif detail: *all-atom*, where the model is conditioned on the complete atomic structure of the motif residues (backbone and side-chain), and *tip-atom* scaffolding, where we only prespecify important functional groups after the final rotatable bond and let the model decide the relative backbone and rotamer placement. For each of these two tasks we test both an *indexed* version, where the sequence indices of the motif residues are provided, and an *unindexed* version, where the model must also infer these positions, resulting in four evaluation setups. Across all setups a design is successful if it is all-atom co-designable, has an $\alpha$-carbon motifRMSD <1Å, and an all-atom motifRMSD <2Å. Complete details in App. F.

Fig. 7 presents the results for 26 atomistic scaffolding tasks, grouped by the number of continuous residue segments forming the motif. Our model consistently outperforms both Protpardelle-1c, which is limited to indexed all-atom scaffolding, and its predecessor Protpardelle, which is restricted to indexed scaffolding. *La-Proteina* successfully solves most benchmark tasks across all four regimes: *all-atom* and *tip-atom*, for both the *indexed* and *unindexed* setups. Interestingly, for motifs comprised of three or more distinct residue segments, the unindexed version of *La-Proteina* consistently outperforms its indexed counterpart. We hypothesize this is because fixing the positions of multiple segments limits the model's flexibility to explore diverse structural solutions; the freedom to determine the placement of the motif's residues in the unindexed setup is crucial for discovering a wider range of scaffolds. A similar effect was observed by concurrent work (Faltings et al., 2025; Ahern et al., 2025). Example scaffolds illustrating *La-Proteina*'s diverse and successful designs are shown in Fig. 3, with additional examples for tip-atom motif scaffolding of relevant enzyme active sites in Figs. 10 to 12. The results presented here evaluate *La-Proteina* structures directly; an alternative, more stringent evaluation using refolded structures (obtained by folding the model-produced sequences) is provided in Fig. 33, where it can be observed that *La-Proteina* achieves state-of-the-art performance under this evaluation as well.

## 4.3 AUTOENCODER EVALUATION AND LATENT SPACE ANALYSIS

We assessed the VAE's reconstruction performance on a held-out test set, where it achieved a mean all-atom RMSD $\approx 0.12$Å and a perfect sequence recovery rate of 1. Beyond reconstruction, we analyzed the properties of the latent space. t-SNE visualization of the latent variables (Fig. 8, left) reveals distinct clusters corresponding to different amino acid residue types, indicating that latent variables effectively capture residue-specific features. In addition, we

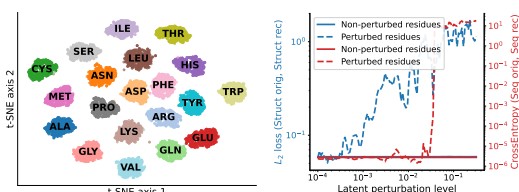

Figure 8: **Analyzing *La-Proteina*'s latent space.** t-SNE plot (left) and perturbation-based locality analysis (right).

see that structurally (GLN/GLU, ASN/ASP) as well as chemically similar amino acids (aromatics like PHE/TYR/TRP) cluster together, indicating the latent space captures biophysically relevant features.

To further probe the learned representation, we conducted a simple perturbation experiment: after encoding a protein, the latent variables associated with a single residue were perturbed with varying mag-

nitudes. We observe that such localized perturbations to a single residue's latent vector predominantly impact the reconstruction of that specific residue, leaving other residues almost unaffected (Fig. 8, right; red: sequence reconstruction loss, blue: structure reconstruction loss). This "local behavior" of the latent representation is noteworthy: Although both the encoder and decoder use transformer architectures capable of modeling long-range dependencies and jointly process the entire protein and all latent variables, our analysis suggests that each per-residue latent variable primarily encapsulates information pertinent to its own corresponding residue, rather than distributing information non-locally.

## 5 CONCLUSIONS

We presented *La-Proteina*, a scalable and efficient all-atom protein structure generative model that achieves state-of-the-art performance in unconditional and conditional atomistic protein design tasks and can generate realistic atomistic structures of up to 800 residues. Our key design choice involves a partially latent flow matching model that inherits the performance benefits of backbone generative models while benefiting from a per-residue fixed-size latent representation for sequence and side-chains, side-stepping scalability and accuracy issues that other methods suffer from. We believe that *La-Proteina* and its strong performance on atomistic design tasks, like unindexed atomistic motif scaffolding, could enable new important protein design applications, like binder and enzyme design.

## 6 ETHICS STATEMENT

The advancement of generative models for de novo protein design, including approaches like *La-Proteina*, offers the potential for significant positive impact across many scientific and societal domains. These technologies can significantly accelerate the discovery and development of protein-based therapeutics. In biotechnology and industry, computationally designed enzymes could pave the way, for instance, to greener chemical processes and sustainable materials. Alongside these benefits, new and improved tools for de novo protein design carry certain risks. While models in this area are developed for beneficial applications, any technology capable of designing novel functional proteins could, in principle, be misused if ethical oversight is not in place. This includes, for instance, the hypothetical design of proteins that could pose biosecurity threats.

## 7 REPRODUCIBILITY STATEMENT

To ensure reproducibility, we release code for *La-Proteina* at `https://github.com/NVIDIA-Digital-Bio/la-proteina`. We also provide comprehensive experimental details in the Supplementary Material. This includes includes the precise filters used for dataset construction (App. C.1), model architectures (App. H), and training parameters like GPU count and training steps (App. C.2 for unconditional generation, App. F.1 for atomistic motif scaffolding). We also include precise descriptions and the exact commands used to compute all reported metrics (App. D for metrics used to evaluate unconditional models, App. F.3 for atomistic motif scaffolding).

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

# Appendix

## A    LIMITATIONS AND FUTURE WORK

This work focuses on the de novo design of monomeric proteins using *La-Proteina*. While this scope allows for a thorough investigation and demonstration of our partially latent approach for single-chain structures, we did not apply *La-Proteina* to the area of protein complex design. In biological systems, proteins typically function as components of larger assemblies. Handling protein complexes is critical for important tasks such as de novo binder design and enzyme design, which inherently require the modeling of full protein complexes and their interfaces. The instantiation of *La-Proteina* presented in this work was not trained to handle protein complexes. Our focus on monomers should be viewed as a limitation of the current application scope rather than a constraint of the underlying *La-Proteina* framework. We anticipate that the principles of combining explicit structural modeling with latent representations could be fruitfully extended in future work to address the challenges of designing functional protein complexes.

## B    ADDITIONAL VISUALIZATIONS

### B.1    UNCONDITIONAL *La-Proteina* SAMPLES

In Fig. 9, we show additional unconditional *La-Proteina* samples. Our model can generate diverse and co-designable fully atomistic proteins across a broad range of sizes (residue count).

### B.2    ATOMISTIC MOTIF SCAFFOLDING *La-Proteina* SAMPLES

In Figs. 10 to 12, we show additional atomistic motif scaffolding visualizations. All three figures show partial side chain scaffolding setups, where only the tips of the conditioning side chains are given. The examples correspond to the scaffolding of enzyme active sites. We observe that the red conditioning motifs are exactly reproduced in almost all cases, and overall valid proteins are generated. Moreover, Fig. 12 demonstrates how *La-Proteina* can scaffold the same atomistic motif in diverse ways.

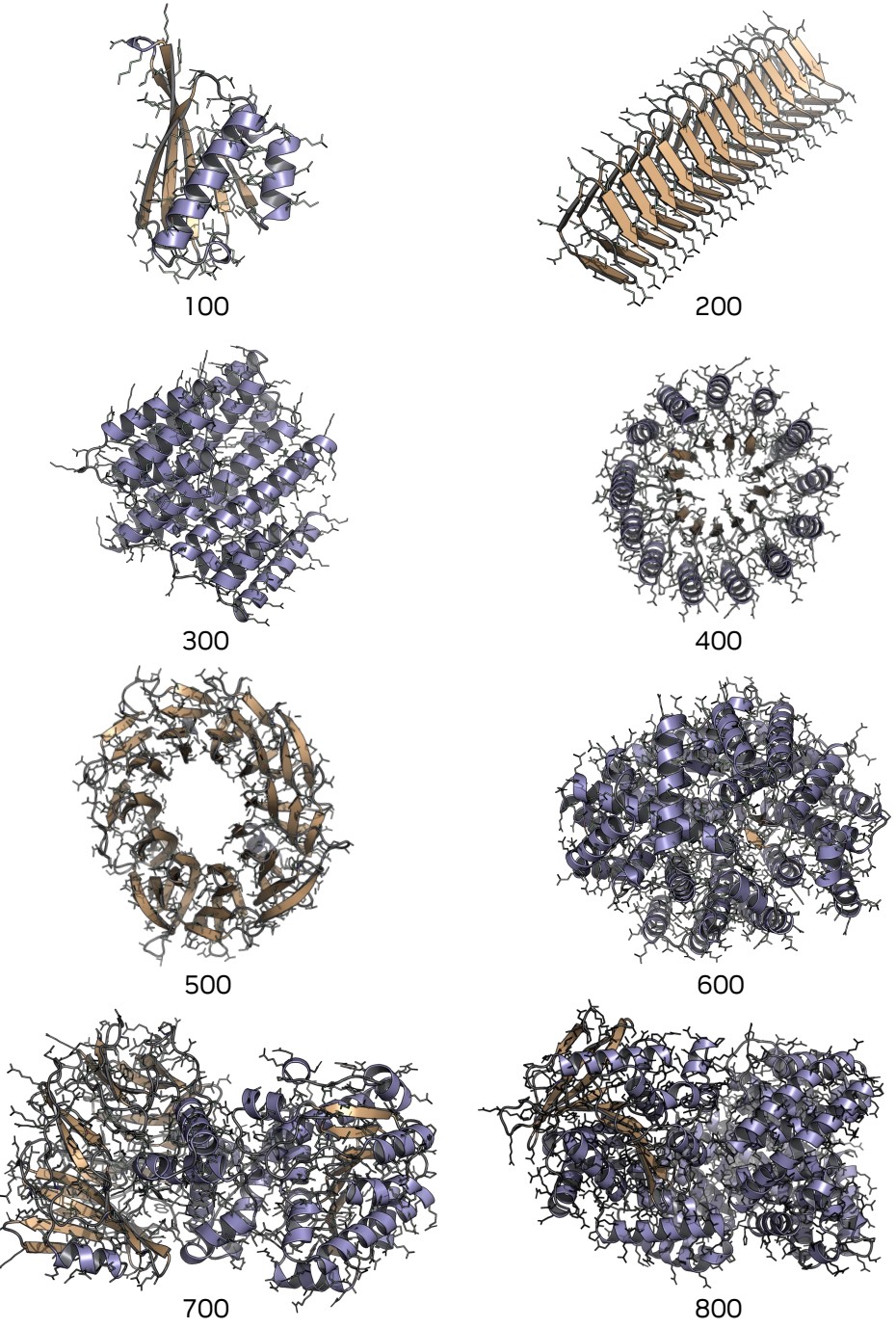

Figure 9: **Fully atomistic unconditional *La-Proteina* samples**. Numbers denote residue count. All samples co-designable.

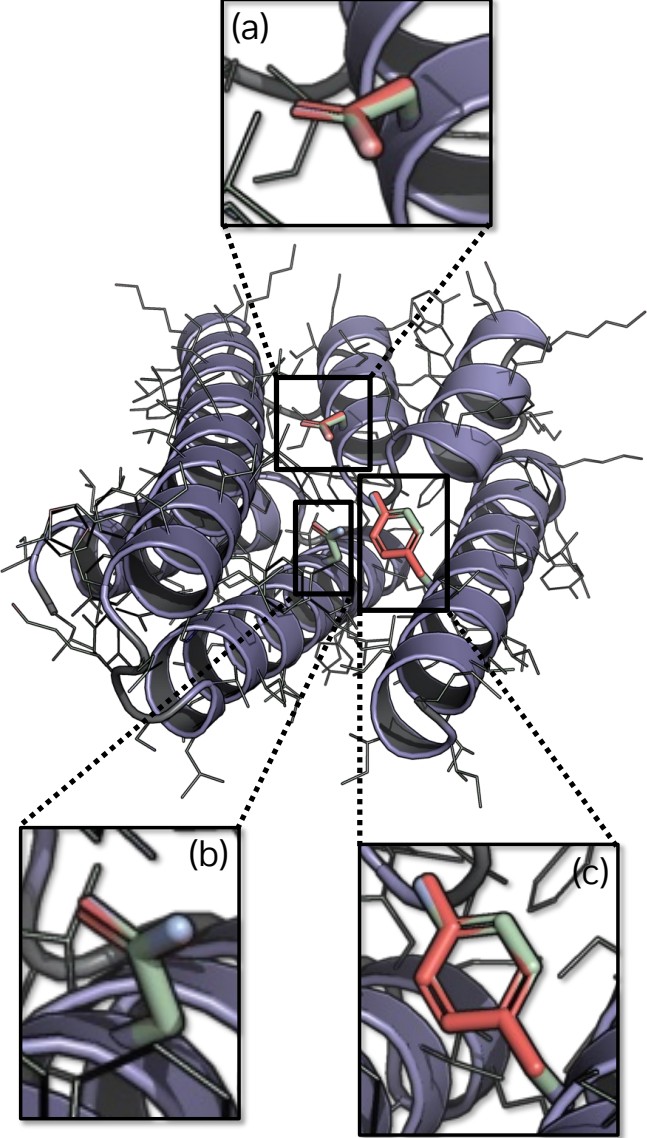

Figure 10: **Atomistic Motif Scaffolding**. Task 1QJG (Delta(5)-3-Ketosteroid isomerase). The active site consists of an ASP that acts as a general base, a TYR that stabilises the oxyanion in the transition state and another ASP that also stabilises the transition state by forming a hydrogen bond with the oxyanion. *La-Proteina* successfully generates a valid atomistic scaffold and accurately reproduces the red conditioning atoms that form the tip of partially given side chains (see zoom-ins (a)-(c)). Side chains that involve conditioning atoms are visualized as thick sticks, all other side chains are shown as thin sticks. Visualization overlays generated protein and atomistic motif.

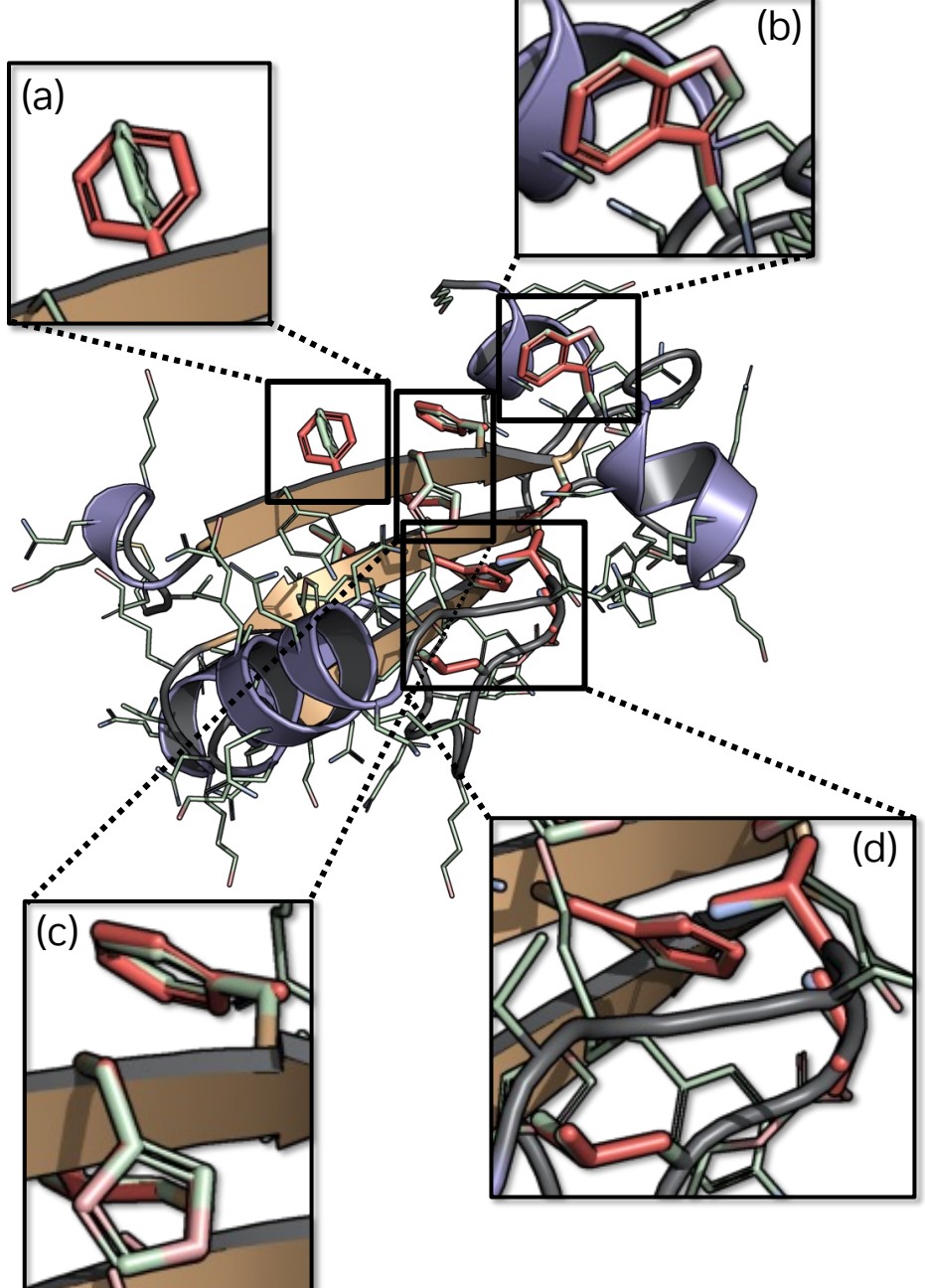

Figure 11: **Atomistic Motif Scaffolding**. Task 5YUI (carbonic anhydrase). The active site here combines a metal coordination site (HIS residues) with a hydrophobic substrate channel (VAL and TRP residues). *La-Proteina* successfully generates a valid atomistic scaffold and accurately reproduces the **red** conditioning atoms that form the tip of partially given side chains (see zoom-ins (b)-(d)). A small inconsistency can be observed in (a), where the model generates an incorrectly rotated ring (we found such inconsistencies to be extremely rare). Side chains that involve conditioning atoms are visualized as thick sticks, all other side chains are shown as thin sticks. Visualization overlays generated protein and atomistic motif.

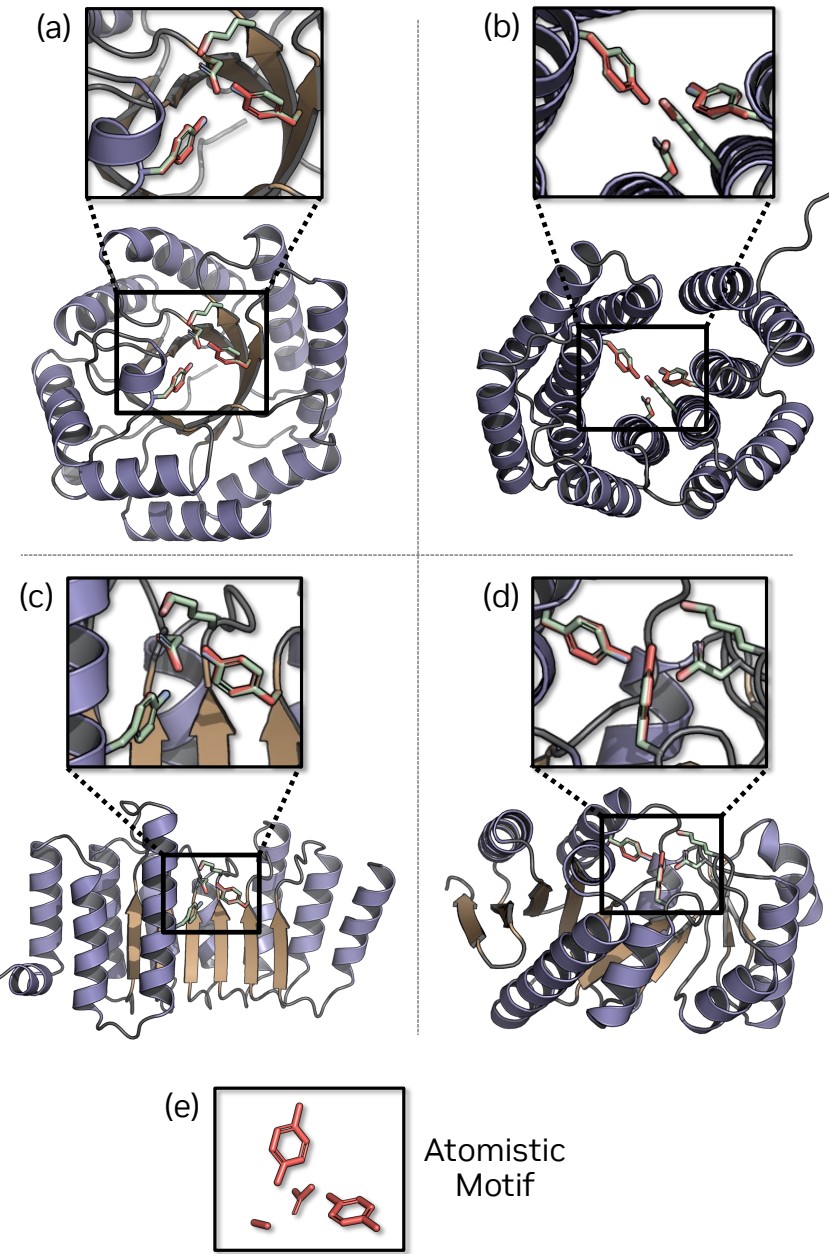

Figure 12: **Atomistic Motif Scaffolding**. Task 5AOU (retro-aldolase). *La-Proteina* successfully generates diverse valid atomistic scaffolds and accurately reproduces the **red** conditioning atoms that form the tip of partially given side chains (see zoom-ins (a)-(d)). The atomistic motif is shown in (e) consisting of a catalytic tetrad that emerged during directed evolution in the laboratory Obexer et al. (2017), with the LYS acting as catalytic nucleophile, the two TYR stabilizing the transtion state and participating in proton transfer and the ASN maintaining the hydrogen-bond network that connects and spatially arranges all tetrad residues. We see that *La-Proteina* can produce diverse solutions to the scaffolding task (shown in the four quadrants of the figure; note that each protein is visualized from different angles for best views of the active site). For clarity, we are only showing side chains of residues that involve conditioning atoms; all other side chains are generated, too, but not shown. Visualization overlays generated protein and atomistic motif.

## C  Unconditional Generation

### C.1  Datasets

We use two datasets to train our unconditional models, one based on the cluster representatives of the Foldseek (van Kempen et al., 2024) clustered version of the AFBD, and another one based on a custom subset of the AFDB (for our long chain evaluation).

**Foldseek Clustered AFDB.** This dataset, previously used by Lin et al. (2024); Geffner et al. (2025), is a filtered and clustered rendition of the AlphaFold Database (AFDB) (Barrio-Hernandez et al., 2023). The clustering employs both sequence (via MMseqs2 (Steinegger & Söding, 2017)) and structure (via Foldseek (van Kempen et al., 2024)) information. The resulting dataset is composed of cluster representatives, meaning one structure is selected from each cluster. This initially yields approximately three million unique samples. We further refine this set based on several criteria: a minimum average pLDDT score of 80, protein lengths constrained to the 32-512 residue range, and specific secondary structure characteristics. For the latter, samples are retained only if their coil proportion is below 50% and they contain no more than 20 consecutive coil residues (these coil filters are variants of those proposed by Qu et al. (2024)). Critically, we also enforce the presence of beta sheets in the selected samples. This beta sheet filter was introduced because models trained without it, despite achieving state-of-the-art metrics, generated proteins with a low beta-sheet content (around 3-4%). Incorporating this filter corrects this imbalance, leading to models that produce samples with an average beta-sheet content of approximately 10%. These cumulative filtering steps result in a final curated dataset of approximately 350k protein samples.

**Custom AFDB subset for long length training.** To create a dataset that is focused on longer samples, we created a custom dataset starting from the AlphaFold database. We filtered for a minimum average pLDDT of 70 and a length between 384 and 896, resulting in 46,942,694 structures. For training we then cluster with MMSeqs2 at a sequence similarity of 50% and sample then randomly from the resulting 4,035,594 clusters at training time.

### C.2  Training Details

#### C.2.1  VAE training

The details of the VAE encoder and decoder architecture are given in App. H. Briefly, both networks consist of 12 transformer layers, totaling approximately 130M parameters. These architectures are trained jointly maximizing the Evidence Lower Bound from Eq. (3). We optimize using AdamW (Loshchilov & Hutter, 2017) with a learning rate of 0.0001 and a weight decay factor of 0.01. We also use exponential moving average with a decay of 0.999. VAEs are trained on the Foldseek clustered AFDB (without including the filter for the beta sheet content). We train in multiple stages: (i) Filtering for proteins between 32 and 256 residues, for 500k steps, on 16 NVIDIA A100-80GB GPUs; (ii) Filtering for proteins between 32 and 512 residues, for 140k steps, on 32 NVIDIA A100-80GB GPUs; (iii) Filtering for proteins between 32 and 896 residues, for 180k steps, on 32 NVIDIA A100-80GB GPUs. We use the VAE parameters obtained after stage (ii) to train flow matching models limited up to 512 residues, and use the VAE obtained after step (iii) to train our flow matching model for longer proteins, up to 800 residues. For all models we use exponential moving average with a decay factor of 0.999.

#### C.2.2  Flow Matching Training

The details of the denoiser network architecture are given in App. H. Briefly, it consists of 14 transformer layers, totaling approximately 160M parameters. We train three models for unconditional generation, minimizing the conditional flow matching loss from Eq. (5). First, one without triangular multiplicative update layers, on the Foldseek Clustered AFDB dataset limited to 512 residues. We train this model for 390k steps, using Adam (Kingma, 2014) with a learning of 0.0001, on 48 NVIDIA A100-80GB GPUs. Second, a model with triangular multiplicative update layers, on the Foldseek Clustered AFDB dataset limited to 512 residues. We train this model for 120k steps, using Adam with a learning rate of 0.0001, on 96 NVIDIA A100-80GB GPUs. Third, a model without triangular multiplicative update layers for proteins of longer lengths, trained on our custom AFDB subset for long length proteins up to 896 residues (App. C.1). We train this model for 140k steps, using Adam

with a learning rate of $0.0001$, on 128 NVIDIA A100-80GB GPUs. For all models we use exponential moving average with a decay factor of $0.999$.

As discussed in Sec. 3.2.2, the interpolation times for $\alpha$-carbon coordinates, $t_x$, and for latent variables, $t_z$, are sampled independently using the distributions from Eq. (6). This distributions are visualized in Fig. 13.

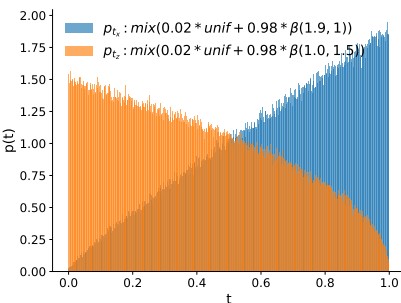

Figure 13: *La-Proteina* sampling distributions for interpolation times $t_x$ and $t_z$.

## C.3 BASELINE SAMPLING

Our main evaluation compares *La-Proteina* against publicly available models for all-atom generation, including P(all-atom) (Qu et al., 2024), PLAID (Lu et al., 2024), Protpardelle (Chu et al., 2024), ProteinGenertor (Lisanza et al., 2023), and APM Chen et al. (2025). For each baseline we produce 100 samples for each protein length in $\{100, 200, 300, 400, 500, 600, 700, 800\}$ (for a total of 800 samples per model) using the official implementation from the corresponding Github repository.

**P(all-atom).** We use the code and weights as described in the original implementation.[3] This model relies on triangular attention layers (Jumper et al., 2021), which have a cubic memory and computational complexity. This limits the length of the proteins that P(all-atom) can generate. Using a GPU with 140GB of RAM, we were unable generate samples beyond 500 residues, due to running out of memory. (This is for generating a single sample.)

**Protpardelle.** We follow the instructions in the original repository using the `allatom_state_dict.pth` checkpoint.[4]

**Protpardelle-1c.** This model is Protpardelle's successor, with a focus on conditional generation (atomistic motif scaffolding and protein complexes). We follow the instructions in the original repository using the `cc91_epoch383` model.[5] We note that this model was trained conditionally for atomistic motif scaffolding, which takes an atomistic motif as input (conditioning information). However, Lu et al. (2025) did not train an all-atom unconditional model. We therefore sample the conditional model unconditionally, simply done by not providing an atomistic motif as conditioning input.

**PLAID.** We use the 100M parameter model, as described in the original implementation.[6] The lengths of proteins sampled with PLAID are $\{96, 200, 296, 400, 496, 600, 696, 800\}$, since the model only supports sampling proteins whose length is divisible by eight.

**ProteinGenerator.** We follow the instructions in the original implementation using the `base` checkpoint,[7] using 100 steps to generate each sample since this is the recommended setting for higher quality, especially at longer lengths.

---

[3] `https://github.com/levinthal/Pallatom`
[4] `https://github.com/ProteinDesignLab/protpardelle`
[5] `https://github.com/ProteinDesignLab/protpardelle`
[6] `https://github.com/amyxlu/plaid`
[7] `https://github.com/RosettaCommons/protein_generator`

**APM.** We follow the instructions for uncondiotinal generation in the original implementation, using the default values for all parameters.[8]

Fig. 4 in the main paper reports metrics for the backbone design task, in which the sequence and all-atoms except the $\alpha$-carbons are ignored. For this specific set of results, we also compare against several backbone design methods, including Chroma (Ingraham et al., 2023), Proteina (Geffner et al., 2025), Proteus (Wang et al., 2024a), Genie2 (Lin et al., 2024), FoldFlow (Bose et al., 2024), RFDiffusion (Watson et al., 2023), FrameFlow (Yim et al., 2023a), FrameDiff (Yim et al., 2023b), and ESM3 (Hayes et al., 2024). For these models, we got the results from Geffner et al. (2025), making sure we use exactly the same metrics reported in that work, to enable direct comparisons.

# D  EVALUATION METRICS

## D.1  CO-DESIGNABILITY, DESIGNABILITY, DIVERSITY, NOVELTY

**Co-designability.** The co-designability metric captures the degree to which between the sequence-structure pairs produced by a model are aligned, by analyzing whether the produced sequence folds into the corresponding structure. This is done by measuring the all-atom RMSD between the structure produced by the model and the structure obtained using ESMFold (Lin et al., 2023) to fold the corresponding sequence. If this all-atom RMSD is less than 2Å, the sample is deemed all-atom co-designable. The metric reported is the percentage of co-designable samples produced by a model.

**Designability.** Designability, on the other hand, aims to capture whether there is a sequence that folds into the produces structure (it ignores the produced sequence). This metric is typically used to evaluate backbone design models, which do not produce sequences. Given the produced structure, ProteinMPNN (Dauparas et al., 2022) is used to generate a set of M sequences (using a sampling temperature of 0.1), ESMFold (Lin et al., 2023) is used to fold all M sequences, and finally the $\alpha$-carbon RMSD between the original structure and each of the ESMFold produced structures is measured. A sample is deemed designable if the minimum of these M RMSD values is less than 2Å. We report two variants of this metric, using M=1 and M=8, denoted as MPNN-1 and MPNN-8 in Tab. 1.

**pLDDT.** The Predicted Local Distance Difference Test (pLDDT) is a per-residue confidence score, scaled from 0 to 100, that estimates the local structural accuracy of a predicted protein model (in our case ESMFold). Generally, the threshold of pLDDT $\geq 80$ is used to indicate high confidence and reliable predictions. The metric for the full protein is obtained by averaging these per-residue scores across the entire structure, averaged over succesfully refolded samples (i.e., all-atom co-designable samples).

**Diversity.** All three diversity metrics ("Str", "Seq", "Str+Seq") reported in Tab. 1 are obtained by clustering the subset of all-atom co-designable samples produced by a model and reporting the number of clusters obtained. The difference between these metrics is the clustering criteria used. Briefly, "Str" measures the diversity in the produced structures (ignoring sequence), "Seq" measures the diversity in the produced sequences (ignoring structures), and "Str+Seq" measures the diversity taking into account both the sequence and structure of the samples produced.

- Structure diversity ("Str"). We cluster using the Foldseek command

  ```
  foldseek easy-cluster <path_samples> <path_results> <path_tmp>
  --cov-mode 0
  --alignment-type 1
  --min-seq-id 0
  --tmscore-threshold 0.5
  ```

  where `<path_samples>` is the path to a directory containing all-atom co-designable samples, `<path_results>` is the directory where results will be stored, and `path_tmp` is the directory used to store temporary files used by the clustering algorithm. This command clusters all produced structures without taking the corresponding sequences into account.

- Joint structure and sequence ("Str+Seq"). We cluster using the Foldseek command

---

[8]https://github.com/bytedance/apm

```
 foldseek easy-cluster <path_samples> <path_results> <path_tmp>
--cov-mode 0
--alignment-type 2
--min-seq-id 0.1
--tmscore-threshold 0.5
```

- Sequence diversity ("Seq"). We cluster using the MMSeqs2 command

```
mmseqs easy-linclust <fasta_input_filepath> pdb_cluster <path_tmp>
--min-seq-id 0.1
--c 0.7
--cov-mode 1
```

where `<fasta_input_filepath>` is the path for the fasta file containing the sequences for all-atom co-designable samples.

**Novelty.** This metric assesses the structural similarity between samples generated by a model and a defined reference set, where lower scores signify greater novelty (i.e., less resemblance to known structures). To calculate this, we compute the TM-Score Zhang & Skolnick (2004) between each all-atom co-designable sample generated by the model and every protein within the specified reference set. For each generated sample, its maximum TM-Score, reflecting its similarity to the closest structure in the reference set, is identified. The average of these maximum scores across every all-atom co-designable samples is then reported as the novelty value. Given that TM-Scores range from 0 to 1, with higher scores indicating higher similarities, lower novelty scores are preferable. Tab. 1 presents novelty values against two reference sets: the PDB, as provided by Foldseek van Kempen et al. (2024) (labeled "PDB" in the table), and a filtered version of the Foldseek Clustered AFDB, detailed in App. C.1 (minimum average pLDDT of 80, lengths 32-512 residues; labeled "AFDB" in the table). We use Foldseek (van Kempen et al., 2024) to compute TM-Scores of the produced samples against the corresponding reference set. The Fodlseek command used to compute this metric is given by

```
foldseek easy-search <path_sample> <reference_database_path>
<path_results> <tmp_path>
--alignment-type 1
--exhaustive-search
--tmscore-threshold 0.0
--max-seqs 10000000000
--format-output query,target,alntmscore
```

where `<path_sample>` is the path for the PDB file containing the generated structure, and `<reference_database_path>` is the path of the dabaset used as reference.

**Note on Foldseek version used for novelty computation.** Tab. 1 reports Novelty results using Foldseek version 9.427df8a. We note that newer Foldseek versions return different values for the exact same command and samples. Tab. 11 shows Novelty results for all methods using Foldseek version 10.941cd33. The table only reports Novelty values, as all other metrics are exactly the same as in Tab. 1.

## D.2 MOLPROBITY FOR STRUCTURAL QUALITY ASSESMENT

MolProbity (Davis et al., 2007) is a widely used software designed for comprehensive validation of 3D macromolecular structures, primarily proteins and nucleic acids. It assesses the quality of a structure by analyzing its geometry, stereochemistry, and interatomic contacts against well-established chemical and physical principles derived from high-resolution experimental data. Its goal is to identify problematic regions in a structure that may indicate errors or physically unrealistic conformations.

For our comparative analysis of generated protein structures, we focused on the following key metrics reported by MolProbity:

**MolProbity Score (MP score):** This is a composite score that combines multiple individual geometric assessments (including clash score, Ramachandran favorability, and side-chain rotamer quality) into a single, log-weighted metric. It provides an overall indication of structural quality. Lower MP scores are better; scores around 1.0-2.0 are generally indicative of well-resolved and accurate experimental

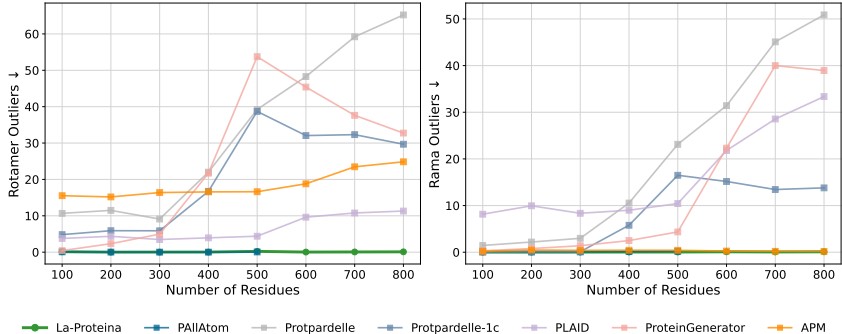

Figure 14: **Additional MolProbity metrics.** Rotamer outliers and Ramachandran outliers. While most baselines degrade especially at longer lengths, *La-Proteina* and P(all-atom) have realistic scores for all lengths. P(all-atom) evaluation only goes up to 500 residues due to memory limitations, for longer samples more than 140GB of GPU memory are needed to produce a single sample. P(all-atom) and *La-Proteina* lines mostly overlap until length 500 when the P(all-atom) line stops.

structures, while scores significantly above 2.5-3.0 often suggest increasing numbers of geometric and stereochemical issues.

**Clash Score:** This metric quantifies the severity of steric clashes by reporting the number of unfavorable all-atom overlaps (where van der Waals shells interpenetrate by $\geq 0.4$Å) per 1000 atoms. A lower clash score signifies a more sterically reasonable structure. While there's no absolute cutoff, high-resolution X-ray crystal structures typically have clash scores below 20, often much lower (e.g., <10). NMR structures or lower-resolution crystal structures may exhibit higher values (e.g., up to 50-60 or more could still be acceptable depending on context), but excessively high scores indicate significant packing problems.

**Ramachandran Angle Outliers:** This evaluates the conformational plausibility of the protein backbone by analyzing the Ramachandran plot, which describes allowed regions for the phi ($\phi$) and psi ($\psi$) dihedral angles of amino acid residues. The metric reports the percentage of residues whose $(\phi, \psi)$ angles fall into disallowed (outlier) regions. For high-quality structures, this value is expected to be very low, ideally less than 0.2%, with modern well-refined structures often achieving <0.1% outliers.

**Covalent Bond Geometry Outliers (Bond Lengths and Angles):** This metric assesses the correctness of covalent geometry by comparing observed bond lengths and bond angles to standard dictionary values. It typically reports the percentage of bonds or angles that deviate significantly (e.g., by more than 4 standard deviations, or other thresholds defined by MolProbity) from these ideal values. A low percentage of outliers (ideally <1% for both lengths and angles combined, or individually) indicates good covalent geometry.

**Rotamer Outliers:** This metric evaluates the plausibility of side-chain conformations by comparing the observed $\chi$ (chi) torsion angles of amino acid residues to distributions derived from high-quality experimental structures. MolProbity uses a comprehensive, data-driven rotamer library (the "ultimate" rotamer library) constructed from a large set of rigorously filtered protein chains to define statistically favored, allowed, and outlier regions for side-chain dihedral angles. A residue is classified as a rotamer outlier if its side-chain conformation falls into a region sampled by less than 0.3% of reference structures, indicating a highly unusual or energetically unfavorable state. High-quality protein structures typically exhibit less than 1% rotamer outliers, with modern structure determination and refinement methods often achieving even lower values. Elevated levels of rotamer outliers may suggest errors in side-chain modeling, poor electron density, or physically unrealistic conformations, and thus serve as a sensitive indicator of local model quality.

Together, these MolProbity metrics offer a robust and multi-faceted evaluation of the atomistic accuracy and realism of the generated protein structures. MP score, clash score, bond length outliers and bond angle outliers are visulized in Fig. 5, while ramachandran angle outliers and rotamer outliers are depicted in Fig. 14. In all of these we see that *La-Proteina* generates highly realistic structures at all lengths, whereas all other baselines generate less plausible structures and especially degrade at longer lengths (the exception is P(all-atom) that also has bad scores for clashes, angle outliers and bond outliers but scores well for rotamer outliers and Ramachandran outliers).

### D.3 Side-Chain Dihedral Angle Distributions

#### D.3.1 Background on amino acid rotamers

When investigating side-chain conformations in protein structures, one quickly recognizes that these side-chain torsion angles (denoted by $\chi_1$, $\chi_2$, etc., down the side chain) do not appear randomly and do not usually occur in broader regions such as backbone torsion angles which are usually visualized in Ramachandran plots Carugo & Djinović-Carugo (2013), but cluster into distinct conformations that are called rotamers, i.e., chemical species that differ from one another mostly due to rotations about one or more single bonds Dunbrack Jr (2002).

This discreteness of the side-chain degrees-of-freedom is caused by steric repulsion between atoms three bonds away from each other, at the end of the atoms making up the plane of the torsion angle under question. To not cause too much steric repulsion, these groups usually prefer to adopt staggered conformations in which they are 60 degrees off-set to the next group instead of eclipsed conformations where they overlap with this next group Clayden et al. (2012). The three possible staggered conformations (gauche plus at 60 degrees, gauche minus at -60 degrees and trans at 180 degrees between the two groups under question) are the major rotamers that are visible in most $\chi_1$ and several $\chi_2$ plots Lovell et al. (2000). For example, in the case of $\chi_1$, the plane of this torsion angle is formed by the CA and CB atom and the atoms under question for staggering are the N and for example the CG1 in the case of VAL and ILE or the OG in the case of SER. Due to this, the angle $\chi_1$ is always rotameric at +60, 180 and -60/300 degrees (i.e. it falls into discrete angles), except for alanine which only has a hydrogen instead of CG and therefore no $\chi_1$ rotamer and glycine which has neither CB nor CG.

However, the populations of these rotamers are different based on amino acid identity. Usually the preference declines in the order of g- (-60), trans (180), and g + (60), but there are exceptions. PRO for example has a tight ring structure that only allows for two $\chi_1$ rotamers at around -30 and +30 degrees (Fig. 27). SER and THR on the other hand prefer the g+ (60) rotamer since in that conformation it can form a hydrogen bond to the backbone with their oxygen atom. ILE, LEU, and THR have two gamma heavy atoms, which cause one rotamer to always be in an unfavorable conformation; these amino acids only show two $\chi_1$ rotamers with significant populations.

There are also non-rotameric degrees of freedom. While in ARG for example both $\chi_1$ and $\chi_2$ are rotamer (Fig. 15), leading to 9 configurations, ASP for example has a non-rotameric $\chi_2$ angle that spreads over a rather continuous spectrum (Fig. 17). These non-rotameric degres of freedom are always the last one in the side chain, i.e. the furthest away from the backbone. In the case of ASN and ASP this is $\chi_2$ (Fig. 16 and Fig. 17), whereas in the case of GLN and GLU this is $\chi_3$ (Fig. 19 and Fig. 20 first row). Beyond this, there are further factors determining rotamer populations, either backbone-independent effects like syn-pentane interactions Dunbrack Jr & Karplus (1994) or backbone-dependent ones Chakrabarti & Pal (2001).

#### D.3.2 Analysis of generated amino acid rotamers

To not only look at outright rotamer outliers, but also rotamer frequencies and mode coverage, we visualize Kernel Density Estimation (KDE) plots for all side chain angles of all amino acids in Figs. 15 to 31. We conduct this analysis for the samples generated for *La-Proteina*, all baselines, and two reference datasets from the PDB and AFDB (100 structures for each length of 100 to 800 in steps of 100). The PDB data set was curated by selecting 100 X-ray structures with a resolution below 2Å of the respective length ±5 residues (for length 800, which leads to 60 structures). The AFDB reference data set was curated similarly, just with the filtering threshold being a pLDDT score above 80 and a radius of gyration of less than 3 to avoid overrepresentation of side-chain angles corresponding to extended alpha-helices.

As in the main text, we see that *La-Proteina* often captures not only the correct modes, but often also at approximately the correct rotamer frequencies with respect to the reference datasets from the PDB and AFDB. This can be seen, for instance, for ARG $\chi_3$ (Fig. 15), HIS $\chi_2$ (Fig. 21) or PRO $\chi_1$ (Fig. 27). P(all-atom) and Protpardelle often miss modes completely, while PLAID and ProteinGenerator often get the modes correctly but represent them in different frequencies compared to the base dataset. We also see that for some side-chain angles, the distribution between PDB and AFDB differ significantly, as for ARG $\chi_4$ (Fig. 15), LYS $\chi_3$ (Fig. 24) and LYS $\chi_4$ (Fig. 24 sixth row

left). In these cases, *La-Proteina* adheres more closely to the AFDB reference since it was trained on AFDB structures; however, interestingly none of the other methods capture the PDB modes here as well despite being trained on datasets including the PDB.

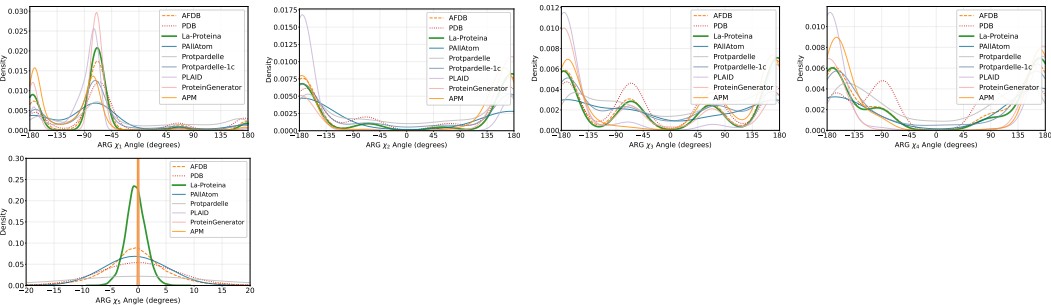

Figure 15: Side-chain angles for amino acid ARG.

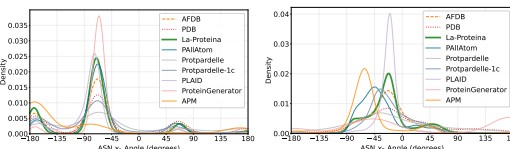

Figure 16: Side-chain angles for amino acid ASN.

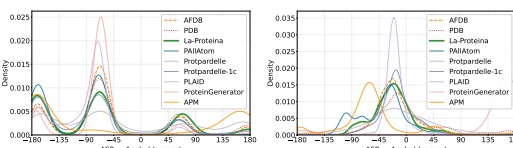

Figure 17: Side-chain angles for amino acid ASP.

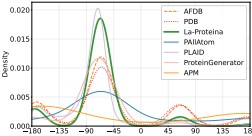

Figure 18: Side-chain angles for amino acid CYS.

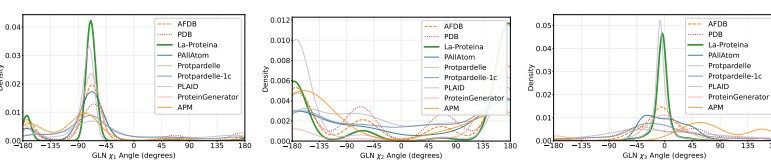

Figure 19: Side-chain angles for amino acid GLN.

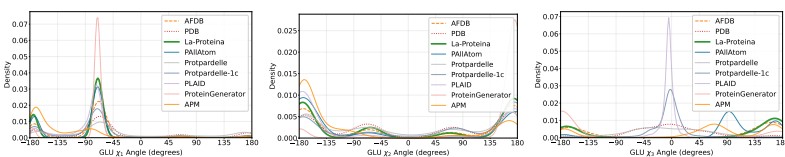

Figure 20: Side-chain angles for amino acid GLU.

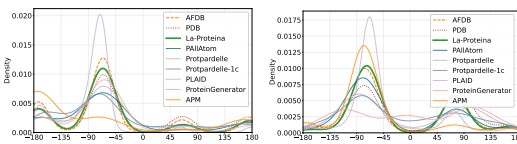

Figure 21: Side-chain angles for amino acid HIS.

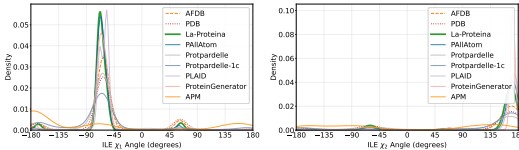

Figure 22: Side-chain angles for amino acid ILE.

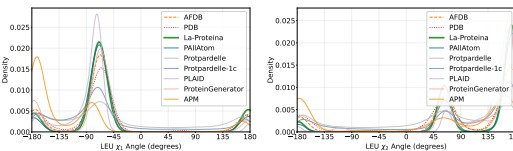

Figure 23: Side-chain angles for amino acid LEU.

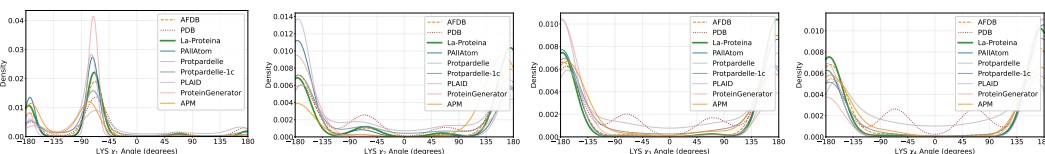

Figure 24: Side-chain angles for amino acid LYS.

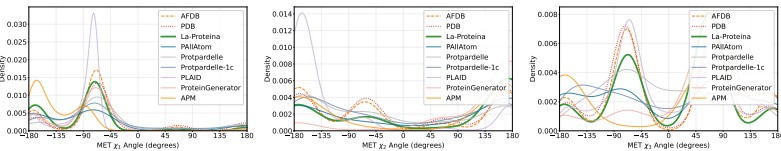

Figure 25: Side-chain angles for amino acid MET.

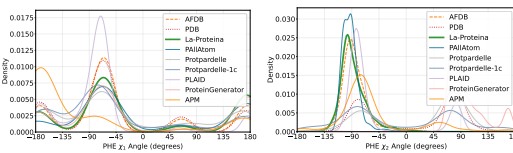

Figure 26: Side-chain angles for amino acid PHE.

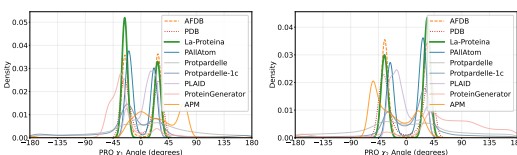

Figure 27: Side-chain angles for amino acid PRO.

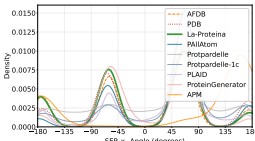

Figure 28: Side-chain angles for amino acid SER.

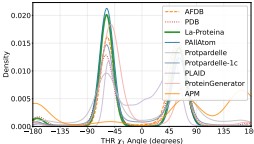

Figure 29: Side-chain angles for amino acid THR.

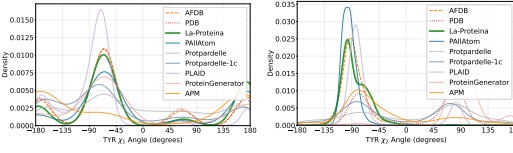

Figure 30: Side-chain angles for amino acid TYR.

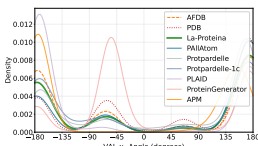

Figure 31: Side-chain angles for amino acid VAL.

# E  SAMPLING

We sample *La-Proteina* by numerically simulating the SDE from Eq. (7). This SDE relies on the score function (gradient of log probability) of intermediate densities. Since we use a Gaussian flow and linear interpolants, we can compute these directly from the learned vector field $v_\theta$ as (Ma et al., 2024; Zheng et al., 2023)

$$\zeta^x(\mathbf{x}_{C_\alpha}^{t_x}, \mathbf{z}^{t_z}, t_x, t_z) = \frac{t\,\mathbf{v}_\phi^x(\mathbf{x}_{C_\alpha}^{t_x}, \mathbf{z}^{t_z}, t_x, t_z) - \mathbf{x}_{C_\alpha}^{t_x}}{1 - t_x} \approx \nabla_{\mathbf{x}_{C_\alpha}^{t_x}} \log p_\phi(\mathbf{x}_{C_\alpha}^{t_x}, \mathbf{z}^{t_z}, t_x, t_z) \quad (8)$$

$$\zeta^z(\mathbf{x}_{C_\alpha}^{t_x}, \mathbf{z}^{t_z}, t_x, t_z) = \frac{t\,\mathbf{v}_\phi^z(\mathbf{x}_{C_\alpha}^{t_x}, \mathbf{z}^{t_z}, t_x, t_z) - \mathbf{z}^{t_z}}{1 - t_z} \approx \nabla_{\mathbf{z}^{t_z}} \log p_\phi(\mathbf{x}_{C_\alpha}^{t_x}, \mathbf{z}^{t_z}, t_x, t_z). \quad (9)$$

Simulating the SDE from Eq. (7) requires selecting the noise scaling parameters $\eta_x$ and $\eta_z$ and the scaling functions $\beta_x(t_x)$ and $\beta_z(t_z)$, which modulate the Langevin-like term in the SDE. For the former, we experiment with values in $[0, 1]$, noting that $\eta_x = \eta_z = 1$ yields "unbiased sampling" (for any choice of $\beta_x$ and $\beta_z$ (Karras et al., 2022)), and smaller values sample distributions which differ from the original one defined by the flow matching model (often referred to as "low temperature sampling" (Geffner et al., 2025; Ingraham et al., 2023)).[9] For the scaling functions we use

$$\beta_x(t_x) = \frac{1}{t_x} \quad \text{and} \quad \beta_z(t_z) = \frac{\pi}{2} \tan\left(\frac{\pi}{2}\left(1 - t_z\right)\right). \quad (10)$$

We show ablations for these choices in App. G.

## E.1  NUMERICAL DISCRETIZATION SCHEME

We simulate the system of stochastic differential equations from Eq. (7) using the Euler-Maruyama method (Higham, 2001). Since $t_x$ and $t_z$ are sampled independently (as discussed in Sec. 3.2.2), the model allows the exploration of different paths going from $(t_x, t_z) = (0, 0)$ to $(t_x, t_z) = (1, 1)$ (that is, different paths in the $[0, 1] \times [0, 1]$, space). We parameterize these paths by defining $t_x = f_x(t)$ and $t_z = f_z(t)$ using a shared time variable $t \in [0, 1]$, where $f_x, f_z : [0, 1] \to [0, 1]$ are monotonically increasing functions. As highlighted in Secs. 1 and 3.4, using distinct schedules $f_x(t)$ and $f_z(t)$ for the $\alpha$-carbon coordinates $\mathbf{x}_{C_\alpha}$ and latent variables $\mathbf{z}$ is critical for good performance. More specifically, our empirical analyses show that schedules evolving $\mathbf{x}_{C_\alpha}$ faster than $\mathbf{z}$ yield the best results (see App. G). We therefore adopt an "exponential" schedule (Geffner et al., 2025) for $f_x(t)$ and a "quadratic" schedule for $f_z(t)$

$$f_x(t) = \frac{1 - 10^{-2t}}{1 - 10^{-2}} \quad \text{and} \quad f_z(t) = t^2, \quad (11)$$

visualized in Fig. 32. The corresponding numerical integration scheme is obtained by uniformly partitioning the interval $t \in [0, 1]$ (i.e., $t_n = n/N$ for $n = 0, 1, \ldots, N$), yielding the discrete steps

$$t_x[n] = f_x(t_n) = \frac{1 - 10^{-2n/N}}{1 - 10^{-2}} \quad \text{and} \quad t_z[n] = f_z(t_n) = \left(\frac{n}{N}\right)^2. \quad (12)$$

Ablations for different choices of $f_x(t)$ and $f_z(t)$ are presented in App. G. For all our experiments we use $N = 400$ integration steps.

# F  ATOMISTIC MOTIF SCAFFOLDING

For atomistic motif scaffolding we included two different tasks: *all-atom motif scaffolding* and *tip-atom motif scaffolding*. For all-atom motif scaffolding, for a certain selection of residues (the motif) information about backbone position, side chain positions as well as amino acid identity is provided and the task of the model is to generate a new protein that includes this motif as part of it. For tip-atom motif scaffolding, the provided information includes only the amino acid identity as well

---

[9]Most existing generative models for protein design rely on some variant of low temperature sampling (Ingraham et al., 2023; Yim et al., 2023b;a; Watson et al., 2023; Lin et al., 2024; Bose et al., 2024; Wang et al., 2024a; Huguet et al., 2024; Campbell et al., 2024; Geffner et al., 2025).

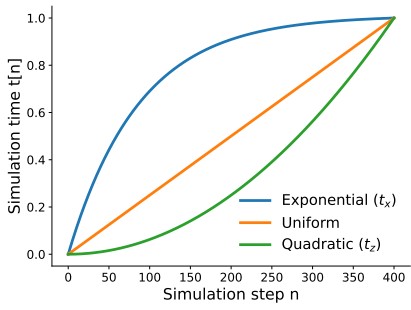

Figure 32: Discretization schemes, including uniform as reference.

as the coordiantes of the side chain atoms after the final rotatable bond. This means the following atoms are made available for the respective amino acids, following the task definition of Protpardelle (Chu et al., 2024):

$$
\begin{aligned}
\text{ALA:} &\quad \{\text{CA, CB}\} \\
\text{ARG:} &\quad \{\text{CD, CZ, NE, NH1, NH2}\} \\
\text{ASP:} &\quad \{\text{CB, CG, OD1, OD2}\} \\
\text{ASN:} &\quad \{\text{CB, CG, ND2, OD1}\} \\
\text{CYS:} &\quad \{\text{CA, CB, SG}\} \\
\text{GLU:} &\quad \{\text{CG, CD, OE1, OE2}\} \\
\text{GLN:} &\quad \{\text{CG, CD, NE2, OE1}\} \\
\text{GLY:} &\quad \{\} \\
\text{HIS:} &\quad \{\text{CB, CG, CD2, CE1, ND1, NE2}\} \\
\text{ILE:} &\quad \{\text{CB, CG1, CG2, CD1}\} \\
\text{LEU:} &\quad \{\text{CB, CG, CD1, CD2}\} \\
\text{LYS:} &\quad \{\text{CE, NZ}\} \\
\text{MET:} &\quad \{\text{CG, CE, SD}\} \\
\text{PHE:} &\quad \{\text{CB, CG, CD1, CD2, CE1, CE2, CZ}\} \\
\text{PRO:} &\quad \{\text{CA, CB, CG, CD, N}\} \\
\text{SER:} &\quad \{\text{CA, CB, OG}\} \\
\text{THR:} &\quad \{\text{CA, CB, CG2, OG1}\} \\
\text{TRP:} &\quad \{\text{CB, CG, CD1, CD2, CE2, CE3, CZ2, CZ3, CH2, NE1}\} \\
\text{TYR:} &\quad \{\text{CB, CG, CD1, CD2, CE1, CE2, CZ, OH}\} \\
\text{VAL:} &\quad \{\text{CB, CG1, CG2}\}
\end{aligned}
$$

We also evaluate two distinct scaffolding setups that differ in their conditioning information. In the standard *indexed* task, the model is provided with the sequence positions for each motif residue. In the more challenging *unindexed* task, these indices are withheld, requiring the model to discover a viable placement for the motif while simultaneously generating the scaffold.

## F.1    TRAINING

We train the motif scaffolding models following the same training procedure as for the main models, with additional input features extracted from the motif. In the case of all-atom motif scaffolding, these features include (for the motif's residues) absolute atomic coordinates, coordinates relative to the corresponding $\alpha$-carbon atom, residue type, side chain angles, and backbone torsion angles. For tip-atom motif scaffolding, these features only include absolute atomic coordinates of the atoms present in the motif (i.e. atoms after the last rotatable bond) and residue type. For the indexed version,

Table 2: Motif data with minimum and maximum lengths, and contig strings (all atom and tip atom).

| Motif Name | Min Length | Max Length | Contig String All Atom | Contig String Tip Atom |
|---|---|---|---|---|
| 1PRW_AA | 60 | 105 | 5-20/A1-20/10-25/B1-20/5-20 | 5-20/A16-22/1/A24/1/A26-32/1/A34-35/10-25/A52-58/1/A60/1/A62-71/5-20 |
| 1BCF_AA | 96 | 152 | 8-15/A92-99/16-30/A123-130/16-30/A47-54/16-30/A18-25/8-15 | 8-15/A92-96/1/A98-99/16-30/A123-128/1/A130/16-30/A47-54/16-30/A18-25/8-15 |
| 5TPN_AA | 50 | 75 | 10-40/A163-181/10-40 | 10-40/A163-181/10-40 |
| 5IUS_AA | 57 | 142 | 0-30/B119-140/15-40/A63-82/0-30 | 1-31/A120-123/1/A125-130/1/A132-140/15-40/A63-73/1/A75-82/0-30 |
| 3IXT_AA | 50 | 75 | 10-40/P254-277/10-40 | 10-40/P254-277/10-40 |
| 5YUI_AA | 50 | 100 | 5-30/A93-97/5-20/B118-120/10-35/C198-200/10-30 | 5-30/A93-97/5-20/A118-120/10-35/A198-200/10-30 |
| 5AOU_AA | 230 | 270 | 40-60/A1051/20-40/A2083/20-35/A2110/100-140 | 40-60/A1051/20-40/A2083/20-35/A2110/100-140 |
| 5AOU_QUAD_AA | 230 | 270 | 40-60/A1051/20-40/A2083/20-35/A2110/60-80/A2180/40-60 | 40-60/A1051/20-40/A2083/20-35/A2110/60-80/A2180/40-60 |
| 7K4V_AA | 280 | 320 | 40-50/A44/3-8/A50/70-85/A127/150-200 | 40-50/A44/3-8/A50/70-85/A127/150-200 |
| 1YCR_AA | 40 | 100 | 10-40/B19-27/10-40 | 10-40/B19-27/10-40 |
| 4JHW_AA | 60 | 90 | 10-25/F196-212/15-30/F63-69/10-25 | 10-25/F196-212/15-30/F63-69/10-25 |
| 5WN9_AA | 35 | 50 | 10-40/A170-189/10-40 | 10-40/A170-186/1/A188-189/10-40 |
| 4ZYP_AA | 30 | 50 | 10-40/A422-436/10-40 | 10-40/A422-429/1/A431-436/10-40 |
| 6VW1_AA | 62 | 83 | 20-30/A24-42/4-10/A64-82/0-5 | 20-30/A24-42/4-10/A64-65/1/A67-82/0-5 |
| 1QJG_AA | 53 | 103 | 10-20/A38/15-30/A14/15-30/A99/10-20 | 10-20/A14/15-30/A38/50-70/A99/25-30 |
| 1QJG_AA_NATIVE | 115 | 135 | 10-20/A14/15-30/A38/50-70/A99/25-30 | 10-20/A14/15-30/A38/50-70/A99/25-30 |
| 2KL8_AA | 79 | 79 | A1-7/20/A28-79 | A1-7/20/A28-79 |
| 7MRX_AA_60 | 60 | 60 | 0-38/B25-46/0-38 | 0-38/B25-30/1/B32-42/1/B44-46/0-38 |
| 7MRX_AA_85 | 85 | 85 | 0-63/B25-46/0-63 | 0-63/B25-30/1/B32-42/1/B44-46/0-63 |
| 7MRX_AA_128 | 128 | 128 | 0-122/B25-46/0-122 | 0-122/B25-30/1/B32-42/1/B44-46/0-122 |
| 5TRV_AA_SHORT | 56 | 56 | 0-35/A45-65/0-35 | 1-36/A46-48/1/A50-55/1/A57-59/1/A61-65/0-35 |
| 5TRV_AA_MED | 86 | 86 | 0-65/A45-65/0-65 | 1-66/A46-48/1/A50-55/1/A57-59/1/A61-65/0-65 |
| 5TRV_AA_LONG | 116 | 116 | 0-95/A45-65/0-95 | 1-96/A46-48/1/A50-55/1/A57-59/1/A61-65/0-95 |
| 6E6R_AA_SHORT | 48 | 48 | 0-35/A23-35/0-35 | 0-35/A23-32/1/A34/1-36 |
| 6E6R_AA_MED | 78 | 78 | 0-65/A23-35/0-65 | 0-65/A23-32/1/A34/1-66 |
| 6E6R_AA_LONG | 108 | 108 | 0-95/A23-35/0-95 | 0-95/A23-32/1/A34/1-96 |

these features are added to the corresponding residue indices of the motif; while for the unindexed task they are concatenated to the initial sequence representation without providing any information related to the motif residue indices to the model. The dataset used was the standard dataset used for training the main models, i.e. the Foldseek-clusters of the AFDB with a maximum length of 356 and a minimum average pLDDT of 80. The indexed all-atom motif model was trained for 150k steps on 64 NVIDIA A100-80GB GPUs, and the indexed tip-atom motif model was trained for 120k steps on 128 NVIDIA A100-80GB GPUs. The unindexed models (all-atom and tip-atom) were trained on 32 NVIDIA A100-80GB GPUs for 650k steps.

## F.2 SAMPLING

For sampling, the standard sampling schedule of the main models was used (App. E). The motifs were sampled according to the specifications in the Protpardelle benchmark/RFDiffusion benchmark, with the only difference being that for tip-atom motif scaffolding the residues that did not include any atoms to be scaffolded (Glycine, or Lysine if the tip atoms specified in the description were not present in the motif structure) were excluded from the motif. This resulted in the definition of benchmark tasks in Table 2.

## F.3 EVALUATION

We evaluate each generated sample via four criteria:

1. The sequence of the motif has to be 100% recovered,

2. The motif $\alpha$-carbon coordinates should have an all-atom RMSD <1Å,

3. The motif coordinates should have an all-atom RMSD <2Å,

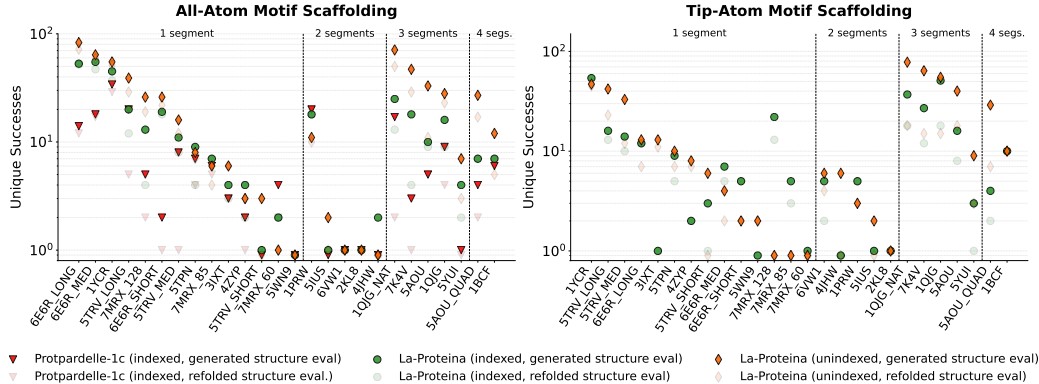

Figure 33: **Atomistic motif scaffolding using generated vs. refolded structures.** 26 atomistic motif-scaffilding tasks (x-axis), comparing Protpardelle-1c (limited to all-atom indexed), *La-Proteina* (indexed) and *La-Proteina* (unindexed). Solid markers indicate performance when evaluating the model produced structures directly (App. F.3), while light markers indicate performance when performing the evaluation using refolded structures instead of the model generated ones (App. F.4). "# segments" refers to the number of residue segments in the motif.

4. The generated protein should be all-atom co-designable, i.e., it should have have an all-atom scRMSD <2Å.

For all methods we generate 200 samples per task. We then evaluate these samples via the criteria above, which results in the number of successes per task. Finally, the number of unique successes is obtained by clustering the successes with Foldseek (van Kempen et al., 2024) and reporting the number of clusters. We use the following command to cluster:

```
foldseek easy-cluster <path_samples> <path_tmp>/res <path_tmp>
--alignment-type 1 --cov-mode 0 --min-seq-id 0
--tmscore-threshold 0.5 --single-step-clustering
```

The full results for all methods can be found in Table 3 for all-atom motif scaffolding and in Table 4 for tip-atom motif scaffolding. Results show that *La-Proteina* outperforms both Protpardelle and Protpardelle-1c. Additionally, we observe that Protpardelle is able to solve 4/26 tasks in both the all-atom and tip-atom setups. This is consistent with the findings reported in the original Protpardelle paper (Chu et al., 2024); our evaluation criteria, as outlined above, align closely with their "strict" definition of success, under which they also report limited task success. While they additionally report results under a more lenient "weak" success criterion, we emphasize that this criterion is easier to satisfy than both their strict definition and our own. Notably, our model already achieves strong performance under the stricter standard, underscoring its robustness even under more challenging evaluation settings.

**Note on indexed vs. unindexed evaluation.** Evaluating motif accuracy via RMSD differs significantly between the *indexed* and *unindexed* scaffolding tasks. In the indexed setting, the motif's sequence indices are known, making the RMSD calculation a straightforward comparison between the known motif residues of the ground truth and generated structures. For the unindexed task, however, these residue indices must first be inferred from the generated output. We address this by employing a greedy matching procedure (Chen et al., 2025): for each residue in the ground truth motif, we identify its structurally closest counterpart in the generated protein. The motif RMSD is then calculated using this newly identified set of residues. Because the model may place the motif at different sequence positions in each sample, this matching process must be performed independently for every generated protein.

## F.4 EVALUATION VIA REFOLDED STRUCTURES

In addition to evaluating the directly generated structures, we conduct an alternative validation by analyzing refolded structures. To this end, we take the sequences produced by each model (*La-*

Table 3: **All-atom motif scaffolding.** "All" indicates total number of successes produced by the model (we produce 200 samples per task), while "Unique" indicates number of unique successes, obtained by clustering all successes as explained in App. F.3. "Indexed" indicates the motif residue indices are provided as input to the model, "Unindexed" indicates that the motif residue indices are not provided as input. "# segments" refers to the number of residue segments in the motif.

| Motif Task | # segments | Protpardelle (indexed) | | Protpardelle-1c (indexed) | | La-Proteina (indexed) | | La-Proteina (unindexed) | |
|---|---|---|---|---|---|---|---|---|---|
| | | All | Unique | All | Unique | All | Unique | All | Unique |
| 6E6R_AA_long | 1 | 0 | 0 | 16 | 14 | 91 | 53 | 110 | 83 |
| 6E6R_AA_med | 1 | 0 | 0 | 19 | 18 | 92 | 55 | 96 | 64 |
| 1YCR_AA | 1 | 1 | 1 | 62 | 34 | 123 | 45 | 97 | 55 |
| 5TRV_AA_long | 1 | 0 | 0 | 22 | 20 | 78 | 20 | 84 | 39 |
| 7MRX_AA_128 | 1 | 0 | 0 | 10 | 5 | 55 | 13 | 83 | 26 |
| 6E6R_AA_short | 1 | 0 | 0 | 2 | 2 | 48 | 19 | 51 | 26 |
| 5TRV_AA_med | 1 | 0 | 0 | 9 | 8 | 68 | 11 | 53 | 16 |
| 5TPN_AA | 1 | 0 | 0 | 16 | 7 | 49 | 9 | 28 | 8 |
| 7MRX_AA_85 | 1 | 0 | 0 | 18 | 6 | 53 | 7 | 100 | 6 |
| 3IXT_AA | 1 | 0 | 0 | 4 | 3 | 25 | 4 | 52 | 6 |
| 4ZYP_AA | 1 | 0 | 0 | 2 | 2 | 7 | 4 | 65 | 3 |
| 5TRV_AA_short | 1 | 0 | 0 | 0 | 0 | 19 | 1 | 7 | 3 |
| 7MRX_AA_60 | 1 | 0 | 0 | 9 | 4 | 64 | 2 | 47 | 1 |
| 5WN9_AA | 1 | 0 | 0 | 0 | 0 | 0 | 0 | 0 | 0 |
| 1PRW_AA | 2 | 0 | 0 | 162 | 20 | 174 | 18 | 120 | 11 |
| 5IUS_AA | 2 | 0 | 0 | 0 | 0 | 5 | 1 | 12 | 2 |
| 6VW1_AA | 2 | 0 | 0 | 14 | 1 | 23 | 1 | 70 | 1 |
| 2KL8_AA | 2 | 80 | 1 | 177 | 1 | 119 | 1 | 193 | 1 |
| 4JHW_AA | 2 | 0 | 0 | 0 | 0 | 2 | 2 | 0 | 0 |
| 1QJG_AA_NAT | 3 | 0 | 0 | 23 | 17 | 62 | 25 | 109 | 71 |
| 7K4V_AA | 3 | 0 | 0 | 11 | 3 | 91 | 18 | 47 | 47 |
| 5AOU_AA | 3 | 0 | 0 | 89 | 5 | 166 | 10 | 35 | 33 |
| 1QJG_AA | 3 | 1 | 1 | 13 | 9 | 50 | 16 | 62 | 28 |
| 5YUI_AA | 3 | 0 | 0 | 1 | 1 | 6 | 4 | 16 | 7 |
| 5AOU_QUAD_AA | 4 | 0 | 0 | 55 | 4 | 144 | 7 | 74 | 27 |
| 1BCF_AA | 4 | 70 | 1 | 196 | 6 | 147 | 7 | 147 | 12 |

*Proteina* and Protpardelle-1c), predict their structures using ESMFold, and then re-evaluate their success. In short, in this case the success criteria is given by

1. The sequence of the motif has to be 100% recovered,

2. The motif $\alpha$-carbon coordinates should have an all-atom RMSD <1Å when comparing to the refolded structure, obtained by running ESMFold on the *La-Proteina* produced sequence,

3. The motif coordinates should have an all-atom RMSD <2Å when comparing to the refolded structure,

4. The generated protein should be all-atom co-designable, i.e., it should have have an all-atom scRMSD <2Å.

The results of this refolding analysis for all-atom and tip-atom motif scaffolding are presented in Fig. 33, Tab. 5 and Tab. 6. We do not include Protpardelle in this more stringent evaluation as its performance is not competitive with *La-Proteina* and Protpardelle-1c. From the results, it can be observed that *La-Proteina* yields state-of-the-art performance under this evaluation as well.

## F.5   BASELINE SAMPLING

We sampled Protpardelle (Chu et al., 2024) (limited to indexed scaffolding) we used the option `--type allatom` and generate template pdb files with the motif coordinates as well as template residues for representing the scaffold in order to represent the correct length sampling ranges. For Protpardelle-1c (Lu et al., 2025) (limited to all-atom indexed scaffolding) we used the samples generated by the authors, as they evaluated their model on the same motif scaffolding benchmark, and we processed the samples using our own evaluation pipeline.

Table 4: **Tip-atom motif scaffolding.** "All" indicates total number of successes produced by the model (we produce 200 samples per task), while "Unique" indicates number of unique successes, obtained by clustering all successes as explained in App. F.3. "Indexed" indicates the motif residue indices are provided as input to the model, "Unindexed" indicates that the motif residue indices are not provided as input. "# segments" refers to the number of residue segments in the motif.

| Motif Task | # segments | Protpardelle (indexed) | | *La-Proteina* (indexed) | | *La-Proteina* (unindexed) | |
|---|---|---|---|---|---|---|---|
| | | All | Unique | All | Unique | All | Unique |
| 1YCR_AA | 1 | 0 | 0 | 112 | 54 | 109 | 47 |
| 5TRV_AA_long | 1 | 0 | 0 | 75 | 16 | 66 | 42 |
| 5TRV_AA_med | 1 | 0 | 0 | 41 | 14 | 67 | 33 |
| 6E6R_AA_long | 1 | 0 | 0 | 30 | 12 | 17 | 13 |
| 3IXT_AA | 1 | 0 | 0 | 1 | 1 | 96 | 13 |
| 5TPN_AA | 1 | 0 | 0 | 20 | 9 | 22 | 10 |
| 4ZYP_AA | 1 | 0 | 0 | 6 | 2 | 38 | 8 |
| 5TRV_AA_short | 1 | 0 | 0 | 14 | 3 | 24 | 6 |
| 6E6R_AA_med | 1 | 0 | 0 | 35 | 7 | 14 | 4 |
| 6E6R_AA_short | 1 | 0 | 0 | 16 | 5 | 3 | 2 |
| 5WN9_AA | 1 | 0 | 0 | 0 | 0 | 5 | 2 |
| 7MRX_AA_128 | 1 | 0 | 0 | 44 | 22 | 0 | 0 |
| 7MRX_AA_85 | 1 | 0 | 0 | 72 | 5 | 0 | 0 |
| 7MRX_AA_60 | 1 | 0 | 0 | 1 | 1 | 0 | 0 |
| 4JHW_AA | 2 | 0 | 0 | 0 | 0 | 8 | 6 |
| 6VW1_AA | 2 | 0 | 0 | 28 | 5 | 63 | 6 |
| 1PRW_AA | 2 | 2 | 1 | 45 | 5 | 25 | 3 |
| 5IUS_AA | 2 | 0 | 0 | 1 | 1 | 2 | 2 |
| 2KL8_AA | 2 | 6 | 1 | 189 | 1 | 154 | 1 |
| 1QJG_AA_NAT | 3 | 0 | 0 | 109 | 37 | 85 | 78 |
| 7K4V_AA | 3 | 0 | 0 | 28 | 27 | 68 | 64 |
| 1QJG_AA | 3 | 2 | 1 | 125 | 51 | 61 | 55 |
| 5AOU_AA | 3 | 0 | 0 | 64 | 16 | 43 | 40 |
| 5YUI_AA | 3 | 0 | 0 | 3 | 3 | 10 | 9 |
| 5AOU_QUAD_AA | 4 | 0 | 0 | 95 | 4 | 35 | 29 |
| 1BCF_AA | 4 | 42 | 1 | 152 | 10 | 146 | 10 |

# G  ABLATIONS

## G.1  VAE ABLATIONS

We first ablate multiple choices in the VAE's design: The weight use for the KL term in the ELBO loss from Eq. (3), the architecture type used for the decoder, and building a fully-latent model that encodes $\alpha$-carbon coordinates as well (in contrast to *La-Proteina*, which models $\alpha$-carbon coordinates explicitly).

### G.1.1  KL PENALTY WEIGHT

**KL-weight.** The weight use for the KL term in the ELBO loss from Eq. (3), for which we tested values in $\{10^{-3}, 10^{-4}, 10^{-5}\}$.

### G.1.2  DECODER ARCHITECTURE

**Decoder arch.** The type of architecture used for the decoder, for which we compare the transformer used by all our models evaluated in the main text, against using a feed forward network with 7M parameters. For this we use a weight of $10^{-5}$ for the KL term in the ELBO loss from Eq. (3).

### G.1.3  ENCODING $\alpha$-CARBONS

**CA-enc.** We test encoding the $\alpha$-carbons as well (with a transformer decoder). In this case, the $\alpha$-carbon coordinates are not modeled explicitly, as in *La-Proteina*, but also encoded into the eight-dimensional latent space. This ablation shows the importance of explicitly modeling the $\alpha$-carbon coordinates. For this we use a weight of $10^{-5}$ for the KL term in the ELBO loss from Eq. (3).

Table 5: **All-atom motif scaffolding, evaluation with refolded structures.** "Evaluation using refolded structures" indicates that a success is given by the criteria specified in App. F.4, which compares refolded structures—where the sequence is produced by *La-Proteina* and then folded with ESMFold—against the motif (instead of comparing the generated structure directly, as explained in App. F.3). "All" indicates total number of successes produced by the model (we produce 200 samples per task), while "Unique" indicates number of unique successes, obtained by clustering all successes as explained in App. F.3. "Indexed" indicates the motif residue indices are provided as input to the model, "Unindexed" indicates that the motif residue indices are not provided as input. "# segments" refers to the number of residue segments in the motif.

| | | Evaluation using refolded structures | | | | | |
|---|---|---|---|---|---|---|---|
| Motif Task | # segments | Protpardelle-1c (indexed) | | *La-Proteina* (indexed) | | *La-Proteina* (unindexed) | |
| | | All | Unique | All | Unique | All | Unique |
| 6E6R_AA_long | 1 | 13 | 12 | 84 | 52 | 97 | 72 |
| 6E6R_AA_med | 1 | 18 | 17 | 78 | 47 | 87 | 59 |
| 1YCR_AA | 1 | 56 | 29 | 117 | 37 | 91 | 51 |
| 5TRV_AA_long | 1 | 5 | 5 | 36 | 12 | 57 | 29 |
| 7MRX_AA_128 | 1 | 2 | 2 | 9 | 4 | 42 | 19 |
| 6E6R_AA_short | 1 | 1 | 1 | 26 | 18 | 44 | 22 |
| 5TRV_AA_med | 1 | 1 | 1 | 25 | 8 | 27 | 12 |
| 5TPN_AA | 1 | 8 | 4 | 25 | 4 | 18 | 7 |
| 7MRX_AA_85 | 1 | 13 | 5 | 25 | 7 | 67 | 4 |
| 3IXT_AA | 1 | 3 | 3 | 21 | 3 | 49 | 6 |
| 4ZYP_AA | 1 | 1 | 1 | 4 | 2 | 27 | 3 |
| 5TRV_AA_short | 1 | 0 | 0 | 1 | 1 | 0 | 0 |
| 7MRX_AA_60 | 1 | 7 | 2 | 49 | 2 | 34 | 1 |
| 5WN9_AA | 1 | 0 | 0 | 0 | 0 | 0 | 0 |
| 1PRW_AA | 2 | 162 | 20 | 174 | 18 | 120 | 10 |
| 5IUS_AA | 2 | 0 | 0 | 1 | 1 | 6 | 1 |
| 6VW1_AA | 2 | 11 | 1 | 22 | 1 | 42 | 1 |
| 2KL8_AA | 2 | 74 | 1 | 51 | 1 | 38 | 1 |
| 4JHW_AA | 2 | 0 | 0 | 0 | 0 | 0 | 0 |
| 1QJG_AA_NAT | 3 | 3 | 2 | 23 | 13 | 71 | 50 |
| 7K4V_AA | 3 | 2 | 1 | 33 | 4 | 29 | 29 |
| 5AOU_AA | 3 | 39 | 5 | 157 | 9 | 11 | 11 |
| 1QJG_AA | 3 | 6 | 4 | 29 | 9 | 37 | 23 |
| 5YUI_AA | 3 | 0 | 0 | 2 | 2 | 7 | 3 |
| 5AOU_QUAD_AA | 4 | 12 | 2 | 80 | 7 | 42 | 17 |
| 1BCF_AA | 4 | 195 | 6 | 144 | 7 | 89 | 5 |

### G.1.4 RESULTS

For each VAE variant, we train a dedicated flow matching model using the Foldseek clustered AFDB dataset (filtered to a maximum protein length of 256 residues). We then evaluate the generative performance by measuring all-atom co-designability and diversity on proteins sampled at lengths of $\{50, 100, 150, 200, 250\}$. We use the sampling hyperparameters detailed in App. E for the **KL-weight** and **Dec-arch** VAE variants. However, this setting is not directly applicable to the **CA-enc** model, as it encodes the entire protein, including $\alpha$-carbon coordinates, into its latent variables and does not explicitly model $\alpha$-carbons separately. To ensure a fair comparison and optimize its performance, we conducted a hyperparameter search for the **CA-enc** model. This involved exploring both Langevin scaling functions ("1/t" and "tan" from Eq. (10)) and all three numerical discretization schemes ("exponential", "uniform", "quadratic" from App. E.1), and selected the combination that yielded best results.

The results from this VAE ablation study are shown in Tab. 7, which reports all-atom co-designability and diversity values for each model. The three main conclusions are: First, Lower weights for the KL divergence term in the ELBO objective ($10^{-4}$ and $10^{-5}$) yield better generative performance than a higher weight ($10^{-3}$). Second, replacing the transformer architecture in the decoder by a feed forward network (7M parameters) leads to worse performance. Third, and most critically, explicitly modeling $\alpha$-carbon coordinates, a cornerstone of *La-Proteina*'s design, leads to substantially better results than an approach that encodes the entire protein structure, including $\alpha$-carbon coordinates, into a unified latent space (as in the **CA-enc** model). This last finding is particularly relevant, as it strongly validates *La-Proteina*'s fundamental design choice of treating the $\alpha$-carbon backbone explicitly, rather than relying on a fully latent representation for the whole protein structure.

Table 6: **Tip-atom motif scaffolding, evaluation with refolded structures.** "Evaluation using refolded structures" indicates that a success is given by the criteria specified in App. F.4, which compares refolded structures—where the sequence is produced by *La-Proteina* and then folded with ESMFold—against the motif (instead of comparing the generated structure directly, as explained in App. F.3). "All" indicates total number of successes produced by the model (we produce 200 samples per task), while "Unique" indicates number of unique successes, obtained by clustering all successes as explained in App. F.3. "Indexed" indicates the motif residue indices are provided as input to the model, "Unindexed" indicates that the motif residue indices are not provided as input. "# segments" refers to the number of residue segments in the motif.

| | | **Evaluation using refolded structures** | | | |
|---|---|---|---|---|---|
| **Motif Task** | **# segments** | *La-Proteina* (indexed) | | *La-Proteina* (unindexed) | |
| | | All | Unique | All | Unique |
| 1YCR_AA | 1 | 111 | 54 | 92 | 45 |
| 5TRV_AA_long | 1 | 46 | 13 | 29 | 23 |
| 5TRV_AA_med | 1 | 19 | 10 | 20 | 12 |
| 6E6R_AA_long | 1 | 28 | 12 | 8 | 7 |
| 3IXT_AA | 1 | 1 | 1 | 94 | 11 |
| 5TPN_AA | 1 | 6 | 5 | 16 | 7 |
| 4ZYP_AA | 1 | 6 | 2 | 32 | 7 |
| 5TRV_AA_short | 1 | 2 | 1 | 0 | 0 |
| 6E6R_AA_med | 1 | 30 | 5 | 2 | 2 |
| 6E6R_AA_short | 1 | 16 | 5 | 3 | 2 |
| 5WN9_AA | 1 | 0 | 0 | 5 | 2 |
| 7MRX_AA_128 | 1 | 18 | 13 | 0 | 0 |
| 7MRX_AA_85 | 1 | 39 | 3 | 0 | 0 |
| 7MRX_AA_60 | 1 | 0 | 0 | 0 | 0 |
| 4JHW_AA | 2 | 0 | 0 | 0 | 0 |
| 6VW1_AA | 2 | 19 | 2 | 61 | 4 |
| 1PRW_AA | 2 | 45 | 5 | 25 | 3 |
| 5IUS_AA | 2 | 1 | 1 | 0 | 0 |
| 2KL8_AA | 2 | 77 | 1 | 6 | 1 |
| 1QJG_AA_NAT | 3 | 38 | 18 | 18 | 18 |
| 7K4V_AA | 3 | 12 | 12 | 15 | 15 |
| 1QJG_AA | 3 | 67 | 18 | 17 | 15 |
| 5AOU_AA | 3 | 52 | 8 | 18 | 18 |
| 5YUI_AA | 3 | 1 | 1 | 3 | 3 |
| 5AOU_QUAD_AA | 4 | 29 | 2 | 7 | 7 |
| 1BCF_AA | 4 | 145 | 10 | 132 | 10 |

Table 7: Ablation study for the VAE design, including different weights for the KL penalty term, a variant of the VAE which uses a feed forward network instead of the transformer in the decoder, and a variant that also encodes the $\alpha$-carbon coordinates (that is, in this specific case, the flow matching model operates entirely in the latent space, without explicitly modeling $\alpha$-carbon coordinates, which are also captured by the latent variables). For all VAEs we train a flow matching model on proteins of length up to 256 residues and report co-designability and diversity metrics. All models were evaluated for multiple noise scaling parameters, and we selected the one that led to the best performance (not reported for simplicity).

| VAE Type | KL weight | Co-designability (%) ↑ | Diversity (# clusters) ↑ | | |
|---|---|---|---|---|---|
| | | All-atom | Str | Seq | Seq+Str |
| Transformer (enc), Transformer (dec) | $10^{-3}$ | 65.2 | 154 | 163 | 248 |
| Transformer (enc), Transformer (dec) | $10^{-4}$ | 83.8 | 246 | 317 | 374 |
| Transformer (enc), Transformer (dec) | $10^{-5}$ | 82.4 | 214 | 295 | 339 |
| Transformer (enc), Feed Forward (dec) | $10^{-5}$ | 58.0 | 151 | 242 | 233 |
| Transformer (enc), Transformer (dec), encode $\alpha$-carbons | $10^{-5}$ | 21.2 | 51 | 105 | 91 |

## G.2 FLOW MATCHING SAMPLING HYPERPARAMETERS

As explained in Sec. 3.4 and App. E, sampling *La-Proteina* requires selecting the discretization scheme used for the $\alpha$-carbon coordinates $\mathbf{x}_{C_\alpha}$ and latent variables $\mathbf{z}$, and the functions to scale the

Table 8: Ablation study over discretization scheme and Langevin term scaling functions for *La-Proteina* sampling. The table includes combinations that yield an all-atom co-designability of at least 0.5. Details for the different discretization schemes and Langevin scaling functions are given in App. E. The diversity metric is computed over the subset of all-atom co-designable samples.

| Method | Discretization | | Langevin scaling | | Noise scaling | | Co-designability (%) ↑ | Diversity (# clusters) ↑ | | |
|---|---|---|---|---|---|---|---|---|---|---|
| | $\alpha$-carbon | Latent $\mathbf{z}$ | $\beta_x(t_x)$ | $\beta_z(t_z)$ | $\eta_x$ | $\eta_z$ | All-atom | Str | Seq | Seq+Str |
| *La-Proteina* | exp. | quad. | 1/t | tan | 0.1 | 0.1 | 68.4 | 206 | 216 | 310 |
| *La-Proteina* | exp. | quad. | 1/t | tan | 0.2 | 0.1 | 60.6 | 198 | 197 | 261 |
| *La-Proteina* | exp. | quad. | 1/t | tan | 0.3 | 0.1 | 53.8 | 180 | 189 | 249 |
| *La-Proteina* | exp. | quad. | 1/t | 1/t | 0.1 | 0.1 | 59.2 | 164 | 198 | 247 |
| *La-Proteina* | exp. | quad. | 1/t | 1/t | 0.1 | 0.2 | 57.0 | 163 | 189 | 253 |
| *La-Proteina* | exp. | quad. | 1/t | 1/t | 0.1 | 0.3 | 53.4 | 190 | 191 | 245 |
| *La-Proteina* | exp. | unif. | 1/t | 1/t | 0.1 | 0.1 | 50.6 | 194 | 189 | 226 |
| *La-Proteina* | exp. | unif. | 1/t | tan | 0.1 | 0.1 | 54.0 | 210 | 197 | 247 |
| *La-Proteina* | exp. | unif. | 1/t | tan | 0.2 | 0.1 | 52.4 | 208 | 185 | 246 |
| *La-Proteina* | exp. | quad. | tan | 1/t | 0.1 | 0.1 | 57.0 | 161 | 212 | 243 |
| *La-Proteina* | exp. | quad. | tan | 1/t | 0.1 | 0.2 | 53.6 | 171 | 203 | 244 |
| *La-Proteina* | exp. | quad. | tan | tan | 0.1 | 0.1 | 57.4 | 168 | 217 | 251 |
| *La-Proteina* | exp. | quad. | tan | tan | 0.1 | 0.2 | 55.4 | 183 | 216 | 252 |

Langevin term in the SDE from Eq. (7). As a brief reminder, App. E introduced three discretization schemes, "exponential", "quadratic" and "uniform"; and also two scaling functions for the Langevin term in the SDE, the "1/t" and "tan", shown in Eq. (10). While our primary *La-Proteina* configuration (evaluated in Tab. 1) uses a specific pairing (namely, "exponential" discretization with "1/t" scaling for the $\alpha$-carbon coordinates, and "quadratic" discretization with "tan" scaling for the latent variables), alternative combinations are viable. To systematically assess how these choices affect performance, we conducted an ablation study by sampling a specific variant of *La-Proteina* (the model from Sec. 4 without triangular multiplicative layers) with all possible combinations of these schemes and functions for generating both the $\alpha$-carbon coordinates $\mathbf{x}_{C_\alpha}$ and the latent variables $\mathbf{z}$.

The outcomes of this ablation are presented in Tab. 8, which includes hyperparameter combinations that yield an all-atom co-designability of at least 0.5. A clear pattern emerges from these results: only sampling configurations that generate $\alpha$-carbon coordinates at an effectively faster rate than the latent variables surpass the 0.5 all-atom co-designability threshold. More specifically, every successful combination listed employs the "exponential" discretization scheme for $\mathbf{x}_{C_\alpha}$, using either the "quadratic" or "uniform" scheme for $\mathbf{z}$. This implies that other pairings, such as applying "quadratic" or "uniform" schedules for $\mathbf{x}_{C_\alpha}$, or the "exponential" schedule for $\mathbf{z}$, did not yield competitive co-designability values. While the choice of Langevin scaling function also influences performance, its impact was observed to be less pronounced than that of the discretization scheme.

### G.3 MAIN CONCLUSIONS FROM ABLATION STUDIES

The primary conclusion from our ablation studies is that achieving strong performance critically depends on two key factors: first, the explicit modeling of $\alpha$-carbon coordinates, and second, generating these coordinates at an effectively faster rate than the latent variables (which encapsulate all remaining atomic and sequence details).

## H ARCHITECTURES

The three neural networks used in *La-Proteina*, the encoder, decoder, and denoiser, rely on the same core architecture based on transformers with pair-biased attention mechanisms (Jumper et al., 2021; Abramson et al., 2024). Our implementation closely follows Geffner et al. (2025), to which we refer for comprehensive details. This architecture processes inputs into two primary tensors: a sequence representation of shape $[L, C_{\text{seq}}]$, which encodes per-residue features (e.g., atomic coordinates, residue type, etc.), and a pair representation of shape $[L, L, C_{\text{pair}}]$, which encodes features between residue pairs (e.g., relative sequence separation, inter-residue distances, etc.). The sequence representation is iteratively updated through the transformer blocks, while the pair representation provides biases to the attention logits via a learned linear projection within each block, effectively incorporating relational information (Jumper et al., 2021). As aforementioned, we

explore two variants for the denoiser network. One that keeps the pair representation fixed throughout the architecture, and one where we use triangular multiplicative update layers to update the pair representation, including one such layer every two transformer blocks (Jumper et al., 2021). While these updates have shown performance gains in complex structural biology tasks (Lin et al., 2024; Jumper et al., 2021; Abramson et al., 2024), they also add considerable computational expense. Most *La-Proteina* models we evaluate do not use triangular update layers and yield state-of-the-art performance. In practice, we use $C_{\text{seq}} = 768$ and $C_{\text{pair}} = 256$, 14 transformer layers for the encoder and decoder, and 16 layers for the denoiser, yielding a total of 130M and 160M parameters, respectively.

The primary distinction between our three networks lies in the specific inputs they receive, how these inputs are featurized to construct the initial sequence and pair representations, and the target outputs they predict. The feature construction follows closely McPartlon & Xu (2023). The sequence representation captures features for each independent residue (e.g. atomic coordinates), while the pair representation captures features for residue pairs (e.g. relative distance and sequence separation).

**Encoder.**   The encoder parameterizes the Gaussian distribution $q_\psi(\mathbf{z} \,|\, \mathbf{x}_{\neg C_\alpha}, \mathbf{s}, \mathbf{x}_{C_\alpha})$, mapping the inputs $(\mathbf{x}_{C_\alpha}, \mathbf{x}_{\neg C_\alpha}, \mathbf{s})$ to the distribution's mean $\mu \in \mathbb{R}^{L \times 8}$ and log-scale $\log \sigma \in \mathbb{R}^{L \times 8}$. The input features used by the encoder to construct the initial sequence representation are: (i) Raw absolute Atom37 coordinates; (ii) Raw Atom37 coordinates, relative to the $\alpha$-carbons; (iii) Residue type, as a one-hot vector; (iv) Side chain angles, consisting of at most four angles (depends on residue type), which are binned into 20 bins between $-\pi$ and $\pi$; (v) Backbone torsion angles, which are binned into 20 bins between $-\pi$ and $\pi$. The input features to construct the initial pair representation are: (i) Relative sequence separation, as one-hot vectors, capped at $\pm 64$; (ii) Relative orientations between pairs of residues Yang et al. (2020), which are binned into 20 bins between $-\pi$ and $\pi$; (iii) Pairwise distances between $\alpha$-carbons and all other backbone atoms, binned into 20 bins between 1Å and 20Å. The initial representations are then processed through 12 transformer blocks. The final sequence representation is fed through a linear layer to produce $\mu$ and $\log \sigma$, and the latent variables are obtained as $\mathbf{z} \sim \mathcal{N}(\mu, \sigma^2) \in \mathbb{R}^{L \times 8}$.

**Decoder.**   The decoder parameterizes the Categorical distribution $p_\phi(\mathbf{s} \,|\, \mathbf{z}, \mathbf{x}_{C_\alpha})$ and the Gaussian distribution $p_\phi(\mathbf{x}_{\neg C_\alpha} \,|\, \mathbf{z}, \mathbf{x}_{C_\alpha})$, mapping the inputs $(\mathbf{z}, \mathbf{x}_{C_\alpha})$ to the logits of the Categorical, $\ell \in \mathbb{R}^{L \times 20}$, and the mean of the Gaussian, $\mu_{\text{dec}} \in \mathbb{R}^{L \times 36 \times 3}$ (variance fixed to one). The input features used by the decoder to construct the initial sequence representation are: (i) Raw $\alpha$-carbon coordinates $\mathbf{x}_{C_\alpha}$; (ii) Raw latent variables $\mathbf{z}$. The input features to construct the initial pair representation are: (i) Relative sequence separation, as one-hot vectors, capped at $\pm 64$; (ii) Pairwise distances between $\alpha$-carbons, binned into 30 bins between 1Å and 30Å. The initial representations are then processed through 12 transformer blocks. The final sequence representation is fed through a linear layer to produce $\ell$ and $\mu_{\text{dec}}$.

**Denoiser network.**   The denoiser network maps time-dependent inputs, the interpolation times $t_x, t_z$ and corrupted coordinates $\mathbf{x}_{C_\alpha}^{t_x}$ and latents $\mathbf{z}^{t_z}$, to velocity fields $\mathbf{v}_\phi^x \in \mathbb{R}^{L \times 3}$ and $\mathbf{v}_\phi^z \in \mathbb{R}^{L \times 8}$, used to sample $p_\phi(\mathbf{x}_{C_\alpha}, \mathbf{z})$. The corrupted inputs are featurized into the initial sequence and pair representations. More specifically, the initial sequence representation uses: (i) Raw corrupted $\alpha$-carbon coordinates $\mathbf{x}_{C_\alpha}^{t_x}$; (ii) Raw corrupted latent variables $\mathbf{z}^{t_z}$. The input features to construct the initial pair representation are: (i) Relative sequence separation, as one-hot vectors, capped at $\pm 64$; (ii) Pairwise distances between corrupted $\alpha$-carbon coordinates, binned into binned into 30 bins between 1Å and 30Å. The initial representations are then processed through 14 transformer blocks. The final sequence representation is fed through a linear layer to produce $\mathbf{v}_\phi^z$ and $\mathbf{v}_\phi^x$. In contrast to the encoder and decoder architecture, the denoiser network also conditions on the interpolation times $t_x$ and $t_z$. This is done directly within its transformer blocks using adaptive layer normalization and output scaling techniques (Peebles & Xie, 2023).

## I   MODEL PARAMETERS, SAMPLING SPEED AND MEMORY CONSUMPTION

Table 9: Sampling time [seconds] for different methods at batch size 1 (top) and maximum batch size (bottom) across varying protein lengths on an A100-80GB GPU. For PLAID and *La-Proteina*, the first parameter count is the diffusion model and the second one is the decoder.

| Method | # Params | Steps | 100 | 200 | 300 | 400 | 500 | 600 | 700 | 800 |
|---|---|---|---|---|---|---|---|---|---|---|
| *Batch size: 1* | | | | | | | | | | |
| P(all-atom) | 17.7M | 200 | 32.9 | 62.1 | 106.1 | OOM | OOM | OOM | OOM | OOM |
| ProteinGenerator | 59.8M | 100 | 197.8 | 239.6 | 428.6 | 642.8 | 981.0 | 1365.4 | 1915.0 | 2690.4 |
| Protpardelle | 25.1M | 200 | 2.3 | 3.2 | 4.3 | 5.2 | 6.1 | 7.3 | 8.4 | 9.5 |
| PLAID | 100M + 3.5B | 500 | 6.2 | 8.0 | 11.6 | 18.1 | 25.4 | 38.1 | 54.4 | 77.6 |
| *La-Proteina* | 158M + 128M | 400 | 2.94 | 3.00 | 3.67 | 4.75 | 6.33 | 8.45 | 10.63 | 13.52 |
| *La-Proteina* $_{tri}$ | 167M + 128M | 400 | 4.22 | 9.72 | 20.78 | 34.85 | 59.95 | 100.00 | 153.14 | 196.46 |
| *Maximum batch size (runtimes normalised to be per 1 sample; batch size values in Table 10)* | | | | | | | | | | |
| PLAID | 100M + 3.5B | 500 | 0.78 | 3.16 | 7.29 | 15.00 | 22.55 | 36.75 | 54.33 | 78.17 |
| *La-Proteina* | 158M + 128M | 400 | 0.34 | 0.99 | 2.04 | 3.34 | 5.01 | 7.01 | 9.46 | 12.31 |
| *La-Proteina* $_{tri}$ | 167M + 128M | 400 | 1.72 | 6.31 | 16.29 | 25.74 | 42.45 | 59.28 | 77.38 | 106.69 |

Table 10: Maximum batch size for samples of varying length (the numbers in the top row indicate protein backbone chain length) on an A100-80GB GPU.

| Method | # Model parameters | Inference steps | 100 | 200 | 300 | 400 | 500 | 600 | 700 | 800 |
|---|---|---|---|---|---|---|---|---|---|---|
| PLAID | 100M + 3.5B | 500 | 792 | 154 | 73 | 35 | 20 | 12 | 9 | 6 |
| *La-Proteina* | 158M + 128M | 400 | 422 | 118 | 49 | 29 | 17 | 13 | 8 | 7 |
| *La-Proteina* $_{tri}$ | 167M + 128M | 400 | 530 | 150 | 60 | 35 | 22 | 17 | 11 | 10 |

To evaluate both model complexity (through parameter counts) and its operational consequences for memory usage and generation speed, we perform three complementary experiments following Geffner et al. (2025):

1. **Single-sequence inference latency**: Measurement of per-sample generation time using batch size 1 on an NVIDIA A100-80GB. Results appear in Table 9 upper part.

2. **Batch-optimized throughput analysis**: Measurement of generation times at maximum batch capacities, with computational efficiency quantified through time-per-sequence normalization. Executed on A100-80GB GPUs as documented in Table 9 lower part.

3. **Memory efficiency assessment**: Determination of maximum viable batch sizes without exceeding memory limits, conducted on an NVIDIA A100-80GB GPU to establish practical scalability thresholds. See Table 10 for detailed comparisons.

All referenced tables include parameter counts for cross-model comparison.

Our implementation capitalizes on the transformer architecture's hardware compatibility through PyTorch's compilation framework (Ansel et al., 2024), which accelerates both training and inference phases. Reported inference metrics for *La-Proteina* as well as other models leveraging compilation such as P(all-atom) reflect performance optimizations achieved via model compilation and report timings excluding compilation overhead at the beginning since it becomes negligible for large-scale inference which is mostly of interested in the protein design setting.

We can see that *La-Proteina* is fast despite the high parameter count; the model without triangle multiplication layers is the fastest togther with Protpardelle. The model with triangle multiplication layers is slower, but still faster than P(all-atom) and Protein Generator, as well as faster than PLAID at short lengths.

Since only *La-Proteina* and PLAID support batched inference, the difference becomes stark there: at maximum batch size *La-Proteina* can generate hundreds of proteins in one batch, resulting in inference times of below a second for short proteins. Interestingly, after compilation of these models the models with triangle multiplication layers is able to fit higher batch sizes than the one without triangle multiplication layers, probably as an artifact of the compilation process.

One also sees the *La-Proteina* benefits a lot more from batched inference speed-ups than PLAID. This is mostly due to the *La-Proteina* decoder being fairly lightweigth and fast, with the majority

Table 11: Novelty results for Foldseek 10.941cd33. Should be contrasted to results in Tab. 1 which uses Foldseek version 9.427df8a (included in the table for reference).

| Method | Novelty (Foldseek 10.941cd33) ↓ | | Novelty (Foldseek 9.427df8a, reference from Tab. 1) ↓ | |
|---|---|---|---|---|
| | PDB | AFDB | PDB | AFDB |
| P(all-atom) | 0.79 | 0.86 | 0.72 | 0.81 |
| Protpardelle-1c | 0.81 | 0.86 | 0.78 | 0.83 |
| APM | 0.88 | 0.92 | 0.84 | 0.89 |
| PLAID | 0.92 | 0.94 | 0.89 | 0.92 |
| ProteinGenerator | 0.87 | 0.92 | 0.83 | 0.89 |
| Protpardelle | 0.82 | 0.85 | 0.79 | 0.82 |
| *La-Proteina* $(\eta_x, \eta_z) = (0.1, 0.1)$ | 0.81 | 0.86 | 0.75 | 0.82 |
| *La-Proteina* $(\eta_x, \eta_z) = (0.2, 0.1)$ | 0.81 | 0.87 | 0.76 | 0.83 |
| *La-Proteina* $(\eta_x, \eta_z) = (0.3, 0.1)$ | 0.82 | 0.87 | 0.77 | 0.86 |
| *La-Proteina* tri $(\eta_x, \eta_z) = (0.1, 0.1)$ | 0.86 | 0.9 | 0.82 | 0.86 |
| *La-Proteina* tri $(\eta_x, \eta_z) = (0.3, 0.1)$ | 0.83 | 0.88 | 0.79 | 0.85 |

of time spent during the diffusion process, while in PLAID the ESMFold-3B decoder is the major bottleneck.

## J    ADDITIONAL RESULTS

### J.1    NOVELTY RESULTS WITH NEWER FOLDSEEK VERSION

Tab. 1 in the main paper reports novelty results using Foldseek version 9.427df8a. We note that newer Foldseek versions return different values for the exact same command and samples. Tab. 11 shows novelty results for all methods using Foldseek version 10.941cd33. The table only reports novelty values, as all other metrics are exactly the same as in Tab. 1.

### J.2    UNCONDITIONAL EVALUATION ON SHORTER PROTEINS

To complement our main evaluation from Tab. 1, we performed an additional comparative analysis using subsets of shorter proteins. This accounts for the diverse training distribution of various baseline models, some of which were trained on proteins shorter than those used to train *La-Proteina*. We assessed all models across three scenarios: (A) 100 proteins of length 100; (B) 200 proteins, with 100 samples each for lengths 100 and 200; and (C) 300 proteins, comprising 100 samples each for lengths 100, 200, and 300. The resulting all-atom co-designability and diversity metrics for this analysis are summarized in Tab. 12, where it can be observed that *La-Proteina* yields state-of-the-art performance across all three scenarios.

### J.3    PLDDT VALUES

Violin plots showing the distribution of confidence scores (pLDDT) for protein structures generated by *La-Proteina* and baselines are shown in Fig. 34. The samples used for those plots are the same as the ones used for the results in Tab. 1, filtered by all-atom co-designable, that is, we only include samples with all-atom RMSD between the model produced structure and the one obtained by folding the model produced sequence with ESMFold $< 2\text{Å}$. The results show that all variants of *La-Proteina*, different temperatures and with/without triangular update layers, consistently achieve the highest median and interquartile range (IQR) pLDDT values.

### J.4    RESIDUE TYPE DISTRIBUTION

Fig. 35 shows the distribution of amino acids generated by each of the comparative methods against the UniProt reference distribution. The samples used for this figure are identical to those used for Tab. 1, filtered by all-atom co-designability. As it can be observed from the figure, all methods over- and under-represent certain residues when compared against the UniProt reference distribution. Notably, as demonstrated in Fig. 36, *La-Proteina* produces a distribution over amino acid types very similar to the ones produced by ProteinMPNN. We view this as a positive outcome, as sequences generated by ProteinMPNN are generally acknowledged to possess highly desirable properties.

Table 12: Results for unconditional generation for proteins of different lengths. Table (A) shows results for 100 proteins of length 100; table (B) for proteins of lengths in {100, 200}, with 100 samples per length; table (C) for proteins of lengths in {100, 200, 300}, with 100 samples per length. $\eta_x$ and $\eta_z$ denote the noise scaling factors during generation (Eq. (7)). Best scores for each table **bold**, second best underlined.

| Method | All-atom co-designability (%) ↑ | Diversity (# clusters) ↑ | | |
| --- | --- | --- | --- | --- |
| | | Structure | Sequence | Sequence+Structure |
| (A) 100 proteins for each length in {100} | | | | |
| P(all-atom) | 88.0 | 41 | 78 | 70 |
| Protpardelle-1c | 76.0 | 13 | 67 | 33 |
| APM | 56.0 | 15 | 48 | 33 |
| PLAID | 22.0 | 14 | 18 | 15 |
| ProteinGenerator | 45.0 | 13 | 27 | 22 |
| Protpardelle | 29.0 | 8 | 26 | 17 |
| *La-Proteina* $(\eta_x, \eta_z) = (0.1, 0.1)$ | **96.0** | 43 | **89** | **82** |
| *La-Proteina* $(\eta_x, \eta_z) = (0.2, 0.1)$ | 90.0 | 54 | 78 | 78 |
| *La-Proteina* $(\eta_x, \eta_z) = (0.3, 0.1)$ | 88.0 | **55** | 80 | 80 |
| (B) 100 proteins for each length in {100, 200} | | | | |
| P(all-atom) | 80.5 | **111** | 132 | 143 |
| Protpardelle-1c | 70.0 | 14 | 115 | 35 |
| APM | 38.0 | 24 | 48 | 56 |
| PLAID | 15.5 | 18 | 24 | 20 |
| ProteinGenerator | 26.0 | 14 | 26 | 26 |
| Protpardelle | 17.0 | 9 | 31 | 19 |
| *La-Proteina* $(\eta_x, \eta_z) = (0.1, 0.1)$ | **89.5** | 93 | **147** | **151** |
| *La-Proteina* $(\eta_x, \eta_z) = (0.2, 0.1)$ | 84.0 | 104 | 145 | 143 |
| *La-Proteina* $(\eta_x, \eta_z) = (0.3, 0.1)$ | 80.5 | **111** | 146 | 149 |
| (C) 100 proteins for each length in {100, 200,300} | | | | |
| P(all-atom) | 60.7 | 131 | 147 | 164 |
| Protpardelle-1c | 58.3 | 14 | 133 | 35 |
| APM | 30.0 | 30 | 54 | 62 |
| PLAID | 15.3 | 23 | 32 | 25 |
| ProteinGenerator | 17.7 | 14 | 26 | 26 |
| Protpardelle | 14.7 | 10 | 36 | 20 |
| *La-Proteina* $(\eta_x, \eta_z) = (0.1, 0.1)$ | **85.3** | **150** | **192** | **217** |
| *La-Proteina* $(\eta_x, \eta_z) = (0.2, 0.1)$ | 77.7 | 144 | 181 | 198 |
| *La-Proteina* $(\eta_x, \eta_z) = (0.3, 0.1)$ | 69.7 | 139 | 173 | 191 |

The sequence results discussed above were obtained by our model using low temperature sampling, specifically with parameters $(\eta_x, \eta_z) = (0.1, 0.1)$. To quantitatively assess the effect of temperature parameters on the resulting sequence distribution, we generated 1000 samples of length 100 using various temperatures. Results from this analysis are shown Fig. 37, where it can be observed that increasing the temperature to 0.5 and 0.9 (for both backbone and latent variables jointly) leads to distributions that get increasingly closer to the UniProt reference. However, this increase in temperature comes at the cost of performance in other metrics, manifesting as reduced all-atom co-designability. For instance, across the samples produced for this specific temperature analysis, the all-atom co-designability decreases from 90% for $(\eta_x, \eta_z) = (0.1, 0.1)$, to 54% for $(\eta_x, \eta_z) = (0.5, 0.5)$, and finally to 11% for $(\eta_x, \eta_z) = (0.9, 0.9)$.

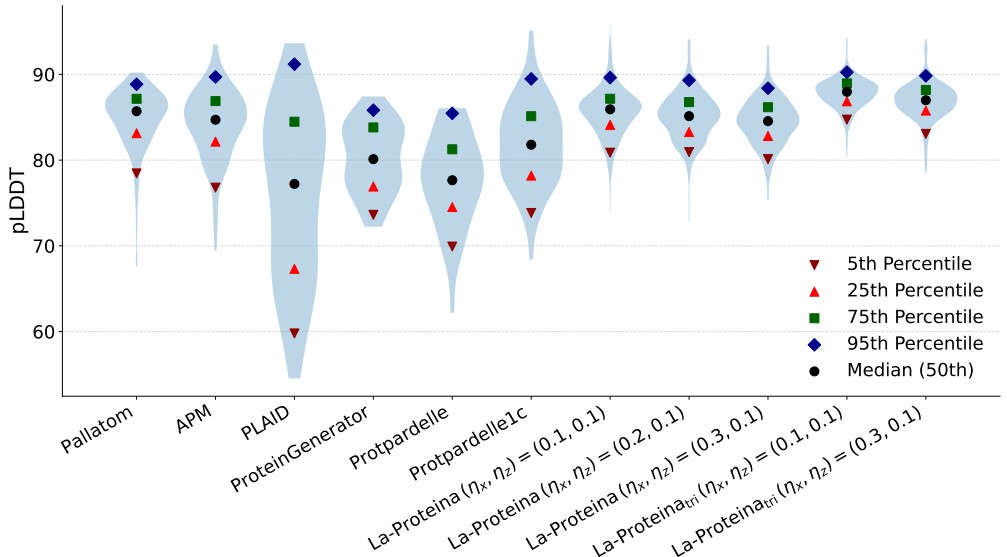

Figure 34: *La-Proteina* **samples with higher pLDDT than existing all-atom generation baselines.** Samples included in the analysis are the same as those use to compute the metrics in Tab. 1. As for most metrics (App. D) we filter samples to keep only all-atom co-designable ones.

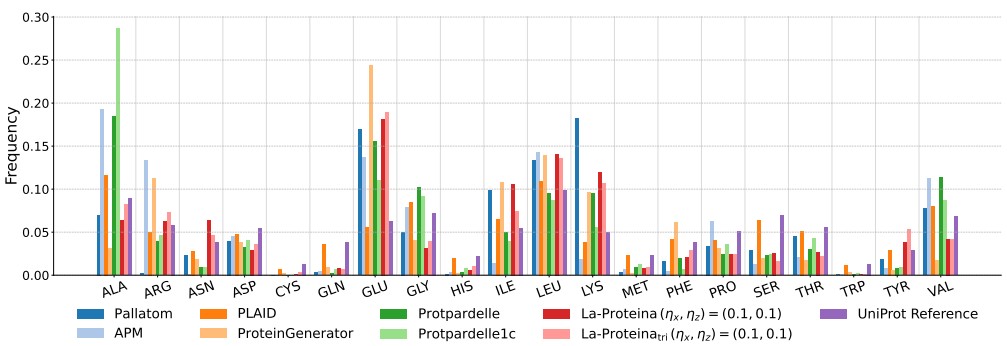

Figure 35: Distribution of amino-acid frequencies of co-designable samples by different methods.

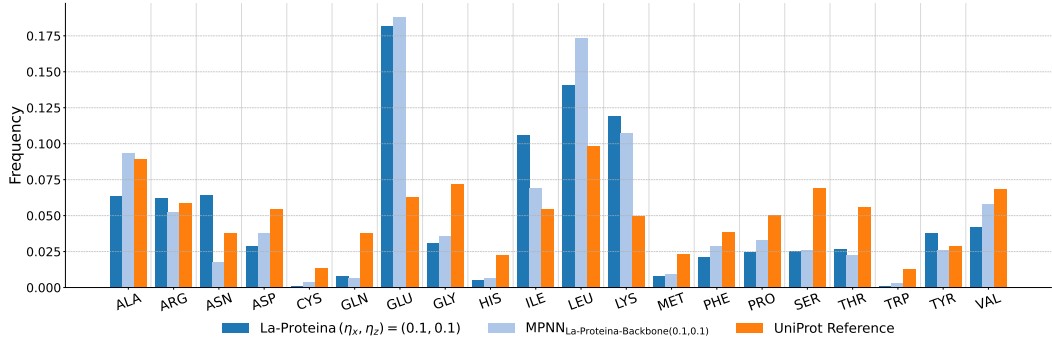

Figure 36: Distribution of amino-acid frequencies of *La-Proteina* for different temperatures. Plots were obtained by filtering for co-designable samples.

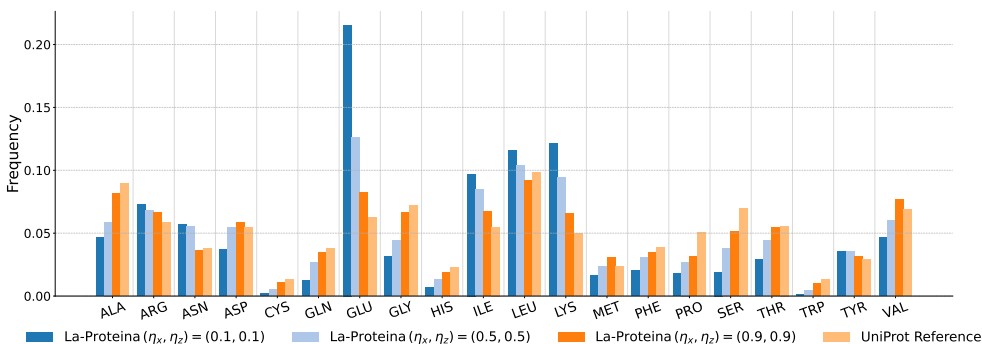

Figure 37: Distribution of amino-acid frequencies of *La-Proteina* for different temperatures. Plots were obtained by filtering for co-designable samples.

## K  DECLARATION OF LARGE LANGUAGE MODELS USAGE IN THIS WORK

Large Language Models (LLMs) were used as a writing aid to enhance clarity and readability during the preparation of this manuscript. The use of these tools was strictly limited to grammatical correction and stylistic refinements. All intellectual content, analyses, and arguments in the paper did not rely on the use of LLMs in any way.

