# OpenReview forum: "La-Proteina: Atomistic Protein Generation via Partially Latent Flow Matching"
_ICLR.cc/2026/Conference — ICLR 2026 Poster_

### Official Review · Reviewer_gg8b · 2025-10-28

**Soundness:** 3
**Presentation:** 3
**Contribution:** 3
**Rating:** 6
**Confidence:** 3

**Summary:**

This paper introduces **La-Proteina**, a generative model for the joint design of full-atom protein structures and their sequences.

The key innovation is a **partially latent representation**:
*   The protein backbone is modeled explicitly in 3D space.
*   The sequence and side-chain atoms are captured in fixed-size latent variables.

Using flow matching on this representation, La-Proteina models the joint distribution of sequence and structure. The authors report that their model achieves state-of-the-art performance on co-designability and structural validity, surpasses previous methods in motif scaffolding, and uniquely scales to generate proteins up to 800 residues long.

**Strengths:**

1.  **Innovative Representation for Full-Atom Generation:** The paper proposes a novel partially latent representation for full-atom protein design. By modeling the backbone explicitly while encoding sequence and side-chain information into a fixed-size latent space, the method effectively addresses the long-standing challenge of variable side-chain dimensionality. The application of flow matching to this hybrid representation is a sound and well-motivated approach for modeling the joint distribution.

2.  **Strong Empirical Performance on Key Benchmarks:** The model demonstrates strong performance across a range of challenging tasks. The results indicate state-of-the-art capabilities in both unconditional generation and various motif scaffolding scenarios. A particularly notable result is the model's ability to generate viable proteins up to 800 residues, demonstrating a level of scalability and robustness that surpasses many existing baselines.

3.  **Thorough and Informative Ablation Studies:** The inclusion of extensive ablation studies is a commendable aspect of this work. The authors systematically evaluate the impact of different design choices related to the model architecture, training process, and sampling strategy. This analysis not only supports the paper's conclusions but also provides useful insights for researchers in the field.

**Weaknesses:**

1.  **Lack of Analysis on Generated Sequence Properties:** For a model that jointly generates full-atom structures and sequences, a notable omission is the characterization of the generated sequence distribution. Prior work in both inverse folding and all-atom generation has shown that models can exhibit significant biases, often over-representing certain amino acids (e.g., Alanine, Glutamate) compared to natural distributions. It is unclear if La-Proteina suffers from a similar sequence bias. An analysis of the amino acid frequencies and other sequence-level properties would be crucial for a complete evaluation of the model's capabilities.

2.  **Potential Confounding Factors in Comparative Evaluation:** The experimental setup in *Table 1* may not provide a fully controlled comparison. La-Proteina was trained on proteins up to 500/800 residues, whereas the baselines were trained on significantly shorter sequences (e.g., APM < 384, Pall-Atom < 128). Evaluating all models on lengths between 100-500 residues means that La-Proteina is largely operating in-distribution, while the baselines are forced to generalize out-of-distribution. This introduces a significant confounding variable (the training data scale and distribution), making it difficult to isolate the true performance benefits of the proposed latent diffusion approach itself. A more compelling comparison would involve evaluating La-Proteina when trained on a dataset of a similar scale to the common baselines.

3.  **Incompleteness in Key Evaluation Metrics:**  While this paper relies on an RMSD threshold for success, other models like APM and Pall-Atom incorporate an additional pLDDT score criterion to ensure that the refolded structures are not only accurate but also predicted with high confidence. The omission of pLDDT analysis is a concern. Could the authors report the distribution of pLDDT scores for the successfully refolded designs?

**Questions:**

1.  **On the Choice of Training Data (AFDB vs. PDB):** The model was trained exclusively on the AlphaFold Database (AFDB), which consists of predicted structures. Could the authors elaborate on the rationale for not incorporating high-resolution experimental structures from the Protein Data Bank (PDB), which are often considered the gold standard by the structural biology community? What is the authors' perspective on the potential impact of using computationally "distilled" data from AFDB versus empirical data from PDB on the model's learned distribution and its generalization to real-world design tasks?

2.  **On the Performance of a Fully Latent Representation:** It is a very interesting result from the ablation study that a fully latent representation (the CA-enc. variant) leads to a significant degradation in performance. Do the authors have a deeper analysis or intuition for this phenomenon? In the generation trajectories of other state-of-the-art models (e.g., Pall-Atom, AlphaFold 3), it is often observed that the backbone structure emerges from the noise before the side-chains are fully resolved, even when the model architecture treats them jointly. Does the failure of the fully latent variant in this work suggest a fundamental difficulty in coupling the degrees of freedom for both backbone and side-chains into a single, unified latent space?

3.  **On Modeling Long-Range Side-Chain Interactions:** *Figure 8* suggests that the VAE's latent representation for side-chains is predominantly local. Given that long-range interactions between side-chains are critical for protein folding, stability, and function, how does the proposed partial-latent modeling framework ensure these non-local dependencies are captured? Unlike methods that explicitly model all atoms in a global context, it is not immediately clear how this is achieved here. Are there specific case studies or quantitative metrics that can demonstrate the model's ability to generate plausible long-range side-chain interaction networks?

---

> ### Author Response · Authors · 2025-11-21
> **Answer (part 1)**
>
> We are grateful for the reviewer's careful reading of the manuscript, for noting the innovativeness of our approach, our strong performance and the thoroughness of the ablation studies. We also truly appreciate all the good points included in the review about ways to improve our empirical evaluation.
>
> **[Lack of Analysis on Generated Sequence ...]**
>
> We have addressed this in the updated manuscript by analyzing the amino acid type distribution for La-Proteina and all baselines, comparing them against the Uniprot reference distribution (Figures 35, 36 and 37 in the updated manuscript). As noted by the reviewer, all methods exhibit some sequence bias, over- or under-representing specific amino acids with respect to the Uniprot reference distribution. La-Proteina allows the user to have explicit control over these biases via the temperature parameter: when sampling the model at full temperature, the generated sequences closely resemble the UniProt distribution (see Figure 37). These sequences have more diverse sequence and structural features, similar to natural proteins, and therefore also have lower co-designability values, comparable to the training dataset. Specifically, as mentioned in Appendix J.3 in the updated manuscript, increasing the temperature from 0.1 to 0.9 reduces all-atom co-designability from 90% to 11%. This effect is common across the field, as low-temperature sampling is widely used and is known to bias the distribution being sampled and have a strong effect on (co-)designability.
>
> When lowering the temperature, La-Proteina smoothly shifts the sequence distribution until it reaches a distribution very similar to the MPNN distribution, which is known to be overrepresented in charged amino acids like glutamate and whose sequences generally have favourable expression and solubility profiles in the lab. With this knob to turn, users have full control over the kind of sequences to generate.
>
> Critically, La-Proteina shows none of the pathologies that other baseline models like APM or Protpardelle show, which include elevated alanine content, indicating that the generated sequences do not have biases that hurt their utility.
>
> **[The experimental setup in Table 1 may not provide a fully controlled comparison. La-Proteina was trained on proteins up to 500 residues, whereas the baselines were trained on significantly shorter sequences ... some baselines operate "out of distribution" in the evaluation ...]**
>
> This is a fair point regarding the training distributions. We appreciate the reviewer's scrutiny on distinguishing between benefits derived from data scale versus the method itself.
>
> However, we emphasize that La-Proteina's ability to train on longer proteins is a direct consequence of its scalable architecture, which is a core contribution of this work. In contrast, baselines like P(all-atom) are architecturally limited. For instance, P(all-atom) relies on memory-intensive triangular updates; generating a single sample of 500 residues requires over 140GB of GPU memory. Consequently, training P(all-atom) on the longer lengths used for La-Proteina is likely computationally infeasible on standard hardware. Therefore, we view our evaluation as a fair reflection of the practical capabilities and scalability inherent to each method.
>
> That said, to directly address the concern about "in-distribution" performance and isolate the benefits of our latent diffusion approach, we have added a new analysis in Table 11 (updated manuscript). We evaluated all-atom co-designability and diversity for all models in three "in-distribution" scenarios where baselines are expected to perform well: (A) 100 proteins of length 100, (B) 200 proteins (100 each for lengths 100 and 200), (C) 300 proteins (100 each for lengths 100, 200, and 300). We observe that La-Proteina achieves state-of-the-art performance in all three scenarios, confirming that La-Proteina’s state-of-the-art performance is driven by the method’s formulation, not merely by evaluating baselines beyond their training regimes.

---

> > ### Author Response · Authors · 2025-11-21
> > **Answer (part 2)**
> >
> > **[Incompleteness in Key Evaluation Metrics ... Could the authors report the distribution of pLDDT scores for the successfully refolded designs?]**
> >
> > We agree with the reviewer that incorporating the pLDDT confidence metric would provide additional insights and strengthen our empirical evaluation. To address this, we analyzed the distribution of pLDDT scores for all co-designable samples generated by La-Proteina and baselines, and we have included these results in the updated manuscript (Table 1 and Figure 34). The analysis confirms the high quality of La-Proteina's designs: our samples yield the highest pLDDT values among all baselines, with over 95% of successfully refolded designs achieving a pLDDT score greater than 80. Furthermore, the standard La-Proteina model (without triangular update layers) achieved a mean pLDDT of approximately 85 (standard deviation 2.7), while the version incorporating triangular update layers yielded an even higher mean of approximately 87 (standard deviation 2).
> >
> > **[On the Choice of Training Data (AFDB vs. PDB) ...]**
> >
> > This is a good point. While the PDB is indeed the gold standard for experimental structures, training on the AlphaFold Database (AFDB) has become a fairly standard and successful choice for large-scale generative modeling. We followed several recent state-of-the-art methods, such as Proteina [1], Genie2 [2], and AmbientProteins [3], which leverage the massive scale of the AFDB. Even P(all-atom), a key baseline, uses a dataset where the vast majority of structures (approx. 77%) come from AFDB.
> >
> > Our rationale was to build on this established practice, and as our results show, it was highly effective for achieving state-of-the-art performance for atomistic design tasks. We agree that combining both PDB and AFDB for training would be an interesting direction, but we did not explore that specific axis in this work.
> >
> > [1] Proteina: Scaling Flow-based Protein Structure Generative Models. Geffner et al.
> >
> > [2] Out of Many, One: Designing and Scaffolding Proteins at the Scale of the Structural Universe with Genie 2. Lin et al.
> >
> > [3] Ambient Proteins: Training Diffusion Models on Low Quality Structures. Daras et al.
> >
> > **[On the Performance of a Fully Latent Representation ... a fully latent representation (the CA-enc. variant) leads to a significant degradation in performance ...]**
> >
> > This is an excellent observation, and this ablation was indeed a critical finding in our work. While we don't have a definitive answer, we believe the primary reason is not a fundamental inability to represent the coupled degrees of freedom, but rather the difficulty to optimally generate them when they are forced into a single, unified flow.
> >
> > Our VAE ablation that encodes the alpha-carbons (CA-enc) was actually successful as an autoencoder, achieving excellent reconstruction (around 0.1Å all-atom RMSD and perfect sequence recovery), thus showing that learning a unified latent representation is possible. However, the generative performance was not good. We believe this is because the separate treatment is what enables the use of different generation schedules and sampling temperatures for the backbone and the side-chain/sequence latents, which is critical for high performance.
> >
> > As the reviewer notes, all-atom models like La-Proteina and P(all-atom) both effectively generate the backbone structure at a faster rate than the side-chain details. La-Proteina does this explicitly. Our partially latent model allows us to use independent schedules for alpha-carbons and latent variables. P(all-atom) does this implicitly through its atom14 representation. Side-chain atoms are noised faster in the forward (noising) process and thus generated later (i.e., on a slower effective schedule) in the reverse (generative) process.
> >
> > We hypothesize that this is beneficial due to the different properties of the mappings between backbone -> sequence/side-chain and sequence/side-chain -> backbone. It is possible that one of these conditional directions is significantly easier for the model to learn, and the schedules mentioned above allow the model to leverage this asymmetry during generation.
> > In summary, we believe the failure of the fully latent variant isn't because the representation is impossible to learn, but because it prevents the decoupled generation process (different schedules, temperatures) that our ablations show is essential for high-quality, co-designable atomistic generation.

---

> > > ### Author Response · Authors · 2025-11-21
> > > **Answer (part 3)**
> > >
> > > **[Modeling Long-Range Side-Chain Interactions ...]**
> > >
> > > Fig. 8 shows the VAE's latent space operates predominantly in a local way. However, this is a feature of the VAE's representation, not a limitation of the overall generative framework. The long-range dependencies are captured by the partially latent flow matching model. This model relies on a powerful Transformer that learns the joint distribution between alpha-carbon coordinates and latent variables over all residues. By operating on the full sequence of latents, the flow matching model explicitly learns the complex, non-local correlations and inter-residue dependencies. This is conceptually similar to a hypothetical model that operates explicitly in data space (P(all-atom), for instance). In such a model, the coordinates of a single atom are a fully "local" representation. In both cases, it is the generative model (the diffusion / flow matching model) that is responsible for capturing the global context and learning the joint dependencies between all variables.
> > >
> > > As an illustrative analogy, many generative models for images first encode the image into a set of local patches. The latent representation for any single patch is local. However, the latent diffusion model operating on these patch latents learns the global correlations between them to generate a coherent, large-scale image. Our approach is the same: the VAE provides (potentially local) latent variables, and the flow matching model learns their global joint distribution. The strong performance of La-Proteina on co-designability, structural validity, and atomistic motif scaffolding, which often requires long-range interactions to stabilize the motif, serves as the quantitative validation that these dependencies are being successfully captured.
> > >
> > > -----
> > >
> > > We sincerely thank the reviewer for the insightful critique that guided our revisions. Please let us know if our answer fully addresses your concerns. We remain available for any further questions, and welcome any final discussion points that might lead to a reconsideration of your score. Thank you.

---

> > > > ### Comment · Reviewer_gg8b · 2025-11-27
> > > >
> > > > Thank you for your detailed response. You have effectively addressed my concerns/questions, and the revisions have further improved the quality of the paper. Consequently, I will raise my score.

---

> > > > > ### Author Response · Authors · 2025-11-28
> > > > > **Thank you**
> > > > >
> > > > > We’re glad to hear that we addressed your concerns and are happy with the revisions; and we appreciate that you raised your score to 8 (on Nov. 26th). Thank you for your feedback and helping us strengthen our work!

---

### Official Review · Reviewer_UiFG · 2025-10-31

**Soundness:** 4
**Presentation:** 3
**Contribution:** 3
**Rating:** 8
**Confidence:** 3

**Summary:**

La-Proteina introduces a partially latent flow matching framework for joint protein sequence and full-atom structure generation. The key idea is to model the protein backbone explicitly (using α-carbon coordinates) while encoding all side-chain atoms and sequence information in a fixed-size latent vector per residue. This hybrid representation avoids the combinatorial complexity of direct discrete side-chain modeling and enables powerful continuous generative dynamics on the backbone+latent space. The authors first train a VAE to map proteins (sequence plus structure) into the per-residue latent space and back, then train a flow-matching model (a form of diffusion) to jointly generate new backbone coordinates and latent vectors. They claim the following major contributions and results:

**(1)** A novel partially-latent protein generative model that combines explicit backbone modeling with latent side-chain+sequence representation.

**(2)** State-of-the-art performance on unconditional protein generation benchmarks, significantly outperforming prior methods in co-designability (ability to jointly fold sequence+structure), diversity, and structural quality.

**(3)** Unique scalability to very long proteins: La-Proteina can generate plausible 800-residue designs where other all-atom models run out of memory or collapse (essentially failing beyond 500 residues).

**(4)** Successful demonstration of fully atomistic motif scaffolding: given a small active-site motif (set of residues with known 3D arrangement), La-Proteina can build new proteins that accurately incorporate that motif. This holds even in the challenging “unindexed” setting where the motif’s position in the sequence is not specified. The model outperforms previous scaffolding approaches (which were limited to backbone-only or indexed placement) and solves most tasks in a standard benchmark.

**(5)** Extensive analyses are provided: ablation studies show the importance of design choices, and structural evaluations indicate that La-Proteina’s samples are substantially more physically realistic than those of prior all-atom models.

**Strengths:**

- The proposed representation elegantly addresses a core difficulty in protein generation: the mixed discrete (sequence) and continuous (structure) nature with variable side-chain sizes. By encoding side-chain atoms and amino acid type into a fixed continuous latent per residue, La-Proteina avoids having to explicitly model discrete sequence choices and variable atom counts during generation. This is a novel solution not seen in prior protein diffusion models, which either treated all atoms explicitly (incurring huge complexity) or attempted fully latent approaches. The authors demonstrate that this hybrid strategy retains crucial backbone information explicitly while deferring fine atom details to a learned latent space – an approach supported by ablations (if the backbone is also put in latent, performance drops sharply). This design is well-motivated and grounded in prior successes in related domains (e.g. latent diffusions in image generation) and backbone-only protein modeling, but extends them in a non-trivial way for the joint sequence–structure task. By leveraging flow matching (a deterministic counterpart of diffusion) on this simplified space, the method can generate proteins end-to-end while maintaining high fidelity in both structure and sequence, as confirmed by the VAE’s excellent reconstruction (0.12Å all-atom RMSD, 100% sequence recovery).

- State-of-the-Art Unconditional Generation Performance: La-Proteina convincingly achieves SOTA results on standard protein design metrics. On the all-atom unconditional generation benchmark (100–500 residue range), it outperforms a slate of recent models like P(all-atom), APM, PLAID, ProteinGenerator, Protpardelle and Protpardelle-1c, in nearly every metric. In particular, La-Proteina’s designs have dramatically higher co-designability (e.g. 68–75% vs ~9–37% for baselines) and designability scores, showing that its jointly generated sequences truly fold into the generated structures with high fidelity. It also produces orders-of-magnitude more diverse structures (clustering yields ~180–216 unique clusters vs tens for others) without sacrificing novelty. Importantly, structural realism is superior: La-Proteina’s samples have far better MolProbity scores than baseline outputs. Additionally, La-Proteina accurately reproduces the distribution of side-chain dihedral angles (rotamers) seen in natural protein databases. Prior methods often missed certain rotamer states or had incorrect preferences, whereas La-Proteina’s generated residues match reference distributions almost perfectly. These results support the authors’ claim of overall superior quality. The evaluation is extensive, and the improvements are significant. Notably, La-Proteina even surpasses the previous state-of-the-art “Proteina” backbone-only method on its own terms it in fact excels, presumably due to the integrated training of sequence+structure rather than sequential design.


- A standout strength is La-Proteina’s ability to handle much larger proteins than competing methods. The paper demonstrates the generation of proteins up to 800 residues (close to the upper limit in their training set) that remain co-designable, diverse, and physically plausible. All baseline all-atom models failed catastrophically in this regime (e.g. many “collapsed” and produced no co-designable samples beyond 400–500 residues). Empirically, using the same hardware that couldn’t generate a single 600-residue sample for P(all-atom) (a 140GB GPU), La-Proteina comfortably generates 800-residue proteins. The samples are not only valid but high-quality at those lengths. It should be noted that this success is in part due to engineering choices: the authors trained on an enormous dataset of ~46 million structures (AlphaFold DB) to expose the model to long proteins.

**Weaknesses:**

- A notable limitation is that La-Proteina is only demonstrated on single-chain proteins, whereas many real design tasks involve multi-chain complexes or protein–protein interactions. The authors explicitly acknowledge this as future work. While focusing on monomers is reasonable for a first step (and they already handle up to 800 residues in one chain), it means the method currently cannot design binding interfaces or multi-subunit assemblies. Competing approaches have introduced multi-chain generation modes (e.g. for homomers or heteromers). Not testing La-Proteina on complexes is a missed opportunity to show generality. It is unclear how easily the model could be extended to complexes (with potentially even larger lengths), which could be challenging. The authors’ own Limitations section notes that protein assemblies and interfaces were not addressed, which are crucial for tasks like binder design. Thus, the work, while strong for monomer design, falls short on multi-chain design which remains an open challenge.

- La-Proteina’s impressive performance comes at the cost of very large-scale training. The model was trained on 46 million structure-sequence pairs (essentially the entire high-quality samples of AlphaFold DB) using 128×80GB GPUs, a heavy computational load. This far exceeds the training dataset sizes of most baselines (many prior models used on the order of 0.1–1 million PDB-derived examples). While the ability to leverage massive data is a plus for performance, it raises concerns of reproducibility and fairness of comparison. Some of La-Proteina’s gains might come from sheer dataset scale and network size, rather than purely algorithmic superiority. It’s plausible that if others had similar GPU budgets or training data, the gap might narrow. The authors do mention that P(all-atom) and others are constrained by architecture from scaling to AF2 data, which justifies their approach. However, from a practical standpoint, not all researchers can train on tens of millions of samples. The paper would be stronger if it discussed how performance scales with data (e.g. is a subset of 1M or 10M sufficient to beat baselines?). Is it because of the new architecture or simply training on an unprecedented dataset?

- As with most computational design papers, this work stops at in silico metrics. A natural question is whether La-Proteina’s designs are actually foldable and functional in vitro. The paper does not include any experimental assay or even Rosetta energy minimization analysis to check stability. While such validation is beyond the scope of ICLR, it means the ultimate utility is inferred rather than proven. For example, co-designability uses ESMFold (a fast structure predictor) to verify that the sequence can fold into the model’s structure. This is a reasonable proxy, but ESMFold is not as accurate as AlphaFold2 on novel sequences; some false positives/negatives could occur. The model might generate sequences that fool ESMFold but would not truly fold (or vice versa). Also, MolProbity scores being good is encouraging, yet those are still computed on the designed models themselves. Without relaxation or physics-based evaluation, we don’t know if slight clashes would resolve or if the designs have hidden strain. Functionally, the motif scaffolding examples are promising (they preserved active-site geometries), but the paper doesn’t test if the enzymes would actually be active, that would require molecular dynamics or lab experiments. Again, this is not expected in a CS-focused venue, but it’s a gap to be aware of. One could imagine that certain subtleties (like global stability or kinetic foldability) are not captured by the metrics used. This is a general weakness of the field rather than this paper specifically, but since the authors tout “fully atomistic high-quality structures,” an expert might ask for a bit more evidence (e.g. computing Rosetta folding stability or sequence conservation analysis). The paper does perform a rotamer analysis and secondary structure content check, which are good sanity checks. However, one missing evaluation I note is any measure of thermodynamic stability or model confidence for the generated proteins. For example, methods like AlphaFold2 or Rosetta Relax could be used on the designed sequence to see if it strongly prefers the designed structure. Co-designability partly does this (AlphaFold via ESMFold), but reporting an average pLDDT could add insight. In summary, the lack of real-world validation is a weakness insofar as the paper’s claims rest on proxy metrics. This is not a fatal flaw for an ML paper, but it means the true impact (e.g. designing a new enzyme from scratch) is still unproven.

**Questions:**

- How do you envision extending La-Proteina to design protein complexes or multi-chain assemblies? Since your current model represents one continuous chain with per-residue latents, would handling multiple chains simply be a matter of adding chain-break tokens or separate latent sequences per chain? Are there any obstacles (e.g. interfacing latents between chains, increased length) that you foresee? For instance, could the flow matching scale to, say, a heterodimer of 2×400 residues (total 800) as easily as a single 800-res chain?


- Your training set of 46M samples is enormous. Did you observe how performance scales with fewer training examples or smaller models? For example, if only the PDB (~0.5M structures) were used, does La-Proteina still outperform baselines? Similarly, how crucial is the 130M parameter Transformer architecture? It would help to know if the method could work on more limited data/resources (perhaps with reduced output quality but still above prior methods). Any insight on the data efficiency (like learning curves) would be valuable for researchers who cannot train at that massive scale.


- Have you tried a two-stage approach as a baseline, e.g., first generate a backbone with your flow model (ignoring latents), then design sequence/side-chains using an external model (like ProteinMPNN)? Essentially, how much do we gain by co-generating sequence and structure together, versus sequentially? Your results suggest co-generation is superior (given the high co-designability vs baseline designability), but a direct comparison would confirm the benefit of coupling. If not explicitly tested, what’s your intuition: is the improvement mostly because the model can adjust backbone positions to accommodate sequence mutations on the fly (something a sequential process can’t do)?


- You report almost all samples are valid/co-designable and even tricky motifs are scaffolded well. Did you notice any recurring failure modes or patterns in the rare cases where La-Proteina fails? For example, in Figure 11 you show one inconsistency (a rotated ring) in a motif placement. Are such errors random, or do they happen more for certain residue types or structural contexts (e.g. certain ligand-binding motifs or β-sheet-rich motifs)? Understanding when the model struggles could guide future improvements. Additionally, during generation, do you ever get completely broken structures (e.g. chains that don’t fold into a single globule, or severe steric clashes)? If so, how do you detect or filter those? Overall the error rates seem low, but any insight into “edge cases” would be helpful.


- Your diversity and novelty metrics indicate many unique structures are generated, but the average TM-score to known structures is ~0.75, implying the samples are still reminiscent of known folds. Can you provide qualitative examples of how novel the top designs are? For instance, have you found any new topologies (combinations of secondary structure never or rarely seen in PDB)? It would be reassuring to see that La-Proteina isn’t just reproducing common motifs in protein structure (like αα-hairpins, helix-loop-helix, etc.) but can also generate creative arrangements. If you haven’t assessed this, do you plan to? Perhaps tools like Foldseek or manual inspection could identify if any design is a true outlier relative to the training set. This will shed light on the model’s ability to extrapolate versus interpolate known protein geometry.


- Following up on the lack of experimental tests, do you intend to collaborate or perform wet-lab validation of some La-Proteina designs? For example, taking a few long designs or motif scaffolds and expressing them to see if they fold and function. Given the model’s strengths, it seems poised to attempt de novo enzyme design or binder creation. If not already in progress, what would be the first application you’d target with La-Proteina (e.g. designing a new enzyme for a reaction by scaffolding a catalytic motif, or creating a large megadalton protein cage, etc.)? Any discussion on how well the in silico metrics correlate with experimental success (perhaps referencing prior work) would help set expectations for readers interested in deploying these designs.


If the authors address the weaknesses outlined above, or provide a right justification for the points raised in my questions, or clarify points I may have misunderstood, I would be willing to raise my score from 8 to 10.

**Details Of Ethics Concerns:**

**Dual-use biosafety risk.** The paper enables design of fully atomistic proteins and demonstrates atomistic motif scaffolding (including unindexed placement), which could be repurposed to scaffold functional/active sites in harmful proteins. The authors themselves acknowledge that such models “could, in principle, be misused” and pose biosecurity concerns; an ethics review focused on biosafety risk assessment and mitigation is appropriate.

**Lowering barriers to misuse.** The paper provides a general-purpose pipeline for joint sequence+full-atom generation, plus exact training/evaluation details and code to reproduce results. This substantially lowers the effort to generate functional atomistic designs, including catalytic motifs, and could be misapplied without safeguards.

---

> ### Author Response · Authors · 2025-11-21
> **Answer (part 1)**
>
> We appreciate the reviewer's detailed and insightful feedback, for highlighting our method’s performance and scalability, and providing insightful questions and comments to improve our work.
>
> **[La-Proteina is only demonstrated on single-chain proteins, whereas many real design tasks involve multi-chain complexes or protein–protein interactions ... It is unclear how easily the model could be extended to complexes ...]**
>
> This is an excellent point. This was raised by multiple reviewers, and we have included a detailed discussion in our General Response. In short, this work's primary focus was on establishing La-Proteina as the first scalable and high-performing method for fully atomistic design and validating its core methodology on monomer tasks. However, the framework is not limited to monomers. As we detail in the general response and mention in the related work section, a concurrent submission (Anonymous, 2025) successfully extends La-Proteina to binder design, achieving state-of-the-art results and confirming its generalizability to other interesting and relevant conditional tasks involving the modeling of protein-protein interactions.
>
> **[Very large-scale training. The model was trained on 46 million structure-sequence pairs using 128 GPUs. This exceeds the training dataset sizes of most baselines (which used on the order of 0.1–1 million examples) ... The paper would be stronger if it discussed performance when trained on smaller datasets (e.g. 1M or 10M samples). Is it because of the new architecture or simply training on an unprecedented dataset? ... how crucial is the 130M parameter Transformer architecture?]**
>
> *--Dataset size and number of GPUs*
>
> The reviewer is not entirely correct regarding the dataset size and GPU usage for our core models. We appreciate the chance to clarify the differences between our main models and our long-length scaling experiment.
>
> The La-Proteina models presented in our main results table (Table 1) were primarily trained on a dataset consisting of approximately 350k samples (as described in Appendix C.1), which is well within the 0.1M–1M range mentioned in the review. The model without triangular update layers was trained using 48 GPUs (as described in Appendix C.2.2). Overall, this model vastly outperforms baselines training on a dataset of 350k samples and using 48 GPUs.
>
> The setup involving the 46 million samples and 128 GPUs was used exclusively to train La-Proteina for our unconditional long-length generation benchmark (up to 800 residues). This experiment was designed specifically to showcase the model’s unique scalability to protein lengths that baselines cannot handle, which is a key contribution. In summary, La-Proteina achieves SOTA results using a dataset of approximately 350k samples, trained on 48 GPUs. The massive 46M sample dataset was used only to demonstrate the scalability and robustness of the method for generating complex, long proteins.
>
> As an ablation, we are currently training the same model as above (160M parameters, dataset with ~350k samples, no triangular update layers) on 8 GPUs instead of 48. While training is ongoing (results appear to be consistently improving) preliminary numbers show that the model achieves all-atom co-designability of 55.4%, with diversity (# of clusters) values of 208 (struct), 186 (seq), and 269 (seq+struct). While this is expectedly somewhat below the performance of our main La-Proteina model trained on 48 GPUs (first La-Proteina line in Table 1), it outperforms baselines by a large margin, showing that La-Proteina yields SOTA results when trained on just 8 GPUs.
>
> *--Model size*
>
> While the model is large (160M parameters for the latent diffusion), as shown in the paper our model is very scalable. We emphasize this is because of the architecture used, which is deliberately designed for scalability and efficiency despite its large parameter count. In fact, other models are significantly less efficient and scalable than La-Proteina despite having significantly less parameters, due to alternative architectural choices (e.g. Pallatom ~18M parameters). We therefore do not see the 160M parameters in the transformer as limiting the method in any way, as it does not have a negative impact on the method’s scalability nor efficiency. Specifically, our speed and memory analysis in Appendix I shows that La-Proteina (without triangular layers) and Protpardelle enjoy the fastest sampling times, orders of magnitude faster than P(all-atom), the most capable baseline. Additionally, the latter model is extremely memory intensive (despite its low parameter count, due to other architectural choices, as pointed out in the review), requiring over 80GB of memory to produce a single sample of length 500. Overall, while La-Proteina relies on an architecture with a higher parameter count than most baselines, it achieves state-of-the-art performance while remaining faster and more memory efficient than competing methods.

---

> ### Author Response · Authors · 2025-11-21
> **Answer (part 2)**
>
> **[This work stops at in silico metrics ... While such validation is beyond the scope of ICLR, it means the ultimate utility is inferred rather than proven ... (other potential proxy in silico metrics)]**
>
> We appreciate the depth of this analysis, which recognizes the challenges and standards of validation in computational design. We agree that while experimental validation is the ultimate test, it remains, as the reviewer acknowledges, beyond the scope of a methodology-focused CS venue like ICLR. We follow the field's standard practice by reporting the widely used proxy metrics of co-designability, diversity, novelty, and designability. We also take this empirical analysis a step further by providing detailed MolProbity scores and a comprehensive side-chain rotamer distribution analysis, offering superior insight into the geometric and biophysical plausibility of our atomic designs compared to most current ML papers in the field.
>
> We are pleased to report that we have strengthened our evaluation by addressing the need for a confidence metric. We analyzed the distribution of pLDDT scores for all successfully redesigned samples generated by La-Proteina and baselines, and included the results in the updated manuscript (Table 1 and Figure 34). The analysis confirms the high quality of La-Proteina's designs: our samples yield the highest pLDDT values among all baselines, with 95% of successfully refolded designs achieving a pLDDT score greater than 80. Furthermore, La-Proteina (without triangular layers) achieved a pLDDT of $85 \pm 2.7$, while the version incorporating triangular update layers yielded an even pLDDT of $87 \pm 2$.
>
> To address the concern regarding the physics-based relaxation, we performed an additional analysis on the co-designable samples generated by La-Proteina (temperatures 0.1 and 0.1). We applied an Amber restrained minimization protocol, similar to methods used in OpenFold (https://github.com/aqlaboratory/openfold/blob/main/openfold/np/relax/amber_minimize.py), to the generated structures. The results indicate minimal structural deviation post-relaxation: the mean RMSD between the structure before and after this minimization step is just $0.085 \pm 0.009$ Å. This demonstrates that the structures generated by our model are favorable under a physical force field without requiring significant atomic rearrangement, implying their realism and stability.
>
> **[How to extend La-Proteina to design protein complexes?]**
>
> We agree with the reviewer that adding chain-break tokens is a simple but potentially promising way to handle general design tasks involving protein complexes. We do not foresee any issues with that approach. In the concurrent submission that extends La-Proteina to binder design, the main change to the model involves conditioning La-Proteina on the target structure’s coordinates and sequence, appending them to the main sequence processed by the transformer. In that work, a binary pair feature indicating chain identity (e.g. chain-residue-1 == chain-residue-2) was used to identify residues in different chains (closely related to the chain break token proposed in the review).
>
> **[Have you tried a two-stage approach as a baseline, e.g., first generate a backbone with your flow model (ignoring latents), then design sequence/side-chains using an external model (like ProteinMPNN)?]**
>
> We did not thoroughly benchmark a two-stage sequential approach (backbone generation followed by external sequence design, e.g., ProteinMPNN). Our goal was to establish La-Proteina as a foundation model for fully atomistic design that the community can build upon. One successful example of this is the concurrent submission (Anonymous, 2025; mentioned in our related work section) which extends La-Proteina to the binder design task, achieving SOTA results.
> We believe joint modeling, as implemented in La-Proteina, offers several crucial benefits over a sequential, backbone-then-sequence approach. Many protein design tasks (such as atomistic scaffolding, binder design, etc.) critically rely on side-chain conformations and atomistic interactions. We argue that explicitly modeling side-chain atoms (which necessarily requires modeling sequence) aids the generative model in learning these complex atomic-level interactions better. The strength of this joint approach is observed in the concurrent submission that extends La-Proteina to binder design, which outperforms methods using the backbone + ProteinMPNN sequential approach (e.g., RFDiffusion) by a large margin. Additionally, many tasks in protein design involve atomistic constraints (e.g., specific distances between atoms, angles, or covalent bonds between two atoms). Handling such detailed constraints, which are often specified for side-chain atoms, requires the atomistic modeling capability that a joint framework provides. (For an example of such constraints, see the concurrent BoltzGen paper [1].)
>
> [1] BoltzGen:Toward Universal Binder Design. Stark et al.

---

> > ### Author Response · Authors · 2025-11-21
> > **Answer (part 3)**
> >
> > **[Any recurring failure modes or patterns in the rare cases where La-Proteina fails?]**
> >
> > We agree that understanding edge cases is crucial for future development. Overall, La-Proteina exhibits a very low rate of hard failures and almost never produces completely broken structures. We believe this robustness is supported by the biophysical validity observed in our MolProbity evaluations, which demonstrated significantly fewer clashes and geometric errors compared to all baselines across all generated samples, not just the co-designable ones. For atomistic details, such as side-chain rotamers, we rarely observe strange atomic configurations, such as poorly formed rings (e.g., only one atom placed incorrectly).
> >
> > We hypothesize that the robustness of the generated structures stems partly from the Variational Autoencoder (VAE) component. Our decoder is only trained to reconstruct valid structures from the data distribution. We believe that even if the flow matching model introduces minor errors in the latent space during generation, the VAE decoder may act as a "correction" step. This constraint may enable it to map a slightly flawed latent input back into a structurally valid protein that minimizes major errors in atom placement. In contrast, methods that operate directly in atomistic data space lack this “decoding correction” step, which may make them more susceptible to propagating or generating severe atomic errors.
> >
> > **[Your diversity and novelty metrics indicate unique structures are generated, but the average TM-score to known structures is ~0.75, implying the samples are still reminiscent of known folds.]**
> >
> > While our metrics show La-Proteina achieves state-of-the-art co-designability and diversity, and strong novelty compared to most baselines (outperforming every method except for P(all-atom), whose novelty score is only slightly better than ours; 0.72 vs. 0.75, based on the PDB reference set), we acknowledge that an average TM-score of $\approx 0.75$ still implies a structural resemblance to known folds.
> >
> > We conducted a qualitative inspection of the most novel co-designable samples produced by La-Proteina (those with TM-scores close to 0.5 with respect to the PDB, the lowest observed range in the co-designable samples produced by the model). We observe that, while these samples are structurally distinct from existing proteins in the PDB, they generally consist of common structural motifs. Generating truly unique, never-before-seen topologies is a challenging task. Recent work attempts to achieve this by explicitly pushing a model out of its training distribution, by conditioning on structural templates [1] or using 3D shape guidance [2]. La-Proteina’s high performance suggests it is a robust foundation that could be extended with such guidance mechanisms (e.g., shape or motif constraints) to target specific, novel topologies in future work. For the scope of this paper, we demonstrate that La-Proteina achieves the best trade-off between diversity and c0-designability in the general, unconditional setting, while still achieving competitive novelty values.
> >
> > [1] Exploring ‘‘dark-matter’’ protein folds using deep learning. Harteveld et al.
> >
> > [2] ProtComposer: Compositional Protein Structure Generation with 3D Ellipsoids. Stark et al.
> >
> > **[Do you intend to collaborate or perform wet-lab validation of some La-Proteina designs?]**
> >
> > We agree experimental validation is the ultimate test for any de novo design method. We are indeed currently in the process of experimentally evaluating designs from La-Proteina. This is an early but active effort, and it is a separate, significant endeavor that is beyond the scope of this methodological paper, as acknowledged in the review. Our current focus is on assessing how well the proteins express and whether they fold as predicted. We also plan to investigate how properties like thermodynamics, expression, solubility, are affected by design choices. This includes comparing La-Proteina's co-designed sequences versus MPNN-designed sequences and varying the sampling temperature during generation. While this is all very early stages, our initial wet-lab findings regarding expression and structural similarity to design models seem promising.
> >
> > -----
> >
> > We are grateful to the reviewer for the thoughtful feedback on our revised manuscript. Please inform us if any questions or concerns persist, or if there are additional discussion points we could provide to support a score increase. Thank you.

---

### Official Review · Reviewer_duda · 2025-11-01

**Soundness:** 4
**Presentation:** 3
**Contribution:** 3
**Rating:** 8
**Confidence:** 4

**Summary:**

This paper introduces La-Proteina, which addresses the protein codesign problem. La-Proteina tackles this via a "partially latent" representation. The C-alpha backbone coordinates are modeled explicitly, while the sequence identity and all other atomistic details are compressed into fixed-size, per-residue latent variables using a VAE. A scalable flow matching model that circumvents triangular update layers learns the joint distribution in this hybrid explicit-latent space. This also allows separation of generation processes for the coordinates and latents, which allows different generation schedules during inference. Empirically, La-Proteina achieves SOTA on co-designability and diversity benchmarks. The model also does well on long-sequence generation up to 800 residues. Additional conditional capabilities are demonstrated, including motif scaffolding without pre-specifying the positions.

**Strengths:**

1. Strong empirical performance across the board.
2. Hybrid representation with the partially latent dimension is thoughtful. It allocates enough resolution to the key backbone portion while circumventing the issue of needing to pre-specify the number of atoms.
3. Biophysical evaluations are compelling; examining for clash rates and rotamer frequencies sets a new evaluation metric that is meaningful for this subfield.
4. The technical ability to decouple the timesteps necessary is thoughtfully designed, and the evidence for the argument that backbones require fewer inference timesteps than latent is illuminating.
5. Unindexed motif scaffolding is a more meaningful conditional task, great to see it included.

**Weaknesses:**

* A short coming of having two noise schedules is that we now need to tune two noise schedules – authors were able to get good results regardless, but this is a small concern for practical usage.
* Not so much specific to this paper, but generally conditional generation becomes much more interesting than pure unconditional generation. Would strengthen the paper if more of the work was focused on that aspect.

**Questions:**

* AFAICT, the 8-dimension latent space does not report ablations. It would be interesting to better understand this dimensionality choice, especially considering the fact that we have some ground truth knowledge of the average number of atoms in a sidechain residue. Was this chosen based on ablations or intuition?

---

> ### Author Response · Authors · 2025-11-21
> **Thank you for your positive review**
>
> We thank the reviewer for the positive review, emphasizing our strong performance and extensive biophysical evaluations.
>
> **[Two noise schedules ... we now need to tune both ... this is a small concern for practical usage.]**
>
> We found that this is not a significant issue, since these schedules are inference-time hyperparameters. Tuning them is a fast ablation study that does not require retraining. More importantly, these schedules are robust and re-usable. Our ablations (App. G.2) show a clear principle: the backbone must be generated faster than the latents for good co-designability. Also, the schedules we identified are not brittle; a concurrent submission (Anonymous, 2025), which extends our La-Proteina framework to binder design, re-uses the same sampling schedules and achieves state-of-the-art results. This strongly suggests they are a "find once"-style parameter. Our approach is also grounded in prior work; we built upon proven methods by using the same fast "exponential" schedule for the alpha-carbons as the state-of-the-art backbone generator, Proteina [1]. Finally, we also note that the use of multiple schedules is a common pattern for design methods; other high-performance generators for backbones, like SO(3)-based flow matching models (FrameFlow, FoldFlow), also find it critical to use different schedules for rotational and translational components used to model residues.
>
> Additionally, having two separate schedules for backbone and sidechains offers practical advantages, allowing natural extensions of our method to fixed/slightly noised backbone redesign as well as other interesting applications that we leave for future work.
>
> [1] Proteina: Scaling Flow-based Protein Structure Generative Models. Geffner et al.
>
> **[Not so much specific to this paper, but conditional generation is more interesting than pure unconditional generation...]**
>
> We agree with the reviewer that conditional generation is a critical and highly interesting area for protein design. We dedicated a significant portion of our work to this aspect by focusing on the challenging task of atomistic motif scaffolding. We evaluated La-Proteina on this task in four distinct setups: all-atom indexed, all-atom unindexed, tip-atom indexed, and tip-atom unindexed, demonstrating state-of-the-art performance across all of them.
>
> Other conditional tasks, such as binder design, are particularly relevant in this domain and other reviewers asked about it as well. We include a detailed discussion about this in our General Response. In short, this work's primary focus was on establishing La-Proteina as the first scalable and high-performing method for fully atomistic design and validating its core methodology on monomer tasks. However, the framework is not limited to monomers. As we detail in the General Response and mention in the related work section, the concurrent submission (Anonymous, 2025) successfully extends La-Proteina to binder design, achieving state-of-the-art results and confirming its generalizability and applicability in other design tasks.
>
> **[The 8-dimension latent space does not report ablations. It would be interesting to better understand this choice... How was this chosen?]**
>
> While we did not ablate training multiple latent diffusion models on the representations provided by VAEs with different latent dimensionalities, we can provide additional context on how this choice was made. The choice of 8 dimensions for the per-residue latent variable was initially guided by balancing representational capacity with computational efficiency. To validate this choice, we trained the VAE with alternative latent dimensionalities of 4 and 16, in addition to the standard 8 used throughout the paper. In all three cases the VAE achieved perfect sequence recovery. We observed the following trends in structure reconstruction error: (1) The structure reconstruction error for D=8 and D=16 was very similar, around 0.12A; (2) The VAE trained with D=4 resulted in a reconstruction error that was approximately 10% higher than D=8 or D=16, and it required ~50% more training steps to reach that performance.
>
> Based on these VAE training results we chose D=8 as the default to ensure optimal performance while maintaining efficiency. We did not proceed to train the full latent flow matching model with the VAEs resulting from D=4 or D=16. However, given the high representational fidelity achieved by the D=8 latent space, we hypothesize that moving to D=16 would not yield significant improvements in the downstream generative performance, as the dimensionality of 8 appears sufficient to properly encode the sequence and side-chain atoms into a latent space that’s well suited to a diffusion model.
>
> -----
>
> We thank the reviewer again for the thoughtful review. Please let us know if you have any further questions or concerns, or if there’s anything additional discussion points you’d like to cover to consider raising your score. Thank you!

---

### Official Review · Reviewer_4Tcg · 2025-11-01

**Soundness:** 4
**Presentation:** 4
**Contribution:** 4
**Rating:** 10
**Confidence:** 4

**Summary:**

This paper proposed a novel partial latent flow matching for full-atom protein generation. They encode varying-length side chains and categorical residue types into the latent space, and maintain the explicit $\alpha$-carbon-based protein backbone representation in the coordinate space. They achieve SOTA performance on unconditional and conditional full-atom protein generation benchmarks.

**Strengths:**

The motivation is reasonable, proteins' natural hierarchical structure (backbone + side chains) and variable-length side chains are suitable for partial latent generation.

The method is presented clearly; training and inference hyperparameters are provided for reproducibility.

The experimental part is well-designed, and they achieve SOTA results on unconditional and conditional full-atom protein generation benchmarks. They also provide many ablation and analysis experiments to support their method.

**Weaknesses:**

**W1. The La-Proteina model has only been trained and tested on monomeric proteins. Can it be trained or fine-tuned on multi-chain data (protein complex design), such as APM[1]?**

**W2. Folding can be regarded as a kind of conditional full-atom generation. Can La-Proteina compare on the folding benchmark?**


[1].An All-Atom Generative Model for Designing Protein Complexes

**Questions:**

See Weaknesses.

**Q1. Both equivariant rigidity-based methods like FrameFlow[1] or FoldFlow[2], or non-equivariant methods in coordinate space like Proteina or La-Proteina ($\alpha$-carbon), all use a fast scheduler during the inference to ensure the quality of generated proteins. Do you have any insights or possible explanations for this phenomenon?**

[1].Fast protein backbone generation with SE(3) flow matching

[2].SE(3)-Stochastic Flow Matching for Protein Backbone Generation

---

> ### Author Response · Authors · 2025-11-21
> **Thank you for the positive review**
>
> We thank the reviewer for the positive and constructive feedback, highlighting our method’s strong performance, the thoroughness of our ablations, and clarity in the presentation.
>
> **[La-Proteina has only been trained and tested on monomers. Can it be trained / fine-tuned on multi-chain data?]**
>
> Thank you for raising this point. This is a critical question that was raised by several reviewers, and we have included a detailed discussion in our General Response. In short, this work's primary focus was on establishing La-Proteina as the first scalable and high performing method for fully atomistic design and validating its core methodology on monomer tasks. However, the framework is not limited to monomers. As we detail in the general response and mention in the related work section, a concurrent submission (Anonymous, 2025) successfully extends La-Proteina to binder design, achieving state-of-the-art results, confirming its generalizability and capability to model protein-protein interactions.
>
> **[... Can La-Proteina compare on the folding benchmark?]**
>
> We agree with the reviewer about folding being regarded as a full-atom conditional generation task. However, our work is focused on atomistic protein design, with the goal of creating a scalable, high-performance generative model for de novo design, rather than to predict the structure of an existing sequence. We do demonstrate La-Proteina's conditional capabilities on tasks that are central to protein design, such as atomistic motif scaffolding (both all-atom and tip-atom, indexed and unindexed), and show that La-Proteina achieves state-of-the-art performance. While we believe that La-Proteina could be adapted for folding, achieving competitive performance on folding benchmarks would require a different training regime, likely including an MSA pipeline for co-evolutionary information. We consider this a separate and significant research endeavor, which is beyond the scope of our current work on establishing a foundational, scalable model for atomistic design.
>
> **[... equivariant and non-equivariant methods use a fast scheduler during inference. Do you have any insights or possible explanations for this?]**
>
> This is an interesting point shared by all of these methods. While we don't have a definitive answer, we can offer our insights. As the reviewer notes, moving away from the "naive" uniform discretization sampling process is critical for performance. The top-performing all-atom models in our work, our La-Proteina and P(all-atom), both effectively generate the backbone structure at a faster rate than the side-chain and sequence details. La-Proteina does this explicitly. Our partially latent model allows us to use independent schedules for alpha-carbons and latent variables. On the other hand, P(all-atom) does this implicitly through its atom14 representation, under which side-chain atoms are noised faster in the forward (noising) process and thus generated later (i.e., on a slower effective schedule) in the reverse (generative) process.
>
> We hypothesize that this faster generation for the backbone is beneficial due to the different properties of the mappings between backbone -> sequence/side-chain and sequence/side-chain -> backbone. It is possible that one of these conditional directions is easier for the model to learn, and the schedules mentioned above allow the model to leverage this asymmetry during generation. Furthermore, these schedule choices interplay with low-temperature sampling in non-trivial ways. It is plausible that low-temperature sampling has a different impact on the explicit alpha-carbon coordinates versus the compressed latent variables, which in turn influences which schedules yield the best performance.
>
> The reviewer's connection to FrameFlow and FoldFlow is also appreciated. Their task is different, as they focus on backbone-only generation. In their case, they model backbones as rotations and translations, running flow matching on manifolds, and observe that accelerating the rotation schedule improves performance. We unfortunately don't have strong intuition about this, as their method operates very differently from our all-atom, Euclidean-space approach.
>
> -----
>
> We sincerely thank the reviewer again for the positive review.

---

> > ### Comment · Reviewer_4Tcg · 2025-11-27
> >
> > I appreciate the authors' detailed response. I choose to maintain my score of 10 and strongly recommend acceptance.

---

> > > ### Author Response · Authors · 2025-11-28
> > > **Thanks**
> > >
> > > Thank you for your feedback and positive review of our work!

---

### Author Response · Authors · 2025-11-21
**General Answer**

We thank the reviewers for the positive comments about our work and constructive feedback. We address individual comments / questions in separate replies to each of the reviews. All updates to the manuscript are shown in red in the PDF, to make them easily identifiable.

In this general answer we address a question that came up in multiple reviews, about using La-Proteina for tasks beyond monomer design,  such as binder design, or general protein complex design tasks.

We consider this a very interesting point, highlighting a critical task for protein design. Our primary goal for this paper was to introduce and rigorously validate La-Proteina as the first highly scalable and high performing method for fully atomistic protein design. Developing the model and proving its scalability (up to 800 residues) and capacity to generate biophysically realistic structures was a challenge in itself. We chose to focus on the monomer tasks as the cleanest and most direct way to validate our core methodological contributions.

However, the La-Proteina framework is not fundamentally limited to monomers and can be trained on multi-chain data. We viewed the extension to protein complexes as an important “second step”, which is a large endeavor on its own. We are happy to report that this extension is not just hypothetical. As mentioned in our related work section, the concurrent submission “Scaling atomistic protein binder design with generative pretraining and test-time
compute” (Anonymous, 2025) does exactly this: it extends La-Proteina framework to the task of atomistic binder design. That work demonstrates that La-Proteina can be extended to yield state-of-the-art results for binder design, confirming that our partially latent approach is effective for modeling protein-protein interactions.

In summary, we deliberately limited the scope of this paper to thoroughly establish the method. But we can confirm the reviewers’ intuition is correct: the framework generalizes to multi-chain systems, as demonstrated in the concurrent submission.

---

### Meta-Review · Area_Chair_Pc6B · 2026-01-09

**Summary:**

This paper is perhaps the first or certainly among the first to consider full atom protein generation. The results are really cool!

The concerns raised were:
- Needing to tune two noise schedules.
- Very large scale training costs
- Focus on monomers / unconditional generation (these are implicitly the same complaint, since conditional generation usually means "we want binders").
- Some asks for more analysis and metrics.

**Reviewer Concerns:**

I generally agree with the authors that extending beyond monomers seems like a contribution in its own right, which seemed like probably the biggest ask. Everything looks well handled enough...

**Reviewer Scores:**

...Regardless, all of the reviewers were already extremely positive about the paper -- it's hard to imagine based on the rebuttals given that any of the reviewers would have been induced to lower their scores by 4-5 points. Looking at the paper myself, I generally agree it seems good.

---

### Decision · Program_Chairs · 2026-01-26

Accept (Poster)